# Tackling Heavy-Tailed Q-Value Bias in Offline-to-Online Reinforcement Learning with Laplace-Robust Modeling

**Ruibo Guo[1], Lei Liu[1] *, Rui Yang[2], Junjie Shen[1], Guoping Wu[1], Jie Wang[1], Bin Li[1]**
[1]University of Science and Technology of China
[2]China Mobile
{ruiboguo, yr0013, shenjunjie, guoping}@mail.ustc.edu.cn
{liulei13, jiewangx, binli}@ustc.edu.cn

## Abstract

Offline-to-online reinforcement learning (O2O RL) aims to improve the performance of offline pretrained agents through online fine-tuning. Existing O2O RL methods have achieved advances in mitigating the overestimation of Q-value biases (i.e., biases of cumulative rewards), improving the performance. However, in this paper, we are the first to reveal that Q-value biases of these methods often follow a heavy-tailed distribution during online fine-tuning. Such biases induce high estimation variance and hinder performance improvement. To address this challenge, we propose a Laplace-based robust offline-to-online RL (LAROO) approach. LAROO introduces a parameterized Laplace-distributed noise and transfers the heavy-tailed nature of Q-value biases into this noise, alleviating heavy tailedness of biases for training stability and performance improvement. Specifically, (1) since Laplace distribution is well-suited for modeling heavy-tailed data, LAROO introduces a parameterized Laplace-distributed noise that can adaptively capture heavy tailedness of any data. (2) By combining estimated Q-values with the noise to approximate true Q-values, LAROO transfers the heavy-tailed nature of biases into the noise, reducing estimation variance. (3) LAROO employs conservative ensemble-based estimates to re-center Q-value biases, shifting their mean towards zero. Based on (2) and (3), LAROO promotes heavy-tailed Q-value biases into a standardized form, improving training stability and performance. Extensive experiments demonstrate that LAROO achieves significant performance improvement, outperforming several state-of-the-art O2O RL baselines.

## 1 Introduction

Offline-to-online reinforcement learning (O2O RL) has drawn considerable attention in recent research, concentrating on enhancing the performance of offline pretrained agents by online fine-tunings (Lee et al., 2021b; Nair et al., 2020; Zhao et al., 2022). Since the performance of pretrained agents is heavily limited by the quality and state-action space coverage of their offline datasets (Jin et al., 2021), O2O RL proposes to fine-tune pretrained agents in online environments. It has the appealing potential to achieve notable performance improvement within just a few online environmental interaction steps (Feng et al., 2024).

Recently, several O2O RL methods (Lee et al., 2021b; Feng et al., 2024) highlight that pretrained agents confront a significant state-action distribution shift between offline and online data during the fine-tuning process. This shift causes pretrained agents to inaccurately estimate Q-values in online data and misguide update directions, resulting in slow performance improvement and instability during online fine-tuning (Zhang et al., 2024). Therefore, the aforementioned methods employ various techniques to improve the Q-value estimation. Specifically, some methods incorporate a conservative penalty into the estimated Q-values (Zhao et al., 2022; Nakamoto et al., 2023) or employ ensemble models (Zhao et al., 2022; 2024) to enable conservative and stable Q-value estimation in

---

*Corresponding author.

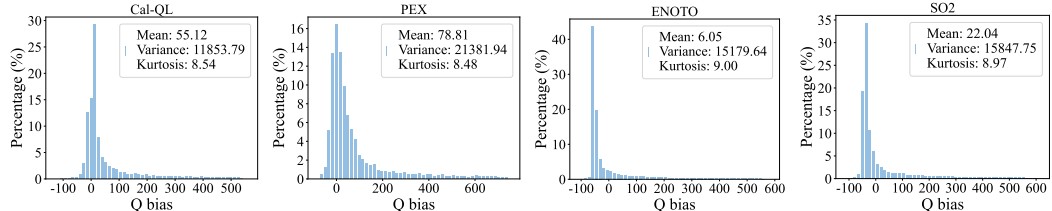

Figure 1: **The heavy-tailed Q bias distribution**. We present Q bias distributions of four O2O RL methods (Cal-QL (Nakamoto et al., 2023), PEX (Zhang et al., 2023), ENOTO (Zhao et al., 2024), SO2 (Zhang et al., 2024)) for online fine-tuning at the 100k online step in the Walker2d-medium task. Please refer to Figure 5 and Figure 6 for more training steps and various tasks.

online data. Other methods (Feng et al., 2024; Zhang et al., 2024) propose to increase the update frequency of models to facilitate the accurate Q-value estimation.

In this paper, based on a comprehensive study of the Q-value estimation in O2O RL, we reveal that Q-value estimation bias (i.e., Q bias) often follows a heavy-tailed distribution during fine-tuning for the first time. In Figure 1, we illustrate the distribution of Q bias for several O2O RL methods. The results clearly show that the Q bias displays substantially heavier tails than the normal distribution, and exhibits positive skewness with notably high biases in the right tail. This heavy-tailed property of Q bias is inherently caused by distribution shift problem and persists throughout online fine-tuning. Consequently, the heavy-tailed Q bias challenges the widely adopted assumptions of finite variance or Gaussian modeling for such biases in prior research (Duan et al., 2021; D'Eramo et al., 2021b; Chen et al., 2021), thereby questioning their broad applicability.

In practice, the heavy-tailed Q bias significantly affects both Q-value estimation and policy updates in O2O RL. These heavy-tailed biases lead to extreme overestimation of Q-values and misguide agents to choose poor actions, causing large performance degradation during online fine-tuning. Meanwhile, high Q biases can lead to large fluctuations and even collapse of Q-value estimation (Zhang et al., 2024; Yang et al., 2024). It is widely recognized that heavy-tailed nature of bias induces large estimation variance and impedes the convergence of Q-values, resulting in slow performance improvement (Shao et al., 2018; Zhuang & Sui, 2021). Since the aforementioned O2O methods fail to account for these heavy-tailed Q biases and mitigate their impact, these methods continue to face challenges of instability and inefficiencies in performance improvement.

To tackle these challenges, we propose a novel **La**place-based **r**obust **o**ffline-to-**o**nline RL (LAROO) approach. The central idea of LAROO is to explicitly model and mitigate heavy-tailed Q-value biases by transferring their heavy-tailed nature into a parameterized Laplace-distributed noise. Since the Laplace distribution is well-suited for modeling heavy-tailed data (Bai, 1995; Yang et al., 2019), this formulation accurately represent the nature of Q-value biases, thus enhancing the Q-value estimation. Specifically, LAROO is built upon three key components: (1) **Adaptive laplace-distributed noise modeling**. LAROO introduces a parameterized Laplace-distributed noise that adaptively captures heavy-tailed characteristics, providing a principled way to represent such heavy-tailed distributions. (2) **Variance reduction via a learning process that incorporates noise**. By combining estimated Q-values with the Laplace-distributed noise to approximate the true Q-values, LAROO transfers the heavy-tailed nature of biases into the noise distribution, effectively reducing the high estimation variance of Q values. (3) **Bias Re-centering**. For stability, LAROO further employs conservative ensemble-based estimates to re-center Q-value biases, shifting their mean toward zero. By jointly leveraging (2) and (3), LAROO promotes the heavy-tailed Q-value biases into a standardized form, thereby significantly enhancing training stability and performance.

This study incorporates the Laplace-distributed noise model and conservative Q-value ensemble models for robust Q-value estimation in O2O RL, facilitating efficient and stable performance improvement. It extends prior analysis of Q bias beyond the mean or variance, to the full distribution in O2O RL. Moreover, we theoretically show that LAROO reduces the single-step estimation bias compared with the $l_2$ loss during training, thereby accumulating less bias over time. Experimental results show that LAROO represents a meaningful advancement by effectively mitigating the impact of heavy-tailed Q bias. We summarize our contributions as follows:

- To the best of our knowledge, for the first time, we reveal that the Q-value estimation bias in existing O2O methods often follows a heavy-tailed distribution during online fine-tuning, which leads to instability and inefficiencies in performance improvement in O2O RL.

- We propose LAROO, a Laplace-based robust offline-to-online RL approach that introduces Laplace-distributed noise to alleviate heavy-tailed biases and uses conservative ensemble-based estimates to re-center bias, improving training stability and performance.

- We provide a theoretical analysis demonstrating that LAROO reduces single-step estimation bias compared to the conventional $l_2$ loss function used in Q-value updates.

- Extensive experiments demonstrate that LAROO achieves significant performance improvement, outperforming several state-of-the-art baselines across various tasks.

## 2 PRELIMINARIES

We follow the standard protocol that formulates a RL environment as a Markov decision process (MDP). The MDP $\mathcal{M}$ is often described as a tuple $\mathcal{M} = \langle \mathcal{S}, \mathcal{A}, \mathbb{P}, R, \gamma \rangle$, where $\mathcal{S}$ is the state space, $\mathcal{A}$ is the action space, $\mathbb{P} : \mathcal{S} \times \mathcal{A} \to \Delta(\mathcal{S})$ is the transition function, $R : \mathcal{S} \times \mathcal{A} \to \mathbb{R}$ is the reward function and $\gamma \in [0, 1)$ is the discount factor. The agent aims to learn a policy $\pi(a|s)$ that maximizes the expected return $\mathbb{E}_\pi[\sum_{t=0}^{\infty} \gamma^t r_t]$, where $r_t = R(s_t, a_t)$.

Off-policy RL methods employ a Q-value network $Q_\theta(s, a)$ to estimate the expected cumulative discounted return starting from the state-action pair $(s, a)$, defined as $\mathbb{E}_\pi[\sum_{t=0}^{\infty} \gamma^t r_t \mid s_0 = s, a_0 = a]$, which is also known as the true Q-value of $(s, a)$. However, the Q-value network inevitably introduces estimation biases in estimated Q-values, i.e., Q biases. Off-policy RL employs policy evaluation (i.e., Q-learning) to update the Q-value network $Q_\theta(s, a)$:

$$J_\theta(Q_\theta) := \mathbb{E}_{(s,a,r,s') \sim \mathcal{B}} \Big[ \big( Q_\theta(s, a) - \mathcal{T}Q_\theta(s, a) \big)^2 \Big] \tag{1}$$

where $\mathcal{B}$ is the replay buffer, $\mathcal{T}Q_\theta(s, a)$ is the Q-value update target defined as $R(s, a) + \gamma \mathbb{E}_{s' \sim T(\cdot|s,a)} \big[ \max_{a'} Q_{\hat{\theta}}(s', a') \big]$, and the temporal difference (TD) error is denoted as $\delta_\theta(s, a) = Q_\theta(s, a) - \mathcal{T}Q_\theta(s, a)$. In this study, given a $(s, a)$ pair, we denote the true Q-value as $\mathcal{Q}(s, a)$.

## 3 Q-VALUE ESTIMATION BIAS IN O2O RL

In this section, we reveal the heavy-tailed Q bias and analyze their influence. We further highlight the limitations of previous O2O methods in effectively mitigating their impact.

***The estimation bias follows a heavy-tailed distribution in online data during fine-tuning***. We evaluate several existing offline and O2O RL methods across various tasks and compute the Q bias in online data. Specifically, we collect online data into the replay buffer at every online step. For each data, we obtain its true Q-value using Monte Carlo strategy and the estimated Q-value with the Q network, then calculate their difference to get the Q bias. As shown in Figure 1, 5 and 6, these results present that Q biases exhibit a heavy-tailed nature in online data, and this heavy-tailed nature is universal across various tasks and persists throughout the online fine-tuning process.

We employ the metric Kurtosis $\kappa = \frac{1}{n} \sum_{i=1}^{n} (x_i - \overline{x})^4 / \left( \frac{1}{n} \sum_{i=1}^{n} (x_i - \overline{x})^2 \right)^2$ to measure the degree of heavy-tailedness relative to a normal distribution (Mardia, 1970; Garg et al., 2021). Additionally, we compute the mean and variance of the Q bias. A positive mean indicates Q-value overestimation, whereas a negative mean indicates underestimation. Our observations reveal that in O2O RL, offline algorithms typically underestimate Q-values, whereas existing O2O RL methods tend to overestimate them during fine-tuning. Importantly, regardless of whether the Q-values are overestimated or underestimated, the Q bias consistently demonstrates high heavy-tailedness. We present detailed experiments for Q bias in Appendix A.2.

***Explaining the Heavy-Tailed Q-Bias in Offline-to-Online RL***. The heavy-tailed nature of Q-bias is attributed to two main factors: (i) the Q-function with Max operator amplifies overestimation by selecting a large bellman target, and (ii) a non-uniform distribution shift between online and offline data. Specifically, pretrained Q-value networks have large Q bias in online samples that lie far from the offline data distribution, which manifest as the Q bias distribution's tails. In contrast, online

samples closer to the offline distribution result in minor biases and constitute the central mass of Q bias distribution.

We validate factor (ii) through experiments by computing the shifted distance of online data from the offline distribution using two distance metrics. We validate a positive correlation between Q bias and shifted distance with Spearman's rank correlation coefficient. As shown in Figure 2, the Q bias of online samples tends to increase as their distance from the offline data distribution grows. Please refer to Appendix A.3 for details.

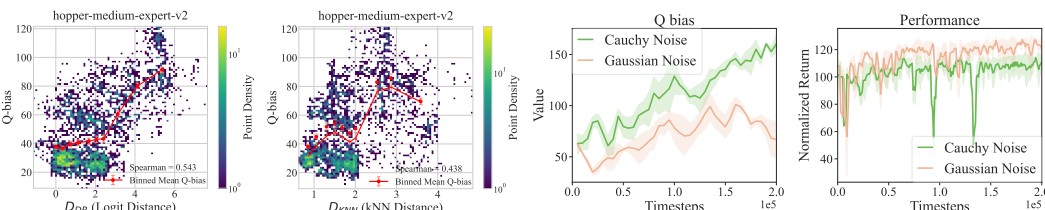

Figure 2: Left figures: **The non-uniform distribution shift contributes to heavy-tailed Q bias**. The Q bias of online samples often increases as their distance ($D_{KNN}$, $D_{DR}$) from offline data distribution grows. Right figures: **Validation of the instability caused by heavy-tailed Q biases**. Heavy-tailed Cauchy noise causes greater training instability and lower performance.

***The heavy-tailed Q bias could induce instability during the online fine-tuning process***. When Q bias exhibits high heavy-tailedness, the back propagation of Q-learning amplifies these extreme biases over time, exacerbating this issue. Additionally, large Q biases result in inaccurately estimated Q-values, which may cause Q-value estimation collapse (Zhang et al., 2024). As a result, additional training steps are required to correct these inaccuracies, resulting in inefficient performance improvement. Furthermore, existing O2O RL methods often rely on the $l_2$ loss function for Q-value updates. This reliance may cause heavy-tailed Q biases to disproportionately influence the Q-value loss and dominate the update gradients, further destabilizing Q-value updates.

We also validate the issues caused by the heavy-tailed Q bias through experiments. For approximating the heavy-tailed Q bias, we incorporate heavy-tailed noise into the estimated Q-values during Q-value updates. Specifically, for a fair comparison, we respectively introduce Gaussian noise and heavy-tailed Cauchy noise, both with the same mean and variance, into two agents from the same pretrained one. The results in Figure 9 show that the heavy-tailed Cauchy noise leads to more inaccuracies in Q-value estimation and lower final performance. See details in Appendix A.4.

***It is necessary to take into account the heavy-tailed Q bias in Q-value estimation***. Previous research mainly focuses on reducing the mean or variance of Q bias, neglecting its underlying distributional characteristics. For instance, Cal-QL (Nakamoto et al., 2023) employs conservative Q-learning to decrease positive Q bias. ENOTO (Zhao et al., 2024) introduces ensemble models with SUNRASE (Lee et al., 2021a) to reduce the mean and variance of Q bias. However, as results shown in Figure 5 and 6, while these methods can effectively reduce the mean of Q bias, they still struggle to decrease its variance, and the kurtosis values remain high. This phenomenon highlights that existing methods still struggle to mitigate heavy-tailed Q biases and reduce their impact. Therefore, it is crucial to address these biases in Q-value estimation to enhance training stability.

In summary, in offline-to-online RL, the Q bias exhibits high heavy-tailedness in online data during fine-tuning, regardless of whether the Q-value is overestimated or underestimated. These heavy-tailed Q biases arise from the non-uniform distribution shift and cause Q-value estimation to oscillate, further resulting in training instability and slow performance improvement. Therefore, it is essential to take into account the distribution of Q bias and enhance the training stability of agents under these heavy-tailed Q biases for stable fine-tuning.

## 4 METHOD

In Section 4.1, we introduces an adaptive noise following a Laplace distribution to explicitly capture the heavy-tailed nature of Q-value biases, improving Q-value estimation. We propose a robust O2O approach in Section 4.2 and provide a theoretical analysis for its effectiveness in Section 4.3.

## 4.1 Laplace-based Q bias Distribution for Q-value Estimation

In robust regression research (Song et al., 2014; Lu & Chang, 2022), a promising approach to handle non-normal error distributions is modeling each error distribution as a Laplace distribution, which is known for its ability to model data with outliers and connection with the robust regression criteria.

Inspired by aforementioned approaches, we introduce an independent approximation noise term using the Laplace distribution to explicitly capture the heavy-tailed characteristic of Q bias. Then, we incorporate the Laplace-based noise into the Q-value loss for robust Q-value estimation.

***Adaptive Laplace-distributed noise modeling for heavy-tailed Q bias***. To facilitate the analysis of estimation bias, we begin with the assumption for the noise term following (Duan et al., 2021):

**Assumption 4.1** *For each $(s, a)$ pair, $\mathcal{Q}(s, a)$ is the true but unknown Q-value, $\varepsilon_{\hat{\theta}}$ and $\varepsilon_{\theta}$ are independent random variables, which denote the approximation noise in target Q-value and estimated Q-value respectively. The approximation noise is assumed to be independent from the $(s, a)$ and $\theta$. Then we have:*

$$\mathcal{T}Q_\theta(s, a) = R(s, a) + \gamma \mathbb{E}_{s' \sim T(\cdot|s,a)} \left[ max_{a'} Q_{\hat{\theta}}(s', a') \right]$$
$$\mathcal{Q}(s, a) = \mathcal{T}Q_\theta(s, a) + \varepsilon_{\hat{\theta}} \qquad \mathcal{Q}(s, a) = Q_\theta(s, a) + \varepsilon_\theta \tag{2}$$

The noise characterizes the discrepancy between true Q-values and their estimates, arising from system noises and function approximation (Duan et al., 2021). Based on the definition of Q-value bias, $\text{Bias}(Q_\theta(s, a)) = \mathbb{E}[Q_\theta(s, a)] - \mathcal{Q}(s, a)$, we obtain the relationship $\text{Bias}(Q_\theta(s, a)) = -\mathbb{E}[\varepsilon_\theta]$, Then, to better capture the heavy-tailed Q bias, we model the noise using the Laplace distribution as follows, and update it adaptively.

**Assumption 4.2** *(Laplace-based noise). $\varepsilon_{\hat{\theta}}$ and $\varepsilon_{\theta}$ follow Laplace distributions with the same mean $\mu$, i.e., $\varepsilon_{\hat{\theta}} \sim \text{Laplace}(\mu, b_1)$, $\varepsilon_\theta \sim \text{Laplace}(\mu, b_2)$, $\mu \in \mathbb{R}$, $b_1, b_2 \in \mathbb{R}^+$.*

Here, $\mu$ is the mean parameter and $b$ is the scale parameter. Specifically, larger values of $b$ correspond to heavier tails and higher variance in the Laplace-based noise. We introduce the adaptive updates of noise in next section. Based on the assumption above, we can derive the Laplace-based likelihood of $\mathcal{Q}(s, a)$ given $Q_\theta(s, a)$, and the Laplace-based likelihood of $\mathcal{Q}(s, a)$ given $\mathcal{T}Q_\theta(s, a)$:

$$q\Big(\mathcal{Q}(s, a)\big|Q_\theta(s, a); \theta\Big) = \frac{1}{2b_2} \exp\left(-\frac{|\mathcal{Q}(s, a) - Q_\theta(s, a) - \mu|}{b_2}\right)$$
$$p\Big(\mathcal{Q}(s, a)\big|\mathcal{T}Q_\theta(s, a); \hat{\theta}\Big) = \frac{1}{2b_1} \exp\left(-\frac{|\mathcal{Q}(s, a) - \mathcal{T}Q_\theta(s, a) - \mu|}{b_1}\right) \tag{3}$$

***The Laplace likelihood incorporates the distributional information of Q bias for robust Q-value updates***. Then, we discuss how to update the Q-value network and how to reduce the impact of extreme biases under the Assumption 4.2. Since the likelihood $p\left(\mathcal{Q}(s, a) \mid \mathcal{T}Q_\theta(s, a)\right)$ incorporates true environmental return information (i.e., future rewards), it serve as a closer approximation to the Bayes-optimal posterior (i.e., $p(\mathcal{Q}_\theta(s, a)|\text{all environmental evidence})$), compared with the $q\left(\mathcal{Q}(s, a) \mid Q_\theta(s, a)\right)$. We explain this process in detail at Appendix B.2. Therefore, we can train Q-network by minimizing the Kullback-Leibler (KL) divergence between likelihoods of the true Q-value, i.e., $p\left(\mathcal{Q}(s, a) \mid \mathcal{T}Q_\theta(s, a)\right)$ and $q\left(\mathcal{Q}(s, a) \mid Q_\theta(s, a)\right)$.

Given a dataset $\mathcal{D} = \{(s_i, a_i, s'_i, r_i)\}_{i=0}^N$, the Q-value network $Q_{\theta_t}$ and the target Q-network $Q_{\hat{\theta}_t}$ at the $t$-th training step, we can estimate $\theta_{t+1}$ as Equation 4:

$$\theta_{t+1} = \arg\min_\theta \sum_{i=0}^N D\Big(p\left(\mathcal{Q}(s_i, a_i)\big|\mathcal{T}Q_{\theta_t}(s_i, a_i)\right) \Big\| q\Big(\mathcal{Q}(s_i, a_i)\big|Q_{\theta_t}(s_i, a_i)\Big)\Big) \tag{4}$$

By integrating over all values of $\mathcal{Q}(s_i, a_i)$, the KL divergence can be transformed as Equation (5). See proof in Appendix B.2. Based on the Bellman equation, the Q-network progressively approximate the true Q-value.

$$D_{KL}\Big(p\Big(\mathcal{Q}(s_i, a_i)\big|\mathcal{T}Q_\theta(s_i, a_i)\Big)\Big\|q\Big(\mathcal{Q}(s_i, a_i)\|Q_\theta(s_i, a_i)\Big)\Big)$$
$$= \frac{b_1 \exp\left(-\frac{|\mathcal{T}Q_\theta(s_i, a_i) - Q_\theta(s_i, a_i)|}{b_1}\right)}{b_2} + \frac{|\mathcal{T}Q_\theta(s_i, a_i) - Q_\theta(s_i, a_i)|}{b_2} + \log\frac{b_2}{b_1} - 1 \tag{5}$$

We now discuss how the KL divergence addresses the issue of heavy-tailed Q bias, leading to a more robust training process. When using a Laplace likelihood $q(\mathcal{Q}(s,a) \mid Q_\theta(s,a))$, the resulting negative log-likelihood is proportional to $|\mathcal{Q}(s,a) - Q_\theta(s,a)|$, which grows linearly rather than quadratically with the Q bias. This down-weights outliers and drives optimization using the central mass of the Q bias, rather than being influenced by rare, extreme biases in the tails.

The Laplace likelihood also better preserves the distributional information of Q bias compared to other robust functions (e.g, $l_1$ loss). If the heavy-tailed bias in bellman target is ignored, i.e., the $p(\mathcal{Q}(s,a) \mid \mathcal{T}Q_\theta(s,a))$ is a Dirac delta function centered on $\mathcal{T}Q_\theta(s,a)$ , the KL divergence reduces to the negative log likelihood $-q(\mathcal{T}Q_{\hat\theta}(s,a)|Q_\theta(s,a))$, which is exactly equivalent to applying an $l_1$ loss under a Laplace-noise assumption. Detailed analysis is provided in Appendix B.2.

***Robust Q-value loss function with Laplace noise modeling***. We propose a robust function $D_b(x)$ in replace of $l_2$ function for Q-value updates. Compared with Equation 5, we set $b = b_1 = b_2$ for practical implementation with fewer hyperparameters. This setting imply states that the $\varepsilon_\theta$ and $\varepsilon_{\hat\theta}$ follow the same distribution. We conduct the Mann-Whitney U test (Hettmansperger & McKean, 2010) to validate it in C.2.1. Furthermore, we show the derivatives of $D_b(x)$ even at $x = 0$ and provide its more details in B.3.

$$D_b(x) = \exp\left(-\frac{|x|}{b}\right) + \frac{|x|}{b} - 1 \tag{6}$$

Finally, we demonstrate the robustness of $D_b(x)$ in the context of online fine-tuning through two properties: (1) the exponential term $\exp(-|x|/b)$ down-weights large Q-value losses in the heavy tails, reducing their impact. (2) the gradient is bounded within the range $[-\frac{1}{b}, \frac{1}{b}]$ and does not increase with $x$, in contrast to the $l_2$ function, where the gradient increases linearly with $x$. These properties validate the effectiveness of the KL divergence in Equation 4.

We observe that the robust property of $D_b(x)$ is inherently connected with the Laplace distribution: as the heavy-tailed Q biases become more frequent, the Laplace-based noise capture the heavy-tailedness by increasing its scale parameter, the gradient bound of $D_b(x)$ tightens. Consequently, these mechanisms alleviate heavy-tailedness of Q biases, reduce the high estimation variance with noise model, and improve training stability with $D_b(x)$ function.

## 4.2 A ROBUST O2O RL METHOD WITH LAPLACE DISTRIBUTED Q BIAS

We develop a complete O2O method with the Laplace modeling for Q bias in this section.

***Adaptively Updating the Laplace-Distributed Noise with Robust Statistics***. To effectively capture the heavy-tailed nature of Q biases with our Laplace noise model, it must adapt to the changing distribution of these biases during online fine-tuning. This requires continuously updating the parameters of the Laplace distribution, particularly the scale (variance) parameter $b$. However, estimators of unbiased sample variance are highly sensitive to kurtosis and unreliable in the presence of significant outliers (Yuan et al., 2005). Robust variance estimators that account for kurtosis have been proposed in prior work (Searls & Intarapanich, 1990; Wencheko & Chipoyera, 2009). Among these, we adopt the MSE-best biased estimator (MBBE) (Wencheko & Chipoyera, 2009), defined as follows.

**Lemma 4.3** *(MBBE of variance).* $s^2$ *is the unbiased sample variance* $s^2 = \frac{1}{n-1}\sum_{i=1}^n (X_i - \bar{X})^2$, $n$ *is the sample size and* $\kappa$ *is the population kurtosis. Then, MSE-best biased estimator* $s_\omega^2$ *is:*

$$s_{\omega^*}^2 = \left(\frac{\kappa}{n} + \frac{n+1}{n-1}\right)^{-1} s^2 \tag{7}$$

In practice, the true Q biases are not accessible during training. We use the readily available TD-error as a statistical surrogate to approximate the variance of Q-bias. This surrogate is reasonable for variance approximation, because we have $\mathcal{T}Q_{\hat\theta}(s,a) - Q_\theta(s,a) = \big(\mathcal{Q}(s,a) - \varepsilon_{\hat\theta}\big) - \big(\mathcal{Q}(s,a) - \varepsilon_\theta\big) = \varepsilon_\theta - \varepsilon_{\hat\theta}$ under Assumption 4.1. Then the variance of TD-errors twice the variance of Q bias. We then compute the scale parameter using the MBBE of the TD-errors within a batch ($b = s_{w^*}/\sqrt{2}$). Our empirical experiments confirm that the scale parameter $b$ estimated from TD-errors closely tracks the one estimated from true Q biases, as shown in Figure 17. Crucially, we can adaptively update our noise model to capture the heavy-tailedness of Q bias and improve Q-value estimation according to Section 4.1.

***Re-centering Heavy-Tailed Q Bias through Ensemble Models***. In our empirical analysis, we observed that large-magnitude Q biases in the heavy tails persist under standard training. To mitigate this, we adopt ensemble models known for effectively reducing estimation bias, and compute the target Q-value by taking the minimum over a random subset of Q-function estimators (Chen et al., 2021). This strategy effectively reduces highly positive biases and re-centers the Q bias distribution because: (1) different Q-functions are approximately independent, positive errors and negative errors are independently sampled across different heads. (2) the probability of selecting the most overoptimistic head is reduced. Then, we integrate the ensemble model with $D_b(x)$ and derive the loss function for robust Q-value estimation as Equation 8, where $K$ is the ensemble size and $M$ is the subset size.

$$\mathcal{L}_b(\theta_t) = \mathbb{E}_{(s,a,r,s')\sim\mathcal{B}}\left[\frac{1}{K}\sum_{k=1}^{K}D_b\left(Q_{\theta_t}^{(k)}(s,a) - y_{\min}(s,a,r,s')\right)\right]$$

$$y_{\min}(s,a,r,s') = r + \gamma\max_{a'\in\mathcal{A}}\left[\min_{1\le k\le M}Q_{\hat{\theta}_t}^{(k)}(s',a')\right].$$

(8)

LAROO mainly consists of two components: the noise model and the ensemble models, which together tackle heavy-tailed biases through two primary effects: (1) By selecting a lower Q-value from the ensemble models, LAROO explicitly re-centers the Q bias distribution, shifting its mean towards zero and mitigating the strong overestimation introduced by large Q biases. (2) Based on the Laplace-based noise model, LAROO alleviates the heavy-tailed Q biases and reduces their impact with robust function $D_b(x)$ on training stability, as discussed in Section 4.1.

In summary, LAROO transforms the poorly-behaved, heavy-tailed Q-bias distribution into a more standardized form during Q-value updates. As a result, LAROO effectively reduces the estimation variance and mitigates heavy-tailed Q biases, improving training stability and efficiency without the need for exploration constraints under distribution shifts. Additionally, we compare it with the standard batch normalization in Appendix B.7 and summarize the complete framework of LAROO in Algorithm 1.

## 4.3 The theoretical analysis

We further provide a theoretical analysis to prove that LAROO can reduce estimation bias compared with the $l_2$ loss function for Q-value updates. We begin to introduce the estimate bias in single update step, following prior research (Duan et al., 2021). For simplicity, we denote the greedy target $R(s,a)+\gamma\mathbb{E}_{s'\sim T(\cdot|s,a)}\left[\max_{a'}Q_{\hat{\theta}}(s',a')\right]$ as $y$. Conventionally, the Q-value network $Q_\theta$ is updated by minimizing the mean square loss $\left(y-Q_\theta(s,a)\right)^2/2$ using gradient descent methods, resulting in $Q_{\theta_{new}}$. Then, the post-update Q-value $Q_{\theta_{new}}(s,a)$ can be approximated by linearizing around $\theta_{new}$ using Taylor's expansion, as shown in Equation 9, where $\beta$ is the learning rate and is sufficiently small.

$$\theta_{new} = \theta + \beta(y - Q_\theta(s,a))\nabla_\theta Q_\theta(s,a)$$
$$Q_{\theta_{new}}(s,a) \approx Q_\theta(s,a) + \beta(y - Q_\theta(s,a))\|\nabla_\theta Q_\theta(s,a)\|_2^2$$

(9)

Similarly, let $\theta_{id}$ represent the ideal post-update parameter obtained based on true target $\tilde{y} = R(s,a) + \gamma\mathbb{E}_{s'}\left[\max_{a'}\mathcal{Q}(s',a')\right]$, and we can approximate the ideal post-update Q-value $Q_{\theta_{id}}(s,a)$ as:

$$Q_{\theta_{id}}(s,a) \approx Q_\theta(s,a) + \beta(\tilde{y} - Q_\theta(s,a))\|\nabla_\theta Q_\theta(s,a)\|_2^2$$

(10)

Then, in expectation, we can define the single-step estimation bias of the $l_2$ update, denoted as:

$$\Delta_{l_2}(s,a) = \mathbb{E}_{\varepsilon_{\hat{\theta}}}\left[Q_{\theta_{new}}(s,a) - Q_{\theta_{id}}(s,a)\right] \approx \beta\left(\mathbb{E}_{\varepsilon_{\hat{\theta}}}[y] - \tilde{y}\right)\|\nabla_\theta Q_\theta(s,a)\|_2^2$$

where $\varepsilon_{\hat{\theta}}$ is the approximation noise in target $y$. Moreover, previous research (Duan et al., 2021; Hasselt, 2010) has verified that the estimate Q-value is usually overestimated due to the max operator and it is clear that: $\mathbb{E}_{\varepsilon_{\hat{\theta}}}[y] - \tilde{y} \ge 0, \quad \Delta_{l_2}(s,a) \ge 0$.

Building on definitions above, we now demonstrate the robustness of LAROO in Q-value estimation. When the Q-value network is updated with $D_b(x)$ function, the following theorem holds:

**Theorem 4.4** *When $b > 1$, the single-step estimation bias of post-update Q-value $Q_{\theta_{new}}$ with function $D_b(x)$ is smaller than that with $l_2$ loss function, i.e., $\Delta_{D_b} < \Delta_{l_2}$.*

Table 1: **Normalized returns before and after the online fine-tuning**. Each result is averaged with five seeds. $\delta_{\text{sum}}(0.1\text{M})$ denotes the sum of performance improvement on all tasks within 0.1M steps. The best results are highlighted in bold, while the second-best results are underlined. The gray "offline" refers to the offline performance of both BOORL and LAROO, which use the same offline backbones. Please refer to Appendix C.2.2 for more results and training curves.

| Task | Type | PEX | SO2 | Cal-QL | ENOTO | Offline | BOORL | LAROO |
|------|------|-----|-----|--------|-------|---------|-------|-------|
| Hopper | random | 6.6→17.5 | 12.9→87.3 | 7.3→6.9 | 7.4→66.5 | 7.9 | 88.3 | **100.6±10.3** |
| | medium | 49.1→78.5 | 59.7→94.4 | 70.5→90.8 | 57.1→96.6 | 56.6 | 102.1 | **106.7±2.6** |
| | medium-replay | 60.0→87.5 | 98.9→100.3 | 89.2→92.8 | 78.2→102.7 | 80.5 | 105.1 | **109.5±1.8** |
| | medium-expert | 85.4→80.2 | 99.3→98.7 | 104.9→108.3 | 94.5→100.2 | 93.6 | 107.5 | **112.3±0.7** |
| | expert | 65.7→73.8 | 86.7→84.7 | 106.0→110.0 | 111.3→85.6 | 109.2 | 103.2 | **112.0±1.3** |
| Walker2d | random | 7.9→37.0 | 4.7→20.8 | 9.7→1.6 | 0.9→38.3 | 1.5 | 6.4 | **71.6±6.8** |
| | medium | 60.5→42.5 | 91.7→100.6 | 83.3→83.7 | 83.6→110.2 | 83.6 | 98.6 | **120.4±2.9** |
| | medium-replay | 32.8→40.5 | 53.7→92.0 | 75.1→80.7 | 78.0→110.2 | 76.8 | 104.6 | **124.8±2.0** |
| | medium-expert | 106.1→78.0 | 109.7→110.8 | 106.0→110.2 | 110.6→118.0 | 110.0 | 109.1 | **126.7±1.0** |
| | expert | 102.1→90.8 | 93.2→102.6 | 109.0→110.1 | 110.0→120.5 | 111.2 | 110.5 | **128.9±2.7** |
| Halfcheetah | random | 15.8→39.8 | 29.6→62.9 | 19.9→12.2 | 9.8→42.4 | 10.4 | **90.6** | 87.4±1.5 |
| | medium | 48.7→55.3 | 71.3→84.4 | 47.6→52.2 | 48.2→84.8 | 47.5 | 89.7 | **92.5±1.2** |
| | medium-replay | 44.5→52.1 | 66.1→76.2 | 45.8→47.9 | 44.7→78.5 | 45.5 | 74.0 | **84.7±1.9** |
| | medium-expert | 91.4→90.3 | 90.6→85.6 | 51.3→91.2 | 93.8→89.8 | 92.6 | 95.5 | **99.5±1.8** |
| | expert | 91.9→73.4 | 65.3→100.8 | 85.8→93.3 | 97.4→89.3 | 96.4 | 93.5 | **101.1±3.8** |
| $\delta_{\text{sum}}(0.1\text{M})$ | | 68.7 | 268.7 | 80.5 | 349.4 | | 355.4 | **550.4** |

**Theorem 4.5** *With Assumption 4.1 and 4.2, when $b > 1$, the variance of post-update Q-value $Q_{\theta_{new}}$ with $D_b(x)$ function is smaller than that with $l_2$ loss function.*

We show the detailed proof in Appendix B.4 and B.5. The theorems above confirm that using $D_b(x)$ to update the Q-value, instead of the $l_2$ function, reduces both the overestimated bias and estimation variance at each update step. This leads to a smaller accumulated bias and more robust Q-value estimation, theoretically reinforcing the effectiveness of LAROO.

## 5 EXPERIMENTS

We present experimental results to evaluate the effectiveness of LAROO. In Section 5.2, we demonstrate how LAROO significantly outperforms existing state-of-the-art O2O methods across various tasks. Finally, we conduct ablation studies to further evaluate the effectiveness of LAROO.

### 5.1 EXPERIMENT SETTING

***Benchmark***. We conduct our experiments on the D4RL benchmark (Fu et al., 2020), which provides various continuous control tasks and training datasets. We choose MuJoCo and the more challenging sparse-reward environment, AntMaze, as our testing tasks. We compare LAROO with several state-of-the-art baselines in O2O RL, including PEX (Zhang et al., 2023), Cal-QL (Nakamoto et al., 2023), SO2 (Zhang et al., 2024), ENOTO (Zhao et al., 2024), BOORL (Hu et al., 2024). We also compare with the RLPD (Ball et al., 2023), which effectively leverages offline datasets to train agents from scratch in online environments. We rerun these baselines with their official implementations. We provide LAROO's codes in https://github.com/USTC-AI4EEE/LAROO.

***Experimental implement details***. In our experiments, we first offline pretrain agents for one million gradient steps and then online fine-tune agents for 100K steps, this training setup is also adopted by (Feng et al., 2024). For Mujoco tasks, LAROO adopts TD3BC (Fujimoto & Gu, 2021) and TD3 (Fujimoto et al., 2018) with ensemble Q-functions as the backbone algorithms for the offline and online phases, respectively. Since TD3BC cannot handle with sparse-reward tasks, LAROO uses LAPO (Chen et al., 2022) as the backbone for Antmaze tasks. These choices of backbone algorithms follow those in the ENOTO baseline (Zhao et al., 2024). Please refer to Appendix C for more details and parameter settings.

### 5.2 MAIN RESULTS

***The performance of LAROO***. We evaluate both the final fine-tuning performance and the performance improvement $\delta_{sum}$ during online fine-tuning. Each experimental result is averaged over five

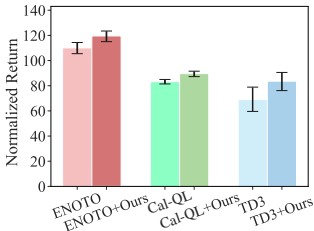 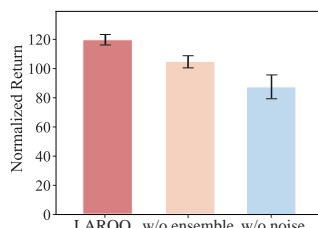 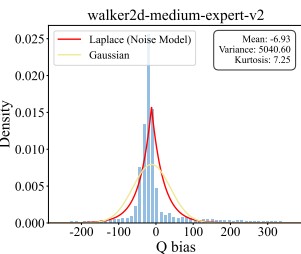

Figure 3: (a) Normalized returns of existing O2O RL methods combining with LAROO on the walker2d tasks. (b) Ablation for components in LAROO: the performance without ensemble models or without noise model respectively in the walker2d-medium-replay task. (c) Laplace-based noise model: we plot the Laplace-based noise against the empirical Q bias.

seeds. As shown in Table 1, LAROO achieves superior final performance and notable performance improvement compared with other O2O RL methods, with an average score improvement of +54.8% over the second-best method BOORL (Hu et al., 2024). In Figure 12, we present the online training curves of different methods in Mujoco and Antmaze tasks. It is evident that LAROO consistently exhibits significant sample efficiency and training stability across various tasks.

***Compatibility of LAROO with other O2O methods***. To demonstrate the compatibility of LAROO with previous O2O RL methods for further performance improvement, we replace the $l_2$ loss function in previous methods with $D_b(x)$ for Q-learning and evaluate the performance improvement. We select TD3, Cal-QL (Nakamoto et al., 2023) and ENOTO (Zhao et al., 2024) for evaluation, as these three methods exhibit varying levels of performance in O2O RL. As shown in Figure 3 (a) and Table 10, the experimental results validate that our proposed robust loss function is highly effective and can serve as a plug-in method to further enhance the performance of existing methods.

***Training stability of LAROO***. We evaluate training stability in O2O RL based on two metrics: (1) the variance, (2) the Normalized Cumulative Performance Drop (NCD), which quantifies the degree of performance degradation during training (Feng et al., 2024). Lower values of variance and NCD indicate better training stability. The results are shown in Table 7. Specially, LAROO generally achieves a superior balance between performance improvement and training stability.

***Empirical validation for Laplace-distributed noise model***. We plot the Laplace-distributed noise against the empirical Q bias, as shown in Figure 3(c) and Figure 10. We note that the Laplace assumption better fits the empirical data than the Gaussian. We further discuss the advantages and limitations of using Laplace distribution to capture heavy-tailed nature of Q bias in Appendix B.1.

## 5.3 ABLATION STUDIES

***Ablation for components of LAROO***. We present distributions of Q bias without ensemble models or without the noise model, respectively, in Figure 15. Ensemble models effectively reduce overestimated Q bias, shifting its mean toward zero, but still suffers from high heavy-tailedness. The noise model alleviates heavy-tailedness of Q biases but remains prone to overestimation. By combining these two components, LAROO promotes the heavy-tailed Q bias into a more standardized form.

We further compare the contribution of ensemble models with Laplace noise model in LAROO. We assess their contributions by plotting training curves and computing related metrics. As shown in Figure 16 and Table 12, the results indicate that the Laplace noise model plays a more critical role in LAROO. Please refer to Appendix C.4 for detailed ablation studies.

***Ablation for UTD ratio and ensemble size***. We conduct ablation experiments for the update frequency of data (UTD) and the ensemble size. A higher UTD has been widely used for improving sample efficiency, while the ensemble of Q function can help mitigate Q bias and improve training stability in O2O RL (Zhao et al., 2024). For a fair comparison, we set the UTD $= 1$ and the ensemble size $N = 1$ in LAROO, then compare it with Cal-QL and PEX. Other baselines, which also leverage high UTD or ensemble models (i.e., BOORL, ENOTO and SO2), are excluded from the comparison. As shown in Table 10, LAROO outperforms Cal-QL in 10 out of 13 experiments, demonstrating its effectiveness even without the benefits of high UTD or ensemble models.

*Ablation for the Q-value loss function*. We evaluate the performance with and without the proposed Q-value loss function $\mathcal{L}_b(x)$. Additionally, we compare $\mathcal{L}_b(x)$ with other commonly used robust loss functions, such as the Huber loss and the Cauchy loss functions, which prior studies have shown to effectively handle outliers in training data (Zahra et al., 2014). The results in Table 9 demonstrate that $\mathcal{L}_b(x)$ outperforms these alternatives for better capturing these heavy-tailed Q biases.

## 6 RELATED WORK

**Offline-to-online RL**. Offline-to-Online RL primarily faces the challenge of a significant distribution shift between offline and online data during the fine-tuning process. This shift leads to inaccurate Q-value estimation—typically overestimated—for out-of-distribution online data, which results in performance degradation during fine-tuning (Lee et al., 2021b; Zhang et al., 2024; Zhou et al., 2025). Various strategies have been proposed to improve training stability. Specifically, some methods retain constraints from offline stage or introduce additional conservative techniques to ensure training stability. For example, Off2OnRL (Lee et al., 2021b) leverages a balanced replay buffer and utilizes pessimistic Q-ensemble networks. Cal-QL (Nakamoto et al., 2023) employs calibrated Q-learning to learn a conservative Q-value function. However, Feng et al. (2024) argue that these constrained methods improve training stability at the cost of sufficient performance improvement. FineTuneRL (Wagenmaker & Pacchiano, 2023) proposes the actor-critic alignment step to bridge the gap between online and offline Q-value learning, mitigating performance drop. Some studies (Feng et al., 2024; Zhang et al., 2024) increase the update frequency of Q-value networks to enhance Q-value estimation. ENOTO (Zhao et al., 2024) uses Q-ensemble models to reduce the Q-value estimation bias. BOORL (Hu et al., 2024) introduces a Bayesian approach to guide agent's online fine-tuning.

**Distributional modeling of Q-values**. Distributional reinforcement learning aims to model and predict the full distribution of returns, for more robust and risk-sensitive decision-making (Bellemare et al., 2017; 2023). Duan et al. (2021) propose the distributional soft actor-critic (DSAC) algorithm, which learns a Gaussian distribution function of state-action returns (i.e., Q-values) to effectively mitigate Q-value overestimation. However, these methods may face incompatibility issues in O2O RL with offline pretrained models, which usually estimate Q-values in expectation rather than the full distribution (Fujimoto et al., 2018; Kumar et al., 2020). In this paper, we model the heavy-tailed Q bias with Laplace distribution instead of directly modeling the Q-values, *thereby capturing the heavy-tailed characteristics of Q bias while maintaining expected estimates of Q-values*.

**Extreme Q-learning**. Garg et al. (2023) apply the Gumbel distribution to estimate the maximum Q-value. This work builds on the extreme value theorem which states that the maximal values drawn from any exponential-tailed distribution follow a Gumbel distribution (Fisher & Tippett, 1928). However, the training objective of extreme Q-learning differs from that of other offline algorithms, which could disrupt well-pretrained Q-value networks during online fine-tuning.

## 7 CONCLUSION

In this paper, we focus on the Q-value estimation in online data for O2O RL. Through extensive experiments, we reveal for the first time that the Q bias follows a heavy-tailed distribution in O2O RL, a phenomenon unexplored in previous research. These heavy-tailed Q biases lead to instability and inefficiencies in performance improvement. To mitigate their influence, we propose LAROO for robust Q-value estimation. LAROO captures the heavy-tailed property with Laplace-distributed noise models, and mitigates the overestimation with conservative ensemble-based estimates. LAROO alleviates the heavy-tailedness of Q bias and promotes it into a standardized form, improving training stability and performance. Future work could explore more robust techniques to mitigate the heavy-tailed Q bias. We look forward to the continued development of robust O2O RL methods that improve online fine-tuning for pretrained agents. These advancements could significantly facilitate the deployment of well pretrained agents in various real world scenarios.

## ACKNOWLEDGMENTS

The authors would like to thank all the anonymous reviewers for their valuable suggestions. This research is supported by Anhui Provincial Natural Science Foundation (Grant No.2408085QF214), the Fundamental Research Funds for the Central Universities (Grant No.WK2080000206), and the Smart-Grid National Science and Technology Major Project (Grant No. 2025ZD0805500).

## ETHICS STATEMENT

Offline-to-online reinforcement learning methods hold significant promise for facilitating the deployment of well-pretrained agents in real-world scenarios. Our study shows that existing approaches continue to suffer from training instability and inefficiencies due to heavy-tailed Q-value bias. We propose strategies that effectively mitigate the influence of this bias, leading to more robust Q-value estimation. We believe our research does not contain potentially harmful insights and any discrimination issues.

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

# APPENDIX

# A   THE Q BIAS IN OFFLINE-TO-ONLINE RL

## A.1   THE HEAVY-TAILED DISTRIBUTION

There exist several mathematical definitions of heavy-tailed distribution. These definitions characterize "heavy-tailed" respectively based on (i) the convergence rate of the tail (Embrechts et al., 2013) and (ii) the moment properties of the distribution function (El Adlouni et al., 2008; Werner & Upper, 2002). We adopt the following definitions:

**Definition A.1** *(Embrechts et al., 2013) A non-negative random variable w is called heavy-tailed if its tail probability $F_w(t) := P(w \geq t)$ is asymptotically equivalent to $t^{-\alpha^*}$ as $t \to \infty$ for some positive number $\alpha^*$. Here, $\alpha^*$ determines the heavy-tailedness and $\alpha^*$ is called tail index of w. A lower tail index indicates heavier tails.*

In practice, a widely used moment-based criterion defines heavy tails as follows.

**Definition A.2** *(El Adlouni et al., 2008; Werner & Upper, 2002) If $X$ is a random variable, $\mu_X$ and $\sigma_X$ are the mean and the standard deviation of $X$, then $X$ is called heavy-tailed if*

$$E\left[\frac{(X - \mu_X)^4}{\sigma_X^4}\right] > 3$$

This criterion (kurtosis > 3) serves as a well-established measure for heavy-tailedness. In this study, we explore two estimators as heuristic measures to understand heavy tails and non-Gaussianity of Q bias.

**The Kurtosis**. We will compute the Kurtosis values of Q bias distribution as Definition A.2.

**The log-log plot**. A log-log plot directly visualizes the distribution's tail-index for diagnosing the heavy-tailed behavior. The utility of this plot stems from its ability to transform a power-law relationship, defined by a probability density function $p(x) \propto x^{-\alpha}$, into a linear form: $\log(p(x)) = -\alpha \log(x) + C$. Consequently, the slope of the linear segment in the plot's tail region serves as a direct graphical proxy for the negative tail-index $-\alpha$, as discussed in Definition A.1

## A.2   THE HEAVY-TAILED Q BIAS

In this section, we present the heavy-tailed distribution of Q bias in online samples when fine-tuning pretrained agents with different algorithms on Mujoco tasks. We calculate the kurtosis value to quantify the heavy-tailedness of Q bias. It should be noted that the Q bias is defined as $\mathbb{E}[Q_\theta(s,a)] - \mathcal{Q}(s,a)$ in our paper.

We first present the Q bias distribution when fine-tuning with offline algorithms. Specifically, we employ the same algorithm in offline pre-training and online fine-tuning stage. The results show that the distribution of Q bias has a longer tail than the normal distribution, with notably high Q biases in the tail.

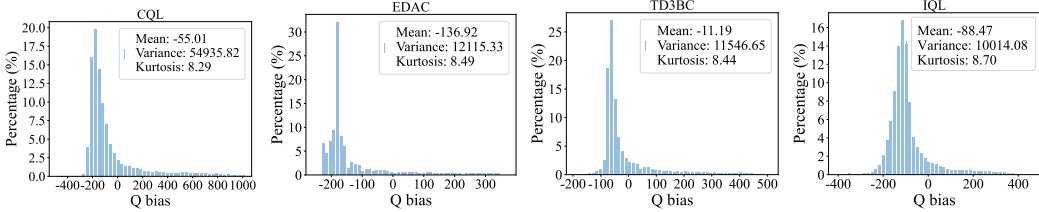

Figure 4: The heavy-tailed Q bias distribution in offline RL methods. We present Q bias distributions of four offline RL methods (CQL (Kumar et al., 2020), EDAC (An et al., 2021), TD3BC (Fujimoto & Gu, 2021), IQL (Kostrikov et al., 2022)) for online fine-tuning at the 100k online step in the Walker2d-medium task.

Next, we illustrate the distribution of Q bias when fine-tuning with different O2O RL algorithms. We pretrain agents using the corresponding offline backbone algorithms and further fine-tune them online with O2O RL algorithms. Recent work such as ENOTO (Zhao et al., 2024), which applies a Q-ensemble strategy to reduce bias, and SO2 (Zhang et al., 2024), which increases the update frequency to improve Q-value estimation, still exhibit Q bias distributions with heavy-tailed characteristics. In Figure 1 and Figure 5, we illustrate the Q bias distribution at 50k, 100k and 150k steps in the Walker2d task, respectively. In Figure 6, we test the distribution of Q bias in more tasks, including hopper and halfcheetah.

For each distribution of Q bias, we also calculate its mean and variance. A positive mean indicates the Q-value overestimation while a negative mean implies Q-value underestimation, and low variance benefits stable performance improvement (Chen et al., 2021). We observe that the Q-values tend to be underestimated with offline RL methods and overestimated with several O2O RL methods during fine-tuning process. Furthermore, they struggle to decrease its high variance, indicating ongoing instability during training.

Based on extensive experiments above, for the first time, we empirically reveal that the Q bias follows a heavy-tailed distribution in online data during online fine-tuning. This finding inspires us to identify the influence of heavy-tailed Q bias in O2O RL, and further design our methods to improve fine-tuning performance and stability.

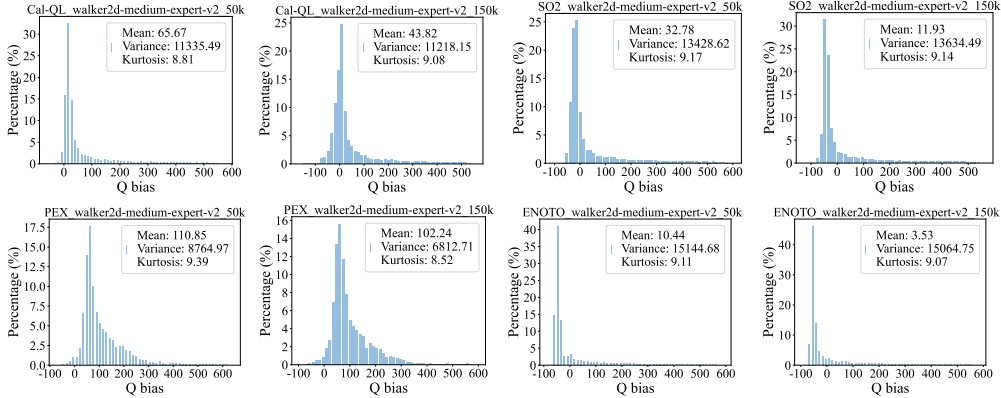

Figure 5: The heavy-tailed distribution of Q bias on wilder choices of online steps with four O2O RL methods (SO2 (Zhang et al., 2024), Cal-QL (Nakamoto et al., 2023), ENOTO (Zhao et al., 2024), PEX (Zhang et al., 2023)).

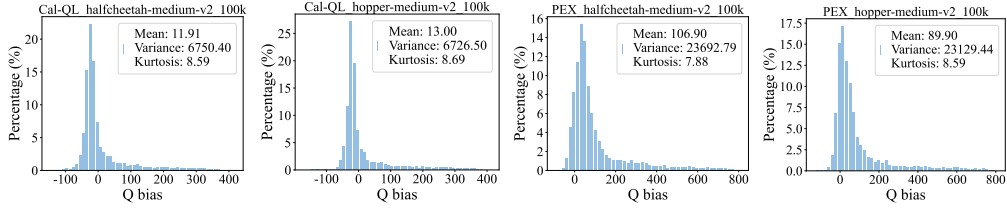

Figure 6: The heavy-tailed distribution of Q bias on wilder tasks.

**We illustrate the Log-log plot of Q bias to further validate its heavy-tailedness**. As shown in Figure 7, the Q biases in the tail are roughly arranged in a straight line, indicating that the Q bias may follow a heavy-tailed distribution. We also compute their tail index.

### A.3 THE REASONS CAUSING HEAVY-TAILED Q BIAS

**We discuss the underlying reasons causing the heavy-tailed Q bias in this section**. In offline-to-online reinforcement learning, a heavy-tailed distribution of Q-value estimation bias (Q bias) is primarily caused by the distributional shift between offline and online data. The Q-function, pre-trained on a static offline dataset, must extrapolate values for out-of-distribution (OOD) samples encountered during online interaction. The heavy-tailed nature of the bias stems from twe rea-

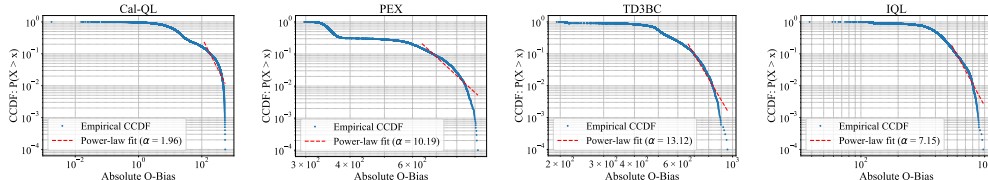

Figure 7: **The Log-log plot of Q bias**. The blue markers represent the cumulative distribution function (CCDF) for the absolute Q bias. The red dashed line shows a power-law fit applied to the tail of the distribution (the largest 15% of values). These experiments are conduct in the walker2d-medium-expert tasks.

sons: (1) **the non-uniformity of this shift**; online samples highly dissimilar to the offline data induce large extrapolation errors, which manifest as the extreme values in the distribution's tails. Conversely, samples with smaller shifts result in minor biases, forming the central mass of the distribution. (2) **the Q-function with Max operator**; In Q-learning, Q-value targets are bootstrapped and include a maximization step, $y = r + \max_{a'} Q_{\hat{\theta}}(s', a')$, the max operator induces a systematic selection of overestimated actions (the "winner's curse"). Rare samples with large positive noise are preferentially chosen and then propagated through bootstrapping, which amplifies extremal overestimation (Fujimoto et al., 2018; D'Eramo et al., 2021a). Thus, (1) and (2) cause the overestimated Q bias yielding a heavy right tail in its distribution.

Guided by the analysis above, our methods mainly focus on the latter cause of heavy-tailed Q bias, since the non-uniformity of data shift is hard to control during online exploration. Our method captures the heavy-tailed Q biases with parameterized noise models, and provides a theoretical guarantee of reduced post-update bias even with Max operator, as shown in Method 4.

**We conduct experiments to investigate whether the heavy-tailed Q bias partly arises from the non-uniformity of distribution shift**. Specifically, we first measure the distance between the online data and the offline dataset, and then compute Spearman's rank correlation coefficient $\rho$ to quantify the relationship between the Q bias and the measured distance.

To assess the distance between the online data and the offline dataset, we employ two complementary metrics: (1) the k-nearest neighbor distance $D_{\text{kNN}}$; (2) the classifier-based density ratio $D_{\text{DR}}$. The $D_{\text{kNN}}$ calculates the average Euclidean distance to the $k$ nearest offline samples in the state-action space. This metric captures the geometric similarity between the offline and online data.

On the other hand, the classifier-based density ratio $D_{\text{DR}}$ is widely adopted to quantify the discrepancy between the two distributions (i.e., online and offline data) (Sugiyama et al., 2012; Yamada et al., 2013). Let $z = (s, a)$, $p_{\text{off}}(z)$ and $p_{\text{on}}(z)$ denote the offline and online state-action distributions. We train a discriminator $h_\phi(z) \in (0, 1)$ to distinguish online (label 1) from offline (label 0) by minimizing the logistic loss on a class-balanced sample. The Bayes-optimal discriminator satisfies

$$h^*(z) = \frac{p_{\text{on}}(z)}{p_{\text{on}}(z) + p_{\text{off}}(z)} \quad \Rightarrow \quad g^*(z) = \log \frac{h^*(z)}{1 - h^*(z)} = \log \frac{p_{\text{on}}(z)}{p_{\text{off}}(z)},$$

so the logit $g_\phi(z)$ of a well-trained classifier is a consistent estimator of the log density ratio $\log\left(\frac{p_{\text{on}}(z)}{p_{\text{off}}(z)}\right)$. This makes $g_\phi(z)$ a principled, scalar per-sample distance: larger $D_{DR}$ indicates that online data lies farther away from offline support.

In summary, $D_{\text{kNN}}$ provides a simple, geometric distance, while $D_{\text{DR}}$ offers a distribution-aware measure that reflects how likely a sample is to belong to the online versus offline distribution. Taken together, these metrics give a comprehensive characterization of how online samples shift relative to the offline distribution.

**Finally, we validate that the non-uniformity of distribution shift could contribute to the heavy-tailed Q bias**. We present scatter plots illustrating the relationship between Q-bias and two distance metrics. To quantify this relationship, we compute Spearman's rank correlation coefficient, which measures the strength of the monotonic association between Q-bias and the distance. As shown in

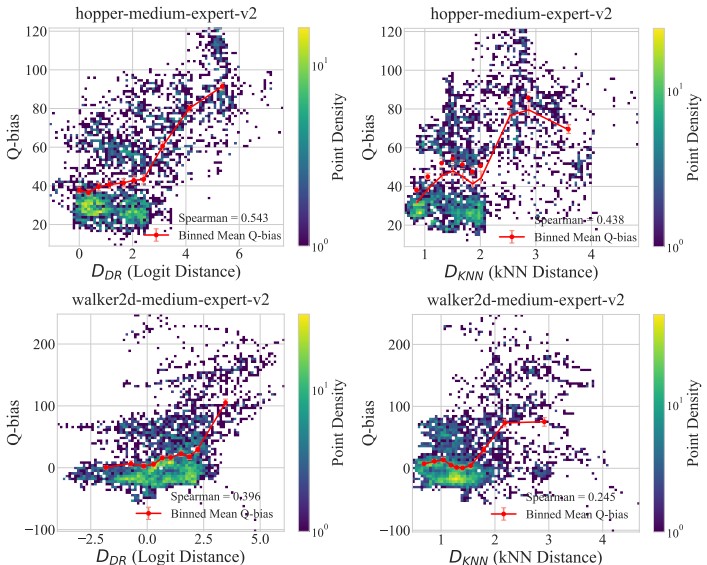

Figure 8: Scatter plots of Q bias and the two distance metrics $D_{knn}, D_{DR}$. We compute the binned mean of Q bias with 95% confidence interval, and spearman correlation coefficient.

Figure 8, there is a clear positive correlation: the Q-bias for state-action pairs tends to increase as their distance from the offline data distribution grows.

### A.4 THE INFLUENCE OF HEAVY-TAILED Q BIAS

**In this subsection, we validate the influence of the heavy-tailed Q bias in Q-value estimation**. In robust regression research (Song et al., 2014; Huang et al., 2023; Xu et al., 2025b), they have identified that the heavy-tailed errors can lead to instability in the coefficient estimation, particularly when the errors contain a significant number of extreme values.

We conduct a simple validation experiment to verify the oscillation in Q-value estimation induced by heavy-tailed Q bias. For approximating the heavy-tailed Q bias, we introduce heavy-tailed noise into the estimated Q-values during the update of the Q-value network. Specifically, we train two agents online based on TD3 (Fujimoto et al., 2018) with ensemble Q-functions. We introduce heavy-tailed Cauchy noise with a mean of 0 and a standard deviation of 10 in one agent, and introduce Gaussian noise with the same mean and standard deviation in the other for a fair comparison.

As shown in Figure 9, the results demonstrate that, under heavy-tailed Cauchy noise, the agent exhibits higher and continuously increasing Q bias compared to the agent trained with Gaussian noise. This suggests greater inaccuracy and oscillation in Q-value estimation. Furthermore, the final performance is much lower, indicating an inefficient performance improvement due to heavy-tailed Q biases.

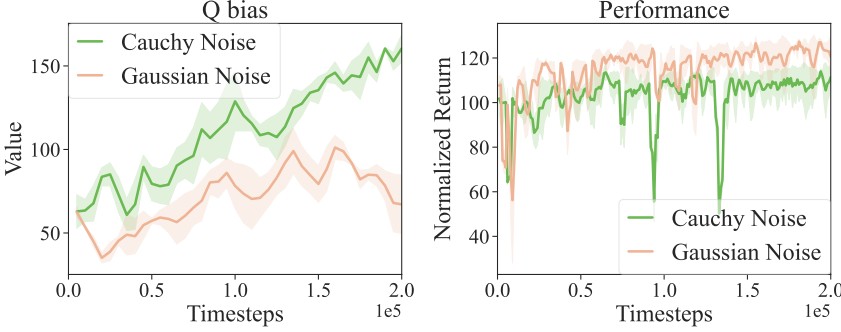

Figure 9: Validation of the instability caused by heavy-tailed Q biases. This experiment is conducted in the Walker2d-medium-expert task.

## B   THE SUPPLEMENTARY FOR METHOD

### B.1   LAPLACE MODELING FOR Q BIAS

**In this section, we analyze the feasibility and rationality of using Laplace distribution to capture the heavy-tailed nature of Q bias in O2O RL**. In robust regression research (Song et al., 2014; Huang et al., 2023; Xu et al., 2026; 2025a), the Laplace distribution offers a theoretical foundation for modeling data with many outliers and long tails in robust regression. Specifically, under Laplace errors, the maximum likelihood estimator (MLE) is equivalent to the robust least absolute deviation (LAD) estimator (Bai, 1995), which effectively reduces the influence of non-normal errors by constraining update gradients.

In this paper, we introduce a parameterized Laplace-distributed noise to capture the heavy-tailed Q bias. First, we plot the Laplace distribution against the empirical Q bias in Fig 10. Though the Laplace distribution are significantly better than Gaussian distribution in fitting Q bias, the limitations are clear. The Laplace is not a heavy-tailed distribution and symmetric, leaving a gap to fit Q bias. In practice, it is challenging to identify a distribution that perfectly models Q bias across different baselines. For instance, the distribution of Q bias in ENOTO (Zhao et al., 2024) is highly asymmetric with a heavy right tail as shown in Figure 1, making it difficult to fit.

In this paper, taken into account both the theoretical foundation and practical feasibility, the Laplace distribution for modeling Q bias have the following advantages:

- Laplace distribution can model data with heavier tails and outliers compared to the Gaussian distribution, and it has solid theoretical foundation in robust regression analysis.

- The Kullback-Leibler divergence between Laplace distributions (i.e., Q bias distributions) facilitates the derivation of Equation 6 for Q-value updates. However, for most other complex distributions (e.g., generalized Gaussian distribution), obtaining an analytical solution for the KL divergence is infeasible.

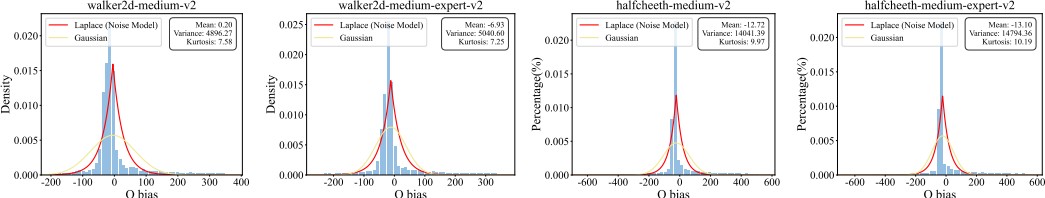

Figure 10: **The Laplace-based noise model of LAROO**. We plot the Laplace distribution against the empirical Q bias. The Laplace assumption better fits the Q bias than the Gaussian.

### B.2   DERIVATION OF THE LOSS FUNCTION

**We provide the derivation of Equation (5)**. First, the Kullback-Leibler (KL) divergence between a probability distribution $q(x)$ and a reference distribution $p(x)$ is

$$D_{KL}(p(x)\|q(x)) = H(p(x), q(x)) - H(p(x))$$
$$= -\int_{-\infty}^{\infty} p(x)\log q(x)dx + \int_{-\infty}^{\infty} p(x)\log p(x)dx \tag{11}$$

where $H(p(x), q(x))$ is the cross entropy between $p(x)$ and $q(x)$ and $H(p(x))$ is the entropy of $p(x)$. If $p(x)$ follows a Laplace distribution, i.e., $p(x) \sim \text{Laplace}(\mu, b)$, where $\mu$ is the mean parameter and $b$ is the scale parameter, $\mu \in \mathbb{R}$, $b \in \mathbb{R}^+$, thus the distribution $p(x)$ becomes:

$$p(x) = \frac{1}{2b}\exp\left(-\frac{|x-\mu|}{b}\right) \tag{12}$$

Next, we derive the KL divergence between two Laplace distributions according to the following Lemma.

**Lemma B.1** *Both $p(x)$ and $q(x)$ are the Laplace distributions, i.e., $p(x) \sim \text{Laplace}(\tilde{\mu}_1, \tilde{b}_1)$, $q(x) \sim \text{Laplace}(\tilde{\mu}_2, \tilde{b}_2)$, the KL divergence between $p(x)$ and $q(x)$ is*

$$D_{KL}(p(x), q(x)) = \frac{\tilde{b}_1 \exp\left(-\frac{|\tilde{\mu}_1 - \tilde{\mu}_2|}{\tilde{b}_1}\right) + |\tilde{\mu}_1 - \tilde{\mu}_2|}{\tilde{b}_2} + \log(\frac{\tilde{b}_2}{\tilde{b}_1}) - 1 \tag{13}$$

Specifically, in our case, we set $x = \mathcal{Q}(s_i, a_i)$, $\tilde{\mu}_1 = \mathcal{T}Q_\theta(s_i, a_i) + \mu$ and $\tilde{\mu}_2 = Q_\theta(s_i, a_i) + \mu$. Thus, we can derive Equation (5).

**We clarify the Equation 5 with Bayesian principles for easy understanding.** When using non-informative priors, the posterior distribution in Bayesian inference is numerically equivalent to the likelihood function in frequentist statistics (Gelman et al., 1995). In ours, the prior $p(\mathcal{Q}(s, a))$ can be non-informative because the true Q-value is always unknown. Therefore, the likelihood of true Q-value $p(\mathcal{Q}(s, a) \mid Q_\theta(s, a))$ is also the posterior given the bellman target. The $p(\mathcal{Q}(s, a) \mid Q_\theta(s, a))$ is the posterior given the estimated Q-values. Thus, minimizing the KL divergence encourages the posterior $p(\mathcal{Q}(s, a) \mid Q_\theta(s, a))$ to approximate the posterior $p(\mathcal{Q}(s, a) \mid \mathcal{T}Q_\theta(s, a))$.

Since the posterior $p(\mathcal{Q}(s, a) \mid \mathcal{T}Q_\theta(s, a))$ incorporates true environmental return information (i.e., future rewards), it serves as a closer approximation to the Bayes-optimal posterior (i.e., $p(\mathcal{Q} \mid \text{all environmental evidence})$), compared with the posterior $p(\mathcal{Q}(s, a) \mid Q_\theta(s, a))$.

**We provide a brief analysis of the advantage of the Equation 5 than the robust $l_1$ loss function.** The likelihood $p(\mathcal{Q}(s, a) \mid \mathcal{T}Q_\theta(s, a))$ explicitly captures the distributional information of heavy-tailed bias in bellman target with Laplace modelings for Q updates. Instead, if the heavy-tailed Q bias in bellman target is ignored, i.e., the $p(\mathcal{Q}(s, a) \mid \mathcal{T}Q_\theta(s, a))$ is a Dirac delta function centered on $\mathcal{T}Q_\theta(s, a)$. The KL divergence becomes:

$$-\int_{-\infty}^{\infty} p(\mathcal{Q}(s, a) \mid \mathcal{T}Q_\theta(s, a)) \log q(\mathcal{Q}(s, a) | Q_\theta(s, a)) d\mathcal{Q}(s, a) = -\log q(\mathcal{T}Q_{\hat{\theta}}(s, a) | Q_\theta(s, a)) \tag{14}$$

Thus, LAROO reduces to the negative log likelihood, which is exactly equivalent to applying an $L1$ loss under a Laplace-noise assumption for TD-errors. Therefore, using $l_1$ or Huber loss corresponds to applying a fixed robust regression objective to TD errors, which does not explicitly model the heavy-tailed Q bias in bellman target and may discard distributional information of Q bias.

In summary, under heavy-tailed Q bias, it is a natural and reasonable choice to model the Q-bias for both estimated Q-values and bellman targets, and derive the Equation 5 for updates. In contrast, directly using $l_1$ loss for updates may lose the distributional information about heavy-tailed bias.

### B.3 PROOF OF DIFFERENTIABILITY

In this section, we will derive the first and second derivatives of the loss function $D_b(x)$ and prove their existence at $x = 0$. We follow the proof process in prior work (Meyer, 2021). The first derivative of $D_b(x)$ is shown as Equation (15), $sgn(x)$ is the sign function. We can get that the gradient of $D_b(x)$ is bounded within the range $[-\frac{1}{b}, \frac{1}{b}]$.

$$D_b'(x) = \begin{cases} \frac{1 - \exp\left(-\frac{x}{b}\right)}{b} & x \geq 0 \\ -\frac{1 - \exp\left(\frac{x}{b}\right)}{b} & x < 0 \end{cases} \tag{15}$$
$$= \frac{\text{sgn}(x)}{b}\left(1 - \exp\left(-\frac{|x|}{b}\right)\right)$$

And its second derivative is:

$$D_b''(x) = \begin{cases} \frac{\exp\left(-\frac{x}{b}\right)}{b^2} & x \geq 0 \\ \frac{\exp\left(\frac{x}{b}\right)}{b^2} & x < 0 \end{cases} = \frac{1}{b^2} \exp\left(-\frac{|x|}{b}\right) \tag{16}$$

Next, to prove the derivatives of $D_b$ and $D_b'(x)$ at $x = 0$, we should show that both $D_b$ and $D_b'(x)$ are differentiable at $x = 0$. The function $f(x)$ is considered differentiable at $x = a$ if the limit

exists. The derivative of any function $f(x)$ at $x = a$ is defined as follows, and the function $f(x)$ is considered differentiable at $x = a$ if the limit exists.

$$f'(a) = \lim_{x \to a} \frac{f(x) - f(a)}{x - a} \tag{17}$$

For the function $D_b(x)$, the derivative of $D_b(x)$ at $x = 0$ is defined as:

$$\begin{aligned} D_b'(0) &= \lim_{x \to 0} \frac{b \exp\left(-\frac{|x|}{b}\right) + |x| - b}{bx} \\ &= \lim_{x \to 0} \frac{b\left(\exp\left(-\frac{|x|}{b}\right) - 1\right)}{bx} + \lim_{x \to 0} \frac{|x|}{bx}. \end{aligned} \tag{18}$$

The right limit of $D_b'(0)$ is:

$$\lim_{x \to 0^+} D_b'(0) = \lim_{x \to 0^+} \left(-\frac{1}{b} + \frac{1}{b} + o(x)\right) = 0 \tag{19}$$

The left limit of $D_b'(0)$ is:

$$\lim_{x \to 0^-} D_b'(0) = \lim_{x \to 0^-} \left(\frac{1}{b} - \frac{1}{b} + o(x)\right) = 0 \tag{20}$$

Both the left limit and right limit of $D_b'(0)$ become zero, which means the limit exists and proves $D_b(x)$ is differentiable at $x = 0$.

For the function $D_b'(x)$, the derivative of $D_b'$ at $x = 0$ is defined as:

$$D_b''(0) = \lim_{x \to 0} \frac{\operatorname{sgn}(x)\left(1 - \exp\left(-\frac{|x|}{b}\right)\right)}{bx} \tag{21}$$

The right limit is the following:

$$\lim_{x \to 0^+} \frac{1 - \exp(-\frac{x}{b})}{bx} = \lim_{x \to 0^-} \left(\frac{1}{b^2} + o(x)\right) = \frac{1}{b^2} \tag{22}$$

Similarly, the left limit is the following:

$$\lim_{x \to 0^-} -\frac{1 - \exp(\frac{x}{b})}{bx} = \lim_{x \to 0^-} \left(\frac{1}{b^2} + o(x)\right) = \frac{1}{b^2} \tag{23}$$

Therefore, both the left and the right limits are $\frac{1}{b^2}$, which implies the limit exists and $D_b'(x)$ is differentiable at $x = 0$.

Finally, we illustrate the function $D_b(x)$ and its first derivative $D_b'(x)$ as follows.

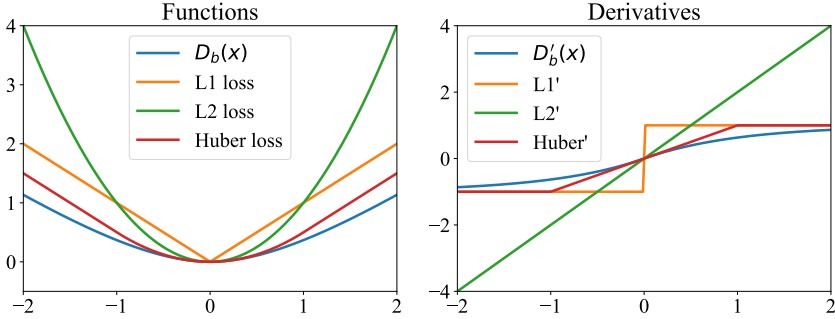

Figure 11: The curves of $D_b(x)$ and its first derivative $D_b'(x)$ when $b = 1$

.

### B.4 PROOF OF THEOREM

***We first provide the prerequisite knowledge for the theorem***. Due to page limitations in the main text, we repeat and present a more detailed definition here as a supplement, following the analysis in DSAC (Duan et al., 2021), which demonstrates the effectiveness of distributional modeling of Q-values by analyzing the reduced estimate bias per training step.

We denote the greedy target $R(s,a) + \gamma \mathbb{E}_{s' \sim T(\cdot|s,a)}\left[\max_{a'} Q_{\hat{\theta}}(s', a')\right]$ as $y$. Conventionally, the Q-value network $Q_\theta$ is updated by minimizing the mean square loss $(y - Q_\theta(s,a))^2/2$ using gradient descent methods, i.e.,

$$\theta_{new} = \theta + \beta(y - Q_\theta(s,a))\nabla_\theta Q_\theta(s,a) \tag{24}$$

where $\beta$ is the learning rate and is sufficiently small. Let $\theta_{id}$ represents the ideal post-update parameter obtained based on true target $\tilde{y} = R(s,a) + \gamma \mathbb{E}_{s'}\left[\max_{a'} \mathcal{Q}(s', a')\right]$, that is,

$$\theta_{id} = \theta + \beta(\tilde{y} - Q_\theta(s,a))\nabla_\theta Q_\theta(s,a) \tag{25}$$

Since $\beta$ is small enough, the post-update Q-value $Q_{\theta_{new}}(s,a)$ can be approximated by linearizing around $\theta$ using Taylor's expansion. Similarly, we can get the ideal post-update Q-value $Q_{\theta_{id}}(s,a)$, **which represents the most ideal and correct Q value that can be achieved currently in one update**. We can approximate:

$$\begin{aligned} Q_{\theta_{id}}(s,a) &\approx Q_\theta(s,a) + \beta(\tilde{y} - Q_\theta(s,a))\|\nabla_\theta Q_\theta(s,a)\|_2^2 \\ Q_{\theta_{new}}(s,a) &\approx Q_\theta(s,a) + \beta(y - Q_\theta(s,a))\|\nabla_\theta Q_\theta(s,a)\|_2^2 \end{aligned} \tag{26}$$

Then, in expectation, we can define the single-step estimation bias of the $l_2$ udpate, denoted as:

$$\Delta_{l_2}(s,a) = \mathbb{E}_{\varepsilon_{\hat{\theta}}}\left[Q_{\theta_{new}}(s,a) - Q_{\theta_{id}}(s,a)\right] \approx \beta(\mathbb{E}_{\varepsilon_{\hat{\theta}}}[y] - \tilde{y})\|\nabla_\theta Q_\theta(s,a)\|_2^2$$

where $\varepsilon_{\hat{\theta}}$ is the Q-value bias in target $y$. $\Delta_{l_2}(s,a)$ **represents the estimation bias introduced by the inaccurate target $y$ in one step with $l_2$ loss function**. Moreover, previous research (Duan et al., 2021; Hasselt, 2010) has verified that the Q-value is overestimated due to the max operator and it is clear that:

$$\mathbb{E}_{\varepsilon_{\hat{\theta}}}[y] - \tilde{y} \geq 0, \quad \Delta_{l_2}(s,a) \geq 0 \tag{27}$$

***Proof of Theorem 4.4***. Building on the above analyses, we now repeat the Theorem 4.4 and demonstrate its proof:

**Theorem B.2** *When $b > 1$, the single-step estimation bias of post-update Q-value $Q_{\theta_{new}}$ with function $D_b(x)$ is smaller than that with $l_2$ loss function, i.e., $\Delta_{D_b} - \Delta_{l_2} < 0$.*

*Proof. When we use $D_b(x)$ to update Q-values, the post-update Q-value $Q_{\theta_{new}}(s,a)$ can be approximated as:*

$$\begin{aligned} Q_{\theta_{id}}(s,a) &\approx Q_\theta(s,a) + \beta D_b'(\tilde{y} - Q_\theta(s,a))\|\nabla_\theta Q_\theta(s,a)\|_2^2 \\ Q_{\theta_{new}}(s,a) &\approx Q_\theta(s,a) + \beta D_b'(y - Q_\theta(s,a))\|\nabla_\theta Q_\theta(s,a)\|_2^2 \end{aligned} \tag{28}$$

*The estimate bias of $Q_{\theta_{new}}(s,a)$ with $D_b(x)$ function in single-step update is:*

$$\begin{aligned} \Delta_{D_b}(s,a) &= \mathbb{E}_{\varepsilon_{\hat{\theta}}}\left[Q_{\theta_{new}}(s,a) - Q_{\theta_{id}}(s,a)\right] \\ &= \beta \mathbb{E}_{\varepsilon_{\hat{\theta}}}\left[D_b'(y - Q_\theta(s,a)) - D_b'(\tilde{y} - Q_\theta(s,a))\right]\|\nabla_\theta Q_\theta(s,a)\|_2^2 \end{aligned}$$

*and the estimate bias of $Q_{\theta_{new}}(s,a)$ with $l_2$ function in singe-step update is:*

$$\Delta_{l_2}(s,a) = \mathbb{E}_{\varepsilon_{\hat{\theta}}}\left[Q_{\theta_{new}}(s,a) - Q_{\theta_{id}}(s,a)\right] \approx \beta(\mathbb{E}_{\varepsilon_{\hat{\theta}}}[y] - \tilde{y})\|\nabla_\theta Q_\theta(s,a)\|_2^2$$

*We define $z = y - Q_\theta(s,a)$, $\tilde{z} = \tilde{y} - Q_\theta(s,a)$ and $f(x) = D_b'(x) - x$. With these definitions, we have:*

$$\begin{aligned} \Delta_{D_b} - \Delta_{l_2} &= \beta \mathbb{E}_{\varepsilon_{\hat{\theta}}}\left[D_b'(z) - D_b'(\hat{z}) - (z - \hat{z})\right]\|\nabla_\theta Q_\theta(s,a)\|_2^2 \\ &= \beta \mathbb{E}_{\varepsilon_{\hat{\theta}}}\left[f(z) - f(\tilde{z})\right]\|\nabla_\theta Q_\theta(s,a)\|_2^2 \end{aligned} \tag{29}$$

*For $\mathbb{E}_{\varepsilon_{\hat{\theta}}}[y] - \tilde{y} > 0$, we have $\mathbb{E}_{\varepsilon_{\hat{\theta}}}[z] - \tilde{z} > 0$.*

*Meanwhile, when $b > 1$, the $f'(x) = D_b''(x) - 1 = \frac{1}{b^2}\exp(-\frac{|x|}{b}) - 1 < 0$, which implies $f(x)$ is monotonically decreasing. So we have:*

$$\Delta_{D_b} - \Delta_{l_2} < 0 \tag{30}$$

B.5  PROOF OF VARIANCE REDUCTION WITH LAPLACE MODELING

In this subsection, we provide additional theoretical analysis of Laplace noise modeling, with a focus on its variance reduction effect. We first introduce the necessary preliminaries in Lemma B.3, and then present the main variance reduction result together with its proof in Theorem B.4.

Theorem B.4 can be understood directly, even if you choose to skip the derivation of Lemma B.3.

**Lemma B.3** *Assume three random variables $X$ $Y$ and $Z$. $Y$ is independent from $X$ and $Z$. $X$ and $Z$ are not independent, i.e., $Cov(X,Z) \neq 0$. Then, the variance of $X + YZ$ is:*

$$Var(X + YZ) = Var(X) + Var(Y)Var(Z) + Var(Y)(E[Z])^2 + \\ Var(Z)(\mathbb{E}[Y])^2 + 2\mathbb{E}[Y]Cov(X,Z) \tag{31}$$

*Proof. We have the following well-known relationship of two independent variables $Y$ and $Z$:*

$$Var(YZ) = Var(Y)Var(Z) + Var(Y)(\mathbb{E}[Z])^2 + Var(Z)(\mathbb{E}[Y])^2 \\ Var(Y \pm Z) = Var(Y) + Var(Z) \tag{32}$$

*Then, we calculate the Cov(X,YZ), we have*

$$Cov(X, YZ) = \mathbb{E}[X \cdot YZ] - \mathbb{E}[X]\mathbb{E}[YZ] = \mathbb{E}[Y \cdot (XZ)] - \mathbb{E}[X]\mathbb{E}[Y]\mathbb{E}[Z] \\ = \mathbb{E}[Y]\big(\mathbb{E}[XZ] - \mathbb{E}[X]\mathbb{E}[Z]\big) = \mathbb{E}[Y]Cov(X,Z) \tag{33}$$

*Therefore, the variance $Var(X + YZ)$ is:*

$$Var(X + YZ) = Var(X) + Var(YZ) + 2Cov(X, YZ) \\ = Var(X) + Var(Y)Var(Z) + Var(Y)(\mathbb{E}[Z])^2 + \\ + Var(Z)(\mathbb{E}[Y])^2 + 2\mathbb{E}[Y]Cov(X,Z) \tag{34}$$

*Proof is over.*

**Theorem B.4** *With Assumption 4.1 and 4.2, when $b > 1$, the variance of post-update Q-value $Q_{\theta_{new}}$ with $D_b(x)$ function is smaller than that with $l_2$ loss function.*

*Proof. We first show the variance of Q-value estimation with $l_2$ loss function. Following Equation 26,*

$$Q_{\theta_{new}}(s,a) \approx Q_\theta(s,a) + \beta\big(y - Q_\theta(s,a)\big)\|\nabla_\theta Q_\theta(s,a)\|_2^2$$

*With Assumption 4.1, we can have*

$$y - Q_\theta(s,a) = \big(\mathcal{Q}(s,a) - \varepsilon_{\hat\theta}\big) - \big(\mathcal{Q}(s,a) - \varepsilon_\theta\big) = \varepsilon_\theta - \varepsilon_{\hat\theta}$$

*Because the approximation noise $\varepsilon_\theta$ and $\varepsilon_{\hat\theta}$ are independent of each other, and both are independent of $s$, $a$ and network $\theta$ under Assumption 4.1. Therefore, the variable $\varepsilon_\theta - \varepsilon_{\hat\theta}$ is independent from $Q_\theta(s,a)$ and $\|\nabla_\theta Q_\theta(s,a)\|_2^2$.*

*Let $X = Q_\theta(s,a)$, $Y = \varepsilon_\theta - \varepsilon_{\hat\theta}$ and $Z = \|\nabla_\theta Q_\theta(s,a)\|_2^2$. The noise $\varepsilon_\theta$ and $\varepsilon_{\hat\theta}$ have the same mean. $\mathbb{E}(Y) = \mathbb{E}(\varepsilon_\theta - \varepsilon_{\hat\theta}) = 0$. For simplicity, we denote the mean and variance of $\|\nabla_\theta Q_\theta(s,a)\|_2^2$ as $\mathbb{E}[\nabla]$ and $Var(\nabla)$, respectively.*

*According to the Lemma B.3 above, the variance of post-update Q-values with $l_2$ function $Var(Q_{\theta_{news}}(s,a))$ is as follows, denoted as $Var_{l_2}(Q)$:*

$$Var_{l_2}(Q) = Var(Q_\theta(s,a)) + \beta^2 Var(\varepsilon_\theta - \varepsilon_{\hat\theta})Var(\nabla) + \beta^2 Var(\varepsilon_\theta - \varepsilon_{\hat\theta})(\mathbb{E}[\nabla])^2 \tag{35}$$

*Then, we demonstrate the variance of Q-value estimation with $D_b(x)$ loss function. We denote it as $Var_{D_b}(Q)$. Similarly, let $X = Q_\theta(s,a)$, $Y = D_b'(\varepsilon_\theta - \varepsilon_{\hat\theta})$, $Z = \|\nabla_\theta Q_\theta(s,a)\|_2^2$. According to the $Q_{\theta_{new}}(s,a)$ derived with $D_b(x)$, we can get $Var_{D_b}(Q)$ as:*

$$Q_{\theta_{new}}(s,a) \approx Q_\theta(s,a) + \beta D_b'\big(y - Q_\theta(s,a)\big)\|\nabla_\theta Q_\theta(s,a)\|_2^2 \\ Var_{D_b}(Q) = Var(Q_\theta(s,a)) + \beta^2 Var\big(D_b'(\varepsilon_\theta - \varepsilon_{\hat\theta})\big)Var(\nabla) + \beta^2 Var\big(D_b'(\varepsilon_\theta - \varepsilon_{\hat\theta})\big)(\mathbb{E}[\nabla])^2 \tag{36}$$

*Next, with Assumption 4.2 of Laplace-based noise, the variance of $\varepsilon_\theta$ and $\varepsilon_{\hat\theta}$ can be approximated with the scale parameter of Laplace distribution, i.e., $Var(\varepsilon_\theta) = Var(\varepsilon_\theta) = 2b^2$. We have $Var(\varepsilon_\theta - \varepsilon_{\hat\theta}) = 4b^2$.*

*On the other hand, the $|D_b'(x)| < 1/b$, the $D_b'(\varepsilon_\theta - \varepsilon_{\hat\theta}) \in (-1/b, 1/b)$, so the $Var\big(D_b'(\varepsilon_\theta - \varepsilon_{\hat\theta})\big) < 1/b^2$ according to Popoviciu's inequality.*

*Finally, we can compare $Var_{D_b}(Q)$ with $Var_{l_2}(Q)$, we have:*

$$
\begin{aligned}
Var_{D_b}(Q) - Var_{l_2}(Q) &= \beta^2 \Big( Var\big(D_b'(\varepsilon_\theta - \varepsilon_{\hat\theta})\big) - Var(\varepsilon_\theta - \varepsilon_{\hat\theta}) \Big)\big( Var(\nabla) + (\mathbb{E}[\nabla])^2 \big) \\
&< (1/b^2 - 4b^2)\big( Var(\nabla) + (\mathbb{E}[\nabla])^2 \big) \\
&< 0
\end{aligned}
\tag{37}
$$

*Proof is done.*

The condition $b > 1$ keeps the same for satisfying the Theorem 4.4 and Theorem B.4 simultaneously. In summary, the estimation bias and variance of Q-value can be reduced using $D_b(x)$ function for Q-value updates, compared with using the $l_2$ loss function.

## B.6 Algorithm Framework

We summarize the framework of LAROO in Algorithm 1.

---

**Algorithm 1** LAROO

1: **Input:** Offline dataset $D_{off}$, offline RL algorithm $\mathcal{F}_{offline}$, value networks $\{Q_{\theta_j}\}_{j=1}^N$, a policy network $\pi_\psi$. Online replay buffer $D_{on}$, online steps $T$.
2: **Offline Phase:** Training $\mathcal{F}_{offline}$ using $D_{off}$
3: **Online Phase:** Remove the constraints in $\mathcal{F}_{offline}$, named online algorithm $\mathcal{F}_{online}$
4: **for** $i = 1$ **to** $T$ **do**
5:      Collect an online sample $\tau = (s, a, r, s')$ with $\pi_\psi$ in online environment, update online replay buffer $D_{on}$
6:      Sample a training batch $B$ from $D_{off} \cup D_{on}$
7:      Update the parameter of Laplace-distributed noise, with Equation (7)
8:      Update each Q-function $Q_{\theta_i}$ with the loss function (8) according to $\mathcal{F}_{online}$
9:      Update policy $\pi_\psi$ according to $\mathcal{F}_{online}$
10: **end for**

---

## B.7 Comparison with batch normalization

The widely adopted batch-normalization (BN) technique mitigate the impact of the heavy-tailed biases through explicit normalization for the Q-value losses. For further show the effectiveness of LAROO, We compare it with the batch-normalization (BN) technique in detail.

**BN**: When updating Q-value with batch normalization with MSE, the loss function is defined as:

$$
\mathcal{L}_{l_2}(\theta) = \mathbb{E}_{(s,a,r,s') \sim \mathcal{B}} \Big( \frac{\big(\mathrm{BatchNorm}(\delta_\theta(s, a))\big)^2}{2} \Big)
$$

where $\mathrm{BatchNorm}(\delta_\theta(s, a))$ is the normalized TD-errors within a training batch. These TD-errors are normalized by their mean and variance.

**LAROO**: When updating Q-value with our methods, the loss function is:

$$
\begin{aligned}
\mathcal{L}_b(\theta_t) &= \mathbb{E}_{(s,a,r,s') \sim \mathcal{B}} \left[ \frac{1}{K} \sum_{k=1}^K D_b\Big( Q_{\theta_t}^{(k)}(s, a) - y_{\min}(s, a, r, s') \Big) \right] \\
y_{\min}(s, a, r, s') &= r + \gamma \max_{a' \in \mathcal{A}} \left[ \min_{1 \le k \le M} Q_{\hat\theta_t}^{(k)}(s', a') \right].
\end{aligned}
\tag{38}
$$

LAROO can also tames heavy-tailed Q bias through distributional transformation: (1) LAROO applies the ensemble models to reduce the mean of Q bias, shifting its mean toward zero. (2) LAROO captures the heavy-tailed nature of O-value biases with a parameterized Laplace-distributed noise, thereby alleviating the heavy tailedness of biases. (1) and (2) dynamically promote the asymmetric, heavy-tailed Q bias distribution into a more centralized form.

We conduct experiments to compare the performance of these two methods. The agent of BN keeps hyperparameter settings be consistent with LAROO. Experimental results show that LAROO is more effective in improving performance in O2O RL in Table 2.

Table 2: Comparison with Batch Normalization (BN).

| Task | Type | LAROO | BN |
|------|------|-------|----|
| Walker2d | medium | $\mathbf{120.4 \pm 2.9}$ | $36.2 \pm 13.3$ |
| | medium-replay | $\mathbf{124.8 \pm 2.0}$ | $62.1 \pm 7.7$ |
| | medium-expert | $\mathbf{126.7 \pm 1.0}$ | $39.1 \pm 14.0$ |
| halfcheetah | medium | $\mathbf{92.5 \pm 1.2}$ | $83.3 \pm 4.8$ |
| | medium-replay | $\mathbf{84.7 \pm 1.9}$ | $78.4 \pm 2.4$ |
| | medium-expert | $\mathbf{99.5 \pm 1.8}$ | $92.9 \pm 1.6$ |

## C  EXPERIMENT DETAILS

### C.1  EXPERIMENT IMPLEMENTATION DETAILS

**Implementation details for LAROO**. Following the setting of baseline ENOTO (Zhao et al., 2024), we implement LAROO based on TD3 (Fujimoto et al., 2018) with ensemble Q-functions for Mujoco tasks and implement it based on LAPO (Chen et al., 2022) for Antmaze tasks. We follow the hyperparameters of offline backbone algorithms. During online fine-tuning, a mini-batch is uniformly sampled for training, with half from the offline buffer and the other half from the online buffer. The scale parameter $b$ is updated according to MBBE at every step. We provide the hyperparameters of LAROO in Table 3.

Table 3: Hyperparameters of LAROO on Mujoco tasks.

| Hyperparameter | Value |
|----------------|-------|
| Optimizer | Adam |
| Learning rate of critic network | $3e^{-4}$ |
| Learning rate of actor network | $3e^{-4}$ |
| Target smoothing coefficient $\tau$ | $5e^{-3}$ |
| Batch size | 256 |
| Discount | 0.99 |
| Policy noise | 0.2 |
| Policy noise clipping | (-0.5,0.5) |
| Policy update frequency | 2 |
| **LAROO Parameter** | **Value** |
| Total environment steps $T$ | $10^5$ |
| Critic UTD | 5 |
| Actor UTD | 1 |
| Ensemble Q-function size $N$ | 5 |
| Subset size $M$ | 2 |

**Implementation details for baseline methods**. We compare LAROO with baseline methods PEX (Zhang et al., 2023), SO2 (Zhang et al., 2024), ENOTO (Zhao et al., 2024), Cal-QL (Nakamoto et al., 2023), BOORL (Hu et al., 2024) and RLPD (Ball et al., 2023) in this paper. We rerun these methods with their official implementations. However, there are some special cases. We observe that SO2 (Zhang et al., 2024) fails in Antmaze tasks, primarily because its backbone algorithm (SAC) struggles to handle with sparse reward tasks. Similarly, the BOORL (Hu et al., 2024) struggles to perform well due to the backbone algorithm TD3BC. And these two O2O methods do not provide official implements for Antmaze and Adroit tasks.

Therefore, we rerun SO2 and BOORL using their official codes only for the Mujoco tasks. For Antmaze and Adroit tasks, in order to ensure a fair comparison, we implement SO2 based on LAPO (Chen et al., 2022), aligning with our setup, and estimate Q-values according to its methods. As for BOORL, we do not test it on the Antmaze and Adroit tasks due to the lack of provided codes. All experiments are conducted on the same experimental setup, which includes eight NVIDIA Tesla V100 GPUs and an Intel Core i7-6700k CPU at 4.00GHz.

## C.2 EXPERIMENT RESULTS

### C.2.1 MANN-WHITNEY U TEST FOR Q BIASES AND TARGET Q BIASES

The Mann-Whitney U test (Mann-Whitney-Wilcoxon test) (Mann & Whitney, 1947) is a non-parametric statistical method. It can test whether two independent samples come from the same distribution. Meanwhile, it does not require the data to follow a normal distribution, making it suitable for handling skewed or non-normally distributed data.

In section 4.1, we set $b_1 = b_2$, which generally states that $\varepsilon$ and $\varepsilon_{\hat{\theta}}$ follow the same distribution. We conduct the Mann-Whitney U test to validate it. We sample a number of samples from $\varepsilon_\theta$ and $\varepsilon_{\hat{\theta}}$ at the same training step. The resulting p-values are significantly higher than 0.05, indicating that the noise $\varepsilon_\theta$ and $\varepsilon_{\hat{\theta}}$ could follow the same distribution. Therefore, the setting of $b_1 = b_2$ is reasonable. This result can be understood from the fact that the target Q-value network copies the parameters of the Q-value network every few steps.

Table 4: **P-values of Mann-Whitney U test for Q biases and target Q biases**.

| Task | Type | P-value |
|---|---|---|
| Halfcheetah | medium | 0.902 |
| | medium-replay | 0.720 |
| | medium-expert | 0.793 |
| Hopper | medium | 0.704 |
| | medium-replay | 0.855 |
| | medium-expert | 0.914 |

### C.2.2 MAIN PERFORMANCE

**We present the training curves of several O2O methods during online fine-tuning**. We illustrate the normalized return curves at mujoco tasks in Figure 12, and the training curves at Antmaze tasks in Figure 13. As discussed in Experiment 5, LAROO outperforms previous state-of-the-art methods in final performance.

**We conduct more experiments on the Antmaze and Adroit domains (Fu et al., 2020)**. Most methods perform poorly in certain scenarios due to the complexity of the Adroit environments. Additionally, previous research studied that the hyperparameters for most offline baseline algorithms in the Adroit environment differ from those used in the MuJoCo environment. Based on the CORL (Tarasov et al., 2023), we adjust the parameters of their offline models accordingly.

**We conduct statistical significance tests on experimental results to further validate the effectiveness**. Specifically, we conduct mean difference testing using Bootstrap resampling on the final results of different methods. Mean difference testing aims to compare the means of two groups and determine if the observed difference in means is statistically significant. The bootstrap resampling is used to solve the limited number of results As shown in Table 6, the p-values are less than 0.05 in most scenarios, indicating statistically significant differences in means of LAROO compared to other algorithms. That is, the advantage of LAROO is significant compared to others in performance.

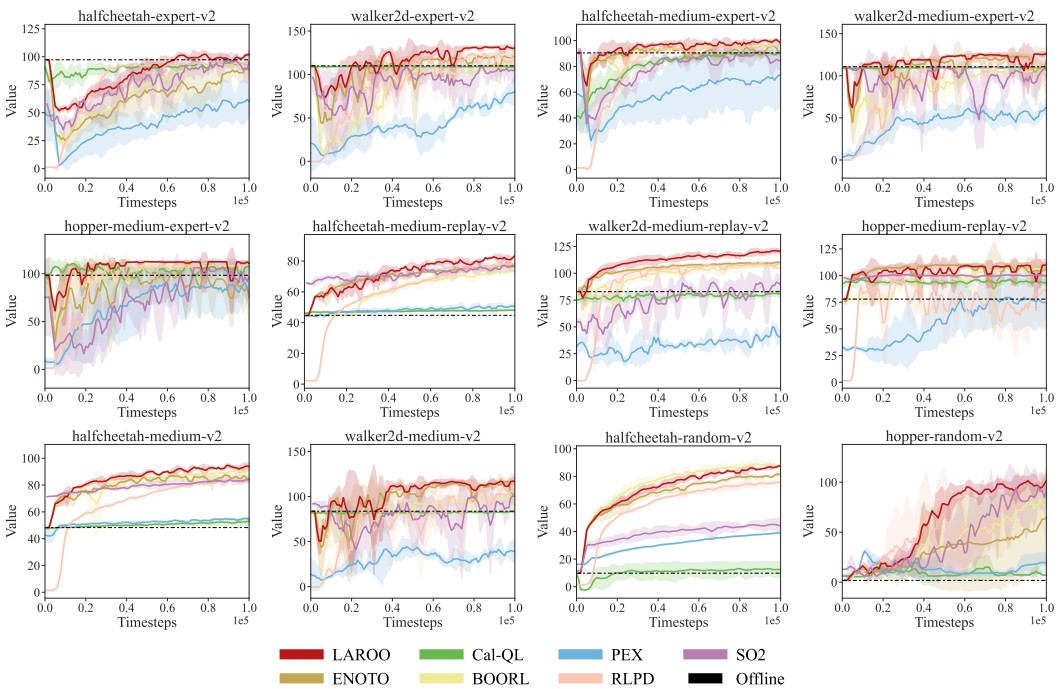

Figure 12: Normalized return curves of several baselines and LAROO in 100K time steps. The reference line is the performance of offline pretrained agents. Every experimental result is averaged with five random seeds. Note that RLPD starts training agents from scratch rather than pretrained.

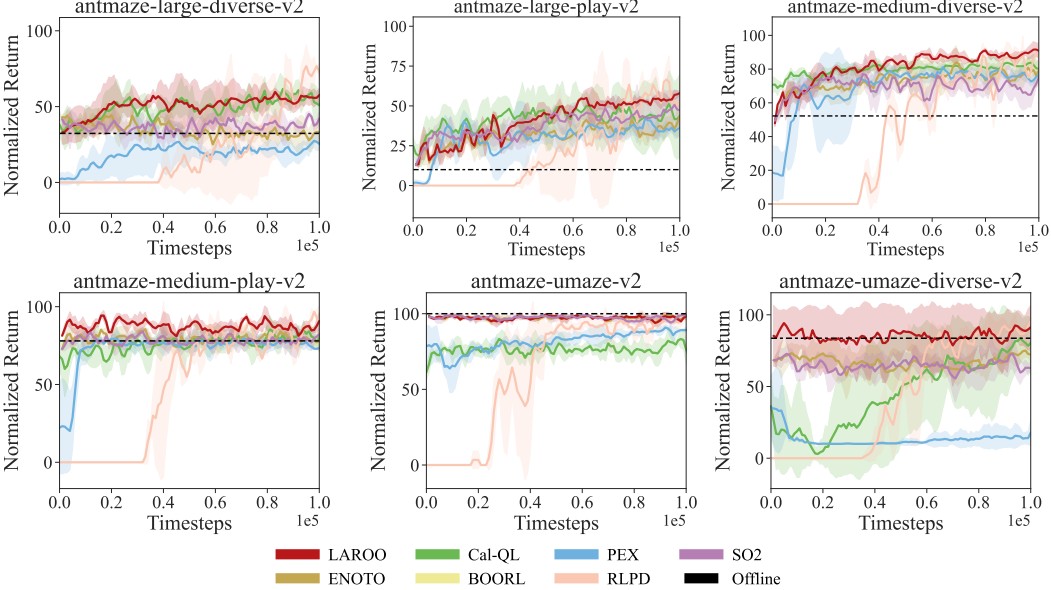

Figure 13: The training curves of various O2O RL methods in Antmaze tasks.

### C.2.3 TRAINING STABILITY

We evaluate and compare the training stability of O2O RL methods in this subsection. In previous research (Feng et al., 2024), it defines the Normalized Cumulative Performance Drop (NCD) metric to quantify the degree of performance degradation in O2O RL. Meanwhile, the variance of performance is widely used to evaluate the training stability. A smaller NCD or variance value indicates

Table 5: Experimental results on Antmaze and Adroit tasks of several O2O algorithms.

| Task | Type | PEX | SO2 | Cal-QL | ENOTO | LAROO |
|------|------|-----|-----|--------|-------|-------|
| Antmaze | umaze-diverse | $60.0 \rightarrow 45.6$ | $66.3 \rightarrow 66.3$ | $47.5 \rightarrow \mathbf{85.0}$ | $63.3 \rightarrow 71.1$ | $65.0 \rightarrow \mathbf{85.5}$ |
| | medium-play | $60.0 \rightarrow 72.3$ | $70.0 \rightarrow 81.1$ | $73.7 \rightarrow 77.5$ | $68.8 \rightarrow 77.7$ | $83.0 \rightarrow \mathbf{83.6}$ |
| | medium-diverse | $70.0 \rightarrow 77.6$ | $63.3 \rightarrow 71.8$ | $73.5 \rightarrow 80.2$ | $55.8 \rightarrow 80.0$ | $54.5 \rightarrow \mathbf{86.6}$ |
| | large-play | $30.0 \rightarrow 38.6$ | $10.0 \rightarrow 44.8$ | $28.0 \rightarrow 47.5$ | $10.0 \rightarrow 45.7$ | $10.0 \rightarrow \mathbf{55.8}$ |
| | large-diverse | $30.0 \rightarrow 24.7$ | $35.0 \rightarrow 39.1$ | $36.2 \rightarrow \mathbf{42.8}$ | $35.5 \rightarrow 31.1$ | $36.6 \rightarrow \mathbf{43.0}$ |
| $\delta_{\text{sum}}(0.1\text{M})$ | | 8.8 | 58.5 | 74.1 | 72.2 | **105.4** |
| relocate | human | $0.30 \rightarrow 0.20$ | $0.11 \rightarrow 0.22$ | $-0.33 \rightarrow \mathbf{0.25}$ | $1.03 \rightarrow -0.12$ | $0.11 \rightarrow \mathbf{0.25}$ |
| | cloned | $0.40 \rightarrow 0.03$ | $0.06 \rightarrow 0.34$ | $-0.31 \rightarrow -0.31$ | $-0.33 \rightarrow -0.33$ | $0.06 \rightarrow \mathbf{0.46}$ |
| pen | human | $75.50 \rightarrow 64.70$ | $82.50 \rightarrow 89.50$ | $63.60 \rightarrow 88.40$ | $58.90 \rightarrow 82.90$ | $82.50 \rightarrow \mathbf{94.70}$ |
| | cloned | $81.00 \rightarrow 88.20$ | $82.90 \rightarrow 93.70$ | $-2.66 \rightarrow -2.68$ | $-2.76 \rightarrow -1.28$ | $82.90 \rightarrow \mathbf{97.10}$ |
| door | human | $1.40 \rightarrow 0.20$ | $4.07 \rightarrow \mathbf{5.90}$ | $-0.34 \rightarrow 0.33$ | $13.28 \rightarrow 0.10$ | $4.07 \rightarrow 5.35$ |
| | cloned | $6.30 \rightarrow 0.22$ | $0.32 \rightarrow \mathbf{10.30}$ | $-0.33 \rightarrow -0.33$ | $-0.33 \rightarrow -0.27$ | $0.32 \rightarrow \mathbf{10.30}$ |
| hammer | human | $0.90 \rightarrow 0.18$ | $3.74 \rightarrow 1.56$ | $0.27 \rightarrow -0.30$ | $0.30 \rightarrow \mathbf{3.26}$ | $3.74 \rightarrow 0.55$ |
| | cloned | $0.90 \rightarrow 0.21$ | $1.32 \rightarrow 18.30$ | $0.25 \rightarrow 0.17$ | $0.56 \rightarrow 2.85$ | $1.32 \rightarrow \mathbf{18.80}$ |
| $\delta_{\text{sum}}(0.1\text{M})$ | | -12.7 | 44.8 | 25.4 | 16.5 | **52.5** |

Table 6: **Mean difference testing of LAROO against other baselines**.

| Task | Type | LAROO vs PEX | LAROO vs SO2 | LAROO vs Cal-QL | LAROO vs ENOTO | LAROO vs BOORL |
|------|------|--------------|--------------|-----------------|----------------|----------------|
| Hopper | random | 0.014 | 0.419 | 0.013 | 0.318 | 0.283 |
| | medium | 0.036 | 0.137 | 0.011 | 0.285 | 0.024 |
| | medium-replay | 0.042 | 0.015 | 0.009 | 0.416 | 0.061 |
| | medium-expert | 0.046 | 0.181 | 0.987 | 0.130 | 0.026 |
| | expert | 0.012 | 0.023 | 0.433 | 0.181 | 0.107 |
| Walker2d | random | 0.041 | 0.016 | 0.008 | 0.065 | 0.013 |
| | medium | 0.017 | 0.346 | 0.046 | 0.887 | 0.326 |
| | medium-replay | 0.013 | 0.016 | 0.006 | 0.020 | 0.021 |
| | medium-expert | 0.005 | 0.017 | 0.006 | 0.056 | 0.022 |
| | expert | 0.065 | 0.553 | 0.481 | 0.655 | 0.017 |
| Halfcheetah | random | 0.013 | 0.017 | 0.007 | 0.205 | 0.104 |
| | medium | 0.005 | 0.110 | 0.010 | 0.432 | 0.024 |
| | medium-replay | 0.010 | 0.090 | 0.003 | 0.511 | 0.018 |
| | medium-expert | 0.080 | 0.048 | 0.019 | 0.511 | 0.183 |
| | expert | 0.026 | 0.033 | 0.857 | 0.175 | 0.018 |
| Antmaze | umaze-diverse | 0.033 | 0.039 | 0.300 | 0.094 | - |
| | medium-play | 0.023 | 0.064 | 0.0230 | 0.096 | - |
| | medium-diverse | 0.023 | 0.037 | 0.117 | 0.075 | - |
| | large-play | 0.022 | 0.064 | 0.488 | 0.217 | - |
| | large-diverse | 0.010 | 0.022 | 0.059 | 0.007 | - |

better training stability. The NCD metric is defined as follows:

$$\frac{1}{T+1} \sum_{t=0}^{T} I\left(R(0) > R(t)\right) \frac{R(0) - R(t)}{R(0)} \tag{39}$$

where $R(\cdot)$ is the performance of policy at the training step $t$, $R(0)$ is the initial pretrained performance, $T$ is the total online fine-tuning step and $I(\cdot)$ is an indicator function that takes the value 1 when the condition is satisfied and 0 otherwise.

Table 7: Variance and NCD values of different O2O RL methods across Walker2d tasks.

| | Task | PEX | SO2 | Cal-QL | ENOTO | BOORL | LAROO |
|------|------|-----|-----|--------|-------|-------|-------|
| Variance | medium | 17.9 | 3.4 | **1.5** | 6.8 | 6.8 | **2.9** |
| | medium-replay | 6.8 | 4.5 | 5.7 | **0.8** | 1.4 | 2.0 |
| | medium-expert | 4.6 | 5.0 | **0.8** | 6.3 | 1.8 | **1.0** |
| NCD | medium | 0.090 | 0.059 | **0.010** | 0.026 | 0.037 | **0.020** |
| | medium-replay | 0.083 | 0.025 | 0.015 | 0.020 | 0.019 | **0.013** |
| | medium-expert | 0.076 | 0.098 | **0.023** | 0.067 | 0.078 | **0.031** |

The experimental results in Table 7 show that LAROO significantly mitigates the performance degradation during fine-tuning. In Figure 14, the standard deviation values of LAROO are noticeably lower than those of most methods. While Cal-QL achieves better stability due to its constraints, LAROO generally demonstrates better performance improvement within a limited number of steps. In conclusion, LAROO can achieve significant training stability and performance improvement.

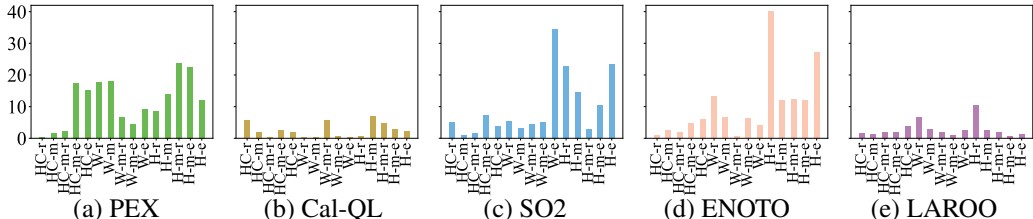

|  |  |  |  |  |
|:---:|:---:|:---:|:---:|:---:|
| (a) PEX | (b) Cal-QL | (c) SO2 | (d) ENOTO | (e) LAROO |

Figure 14: The standard deviation values of O2O RL methods on the D4RL datasets are presented, with each result averaged over 5 seeds. The task environment names are abbreviated; for example, 'W-m' refers to 'Walker2d-medium'.

## C.3 EVALUATION OF COMPATIBILITY OF LAROO

In this study, we propose the Q-value loss function $D_b(x)$ in Equation (6) for robust Q-value estimation. This function can be easily incorporated into previous O2O methods for further performance improvement. To evaluate the compatibility of LAROO with other methods, we select TD3, Cal-QL and ENOTO (Zhao et al., 2024) as these three methods exhibit progressive performance in O2O RL, and then replace the $l_2$ loss function with the function (6) for Q-value updates.

Table 8: Experimental results of integrating LAROO with other O2O methods.

| Task | Type | TD3 | TD3+LAROO | Cal-QL | Cal-QL+LAROO | ENOTO | ENOTO+LAROO |
|---|---|---|---|---|---|---|---|
| Walker2d | medium | $67.4 \pm 9.3$ | $82.1 \pm 6.2$ | $83.7 \pm 0.5$ | $89.4 \pm 3.1$ | $110.2 \pm 6.8$ | $118.6 \pm 4.2$ |
|  | medium-replay | $65.3 \pm 6.4$ | $80.5 \pm 3.2$ | $80.7 \pm 5.7$ | $91.2 \pm 0.9$ | $110.2 \pm 0.8$ | $119.8 \pm 2.0$ |
|  | medium-expert | $86.3 \pm 2.1$ | $101.5 \pm 3.7$ | $110.2 \pm 0.8$ | $111.2 \pm 2.3$ | $118.0 \pm 6.3$ | $127.5 \pm 1.1$ |

The experimental results validate that our proposed robust loss function is generally effective and can serve as a plug-in method to further enhance performance over existing methods. It is worth noting that these experiments also demonstrate the compatibility of LAROO with different offline baselines, e.g., Cal-QL is based on CQL (Kumar et al., 2020), and ENOTO is based on ensemble SAC (Haarnoja et al., 2018), LAROO can integrate with these offline baselines well and be highly compatible with Q-value-based algorithms, further indicating its versatility and applicability.

## C.4 ABLATION STUDY

We conduct ablation studies for LAROO and provide detailed experimental results in this subsection.

### C.4.1 ABLATION FOR LOSS FUNCTION

We conduct ablation studies to validate the effectiveness of $\mathcal{L}_b(x)$ in this subsection. We first evaluate the performance with and without $\mathcal{L}_b(x)$ (i.e., using $l_2$ loss function). Meanwhile, considering that previous research (Zahra et al., 2014; Belagiannis et al., 2015; Cavazza & Murino, 2016) has demonstrated the effectiveness of various robust loss functions, such as the Huber function and Cauchy loss function, in handling outliers in training data, we also compare $\mathcal{L}_b(x)$ with these functions for Q-value updates during online fine-tuning.

The results show that $\mathcal{L}_b(x)$ outperforms others for effectively promoting heavy-tailed Q-value biases into a standardized form, improving training stability and performance, as discussed in Section 4.2.

### C.4.2 ABLATION FOR UTD RATIO AND ENSEMBLE SIZE

We first demonstrate the effectiveness of LAROO with $UTD = 1$ and ensemble size $N = 1$ in Table 10. We compare it with the Cal-QL (Nakamoto et al., 2023) and PEX (Zhang et al., 2023). Other baselines, such as ENOTO (Zhao et al., 2024), SO2 (Zhang et al., 2024), BOORL (Hu et al., 2024), also leverage high UTD or large ensembles, and are concluded from the ablation experiments. As shown in Table 10, LAROO outperforms Cal-QL in 10 out of 13 experiments, demonstrating its effectiveness even without the benefits of high UTD or ensemble models.

Table 9: **Ablation for different loss functions**.

| Task | Type | LAROO | Huber | Cauchy | MSE | MLE |
|---|---|---|---|---|---|---|
| Halfcheetah | medium | **92.5 ± 1.2** | 91.5 ± 4.2 | 52.4 ± 12.6 | 89.7 ± 3.2 | 90.2 ± 1.2 |
| | medium-replay | **84.7 ± 1.9** | 81.5 ± 1.7 | 44.9 ± 10.1 | 81.9 ± 0.8 | 78.1 ± 2.9 |
| | medium-expert | 99.5 ± 1.8 | **100.7 ± 5.3** | 37.5 ± 5.0 | 90.3 ± 4.1 | 92.5 ± 3.2 |
| Walker2d | medium | **120.4 ± 2.9** | 110.7 ± 5.7 | 107.6 ± 5.2 | 88.3 ± 10.1 | 85.2 ± 11.6 |
| | medium-replay | **124.8 ± 2.0** | 110.4 ± 0.9 | 39.4 ± 21.8 | 104.7 ± 1.7 | 118.1 ± 0.7 |
| | medium-expert | **126.7 ± 1.0** | 110.4 ± 10.5 | 103.8 ± 4.7 | 105.4 ± 3.7 | 120.5 ± 0.3 |
| Hopper | medium | **106.3 ± 2.6** | 90.4 ± 15.2 | 1.8 ± 1.2 | 93.6 ± 7.4 | 89.3 ± 16.9 |
| | medium-replay | **109.5 ± 1.8** | 102.2 ± 13.0 | 19.3 ± 11.1 | 90.9 ± 1.3 | 100.2 ± 2.6 |
| | medium-expert | **112.3 ± 0.7** | 98.4 ± 13.7 | 0.8 ± 0.3 | 99.9 ± 17.6 | 101.2 ± 10.2 |
| Antmaze | large-play | 55.8 ± 3.2 | **66.5 ± 5.0** | 48.3 ± 1.7 | 48.7 ± 5.0 | 49.2 ± 2.1 |
| | medium-play | **83.6 ± 3.6** | 83.3 ± 1.6 | 78.7 ± 0.8 | 71.6 ± 1.6 | 78.7 ± 2.1 |
| | medium-diverse | **86.6 ± 3.3** | 75.0 ± 8.3 | 80.0 ± 0.5 | 78.4 ± 11.6 | 85.2 ± 1.0 |

Table 10: **Ablation for UTD and ensemble size**. The normalized returns of LAROO with ensemble size $N = 1$ and $UTD = 1$ for a more fair comparison with Cal-QL and PEX.

| Task | Type | LAROO | PEX | Cal-QL | Cal-QL + LAROO |
|---|---|---|---|---|---|
| Halfcheetah | medium | **80.2 ± 1.6** | 55.3 ± 1.7 | 52.2 ± 1.9 | 58.2 ± 0.5 (+ 6.0) |
| | medium-replay | **82.2 ± 2.3** | 52.1 ± 2.1 | 47.9 ± 0.3 | 50.5 ± 1.2 (+ 2.6) |
| | medium-expert | **93.3 ± 1.6** | 90.3 ± 7.4 | 91.2 ± 2.6 | 96.8 ± 2.1 (+ 5.6) |
| Walker2d | medium | **95.4 ± 10.8** | 42.5 ± 17.9 | 83.7 ± 0.5 | 89.4 ± 3.1 (+ 5.6) |
| | medium-replay | **101.3 ± 8.3** | 40.5 ± 6.8 | 80.7 ± 5.7 | 91.2 ± 0.9 (+ 9.5) |
| | medium-expert | 108.3 ± 6.7 | 78.0 ± 4.6 | **110.2 ± 0.8** | 111.2 ± 2.3 (+ 1.0) |
| Hopper | medium | 85.5 ± 8.2 | 78.5 ± 14.0 | **90.8 ± 7.0** | 100.9 ± 5.2 (+ 10.1) |
| | medium-replay | **104.4 ± 2.1** | 87.5 ± 23.8 | 92.8 ± 4.8 | 105.3 ± 2.7 (+ 12.5) |
| | medium-expert | 95.5 ± 7.3 | 80.2 ± 18.5 | **108.3 ± 2.8** | 111.0 ± 5.8 (+ 2.7) |
| Antmaze | medium-play | **83.7 ± 2.0** | 72.3 ± 3.3 | 77.5 ± 2.5 | 80.0 ± 1.1 (+ 2.5) |
| | medium-diverse | **86.7 ± 1.5** | 77.6 ± 3.3 | 80.2 ± 2.6 | 80.0 ± 3.3 (− 0.2) |
| | large-play | **50.0 ± 3.3** | 38.6 ± 5.0 | 47.5 ± 2.5 | 47.7 ± 4.7 (+ 0.2) |
| | large-diverse | **43.3 ± 2.8** | 24.7 ± 2.7 | 42.8 ± 3.2 | 46.7 ± 3.3 (+ 3.9) |

We conduct ablation experiments only for the UTD, as shown in Table 11. It has been widely studied that the high UTD promotes the sample efficiency and final performance. Our experiment further validate that a higher UTD promotes the performance in offline-to-online scenarios.

Table 11: **Ablation for UTD ratio**. The ensemble size is five, and the updates-to-data ratio (UTD) varies.

| Task | Type | UTD = 1 | UTD = 3 | UTD = 5 (LAROO) | UTD = 10 |
|---|---|---|---|---|---|
| Halfcheetah | medium | 88.5 ± 1.3 | **94.7 ± 0.3** | 92.5 ± 1.2 | 93.2 ± 1.6 |
| | medium-replay | 80.8 ± 2.7 | 83.6 ± 1.8 | 84.7 ± 1.9 | **89.6 ± 2.1** |
| | medium-expert | 92.3 ± 6.6 | **103.1 ± 1.6** | 99.5 ± 1.8 | 98.4 ± 0.6 |
| Hopper | medium | 98.2 ± 5.2 | 99.5 ± 3.5 | **106.7 ± 2.6** | 91.2 ± 10.5 |
| | medium-replay | 105.7 ± 1.6 | 107.5 ± 1.8 | **109.5 ± 1.8** | 109.4 ± 2.3 |
| | medium-expert | 101.1 ± 4.8 | **113.0 ± 3.4** | 112.3 ± 0.7 | 106.4 ± 4.9 |

### C.4.3 ABLATION FOR THE COMPONENTS OF LAROO

LAROO mainly consists of two components: (1) the noise model for capturing the heavy-tailed Q bias, and (2) the ensemble model. To evaluate their effectiveness, we conduct ablation studies under two conditions: one without the noise model and the other without the ensemble model. The results, as shown in Table 12, demonstrate that both components can improve the training stability and the final performance, and the noise model plays a more critical role in LAROO.

The distributions of Q bias in Figure 15 provide insights into the effectiveness of each component in mitigating heavy-tailed Q biases. The ensemble models reduce Q-value overestimation bias, shifting the Q bias distribution closer to zero. However, using ensemble models alone still results in high heavy-tailedness, as indicated by the Kurtosis values in Figure 15(a) and (b), which exceed

10. On the other hand, the noise model effectively alleviates the heavy-tailed nature of the biases, but still suffers from significant overestimation. Finally, LAROO combines the strengths of both components, transforming the heavy-tailed Q-value biases into a more standardized form, as shown in Figure 15 (c) and (f).

Table 12: **Ablation studies for components of LAROO**. We demonstrate the normalized returns without the ensemble term and without the noise model respectively.

| Task | Type | LAROO w/o ensemble | LAROO w/o noise model | LAROO |
|---|---|---|---|---|
| Walker2d | medium-replay | $105.0 \pm 3.8$ | $88.3 \pm 10.1$ | $\mathbf{124.8 \pm 2.0}$ |
| | medium | $106.2 \pm 2.9$ | $104.7 \pm 1.7$ | $\mathbf{120.4 \pm 1.9}$ |
| | medium-expert | $111.8 \pm 6.3$ | $105.4 \pm 3.7$ | $\mathbf{126.7 \pm 1.0}$ |
| Halfcheetah | medium-replay | $81.9 \pm 2.6$ | $83.9 \pm 2.8$ | $\mathbf{84.7 \pm 1.9}$ |
| | medium | $89.5 \pm 2.1$ | $87.7 \pm 1.2$ | $\mathbf{92.5 \pm 1.2}$ |
| | medium-expert | $92.4 \pm 3.1$ | $90.3 \pm 4.1$ | $\mathbf{99.5 \pm 1.8}$ |
| Hopper | medium-replay | $105.6 \pm 1.2$ | $90.9 \pm 1.3$ | $\mathbf{109.5 \pm 1.8}$ |
| | medium | $98.3 \pm 6.3$ | $93.6 \pm 7.4$ | $\mathbf{106.7 \pm 2.6}$ |
| | medium-expert | $108.3 \pm 3.2$ | $99.9 \pm 17.6$ | $\mathbf{112.3 \pm 0.7}$ |

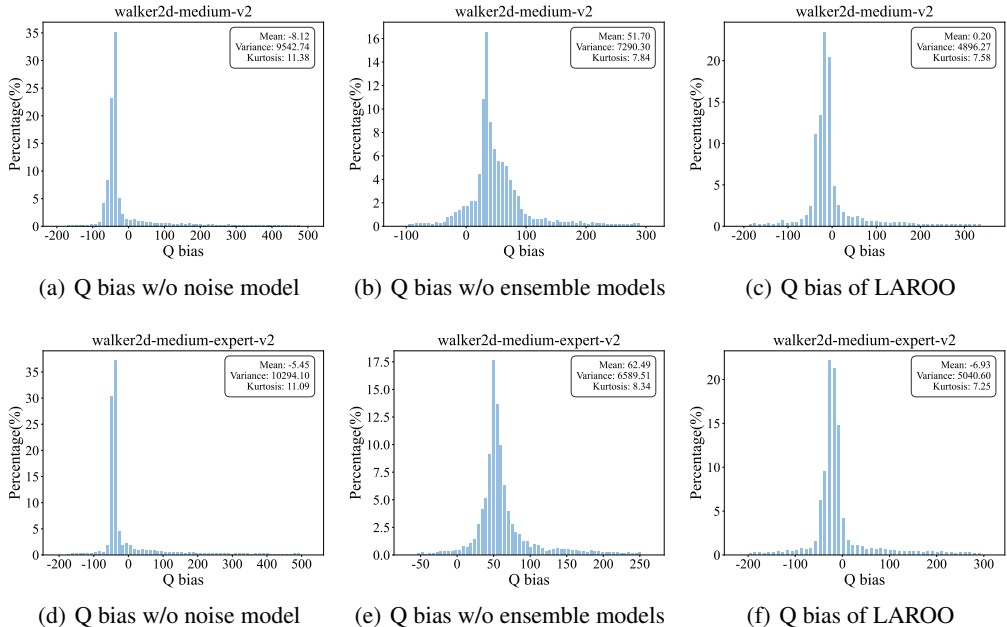

(a) Q bias w/o noise model     (b) Q bias w/o ensemble models     (c) Q bias of LAROO

(d) Q bias w/o noise model     (e) Q bias w/o ensemble models     (f) Q bias of LAROO

Figure 15: **Ablation for the components. Left**: the empirical Q bias distribution during fine-tuning w/o noise model. **Medium**: the empirical Q bias w/o ensemble models. **Right**: the empirical Q bias of LAROO.

We present the training curves on MuJoCo tasks without the Laplace noise model and without ensemble models, respectively, as shown in Figure 16.

Table 13: The NCD values of LAROO without ensemble model and without the Laplace noise modeling. A lower NCD means the better training stability.

| Task | Type | LAROO w/o ensemble model | LAROO w/o noise model | LAROO |
|---|---|---|---|---|
| Walker2d | medium | 0.023 | 0.024 | 0.020 |
| | medium-replay | 0.015 | 0.015 | 0.013 |
| | medium-expert | 0.034 | 0.038 | 0.031 |
| Hopper | medium | 0.017 | 0.018 | 0.016 |
| | medium-replay | 0.014 | 0.017 | 0.014 |
| | medium-expert | 0.034 | 0.036 | 0.029 |

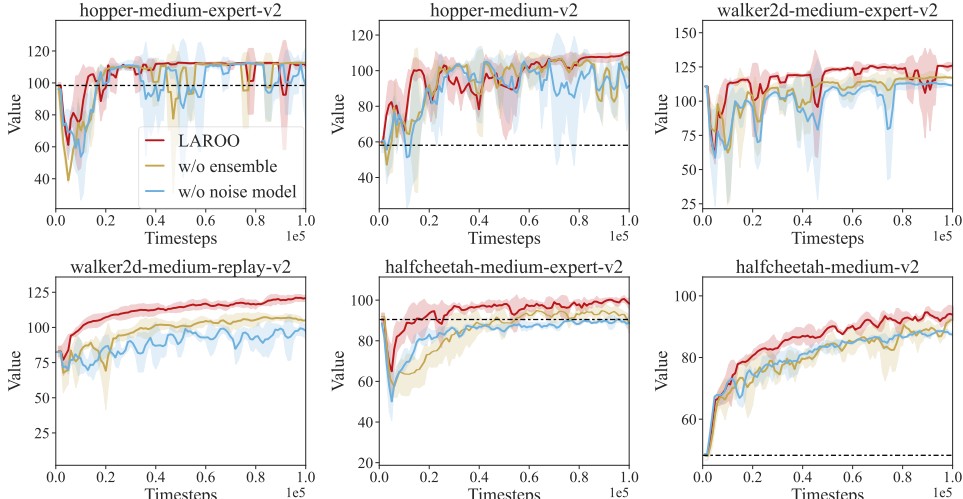

Figure 16: Ablation for components of LAROO: the training curves of LAROO without Laplace noise model and without ensemble models in Mojoco tasks, respectively.

### C.4.4 ABLATION FOR PARAMETER B

We use a parametric approach to model the noise model and use MBBE to estimate the variance of the noise. The parameter $b$ is adaptively updated according to Equation (7) at every online step. We demonstrate that adaptive $b$ is more effective for better approximating the true Q bias distribution. Meanwhile, we approximate the estimation of parameter $b$ by using TD-errors as a substitute of Q biases to facilitate the implementation. The curves in Figure 17 (b) show that the estimated $b$ values obtained from Q bias and TD-error are similar.

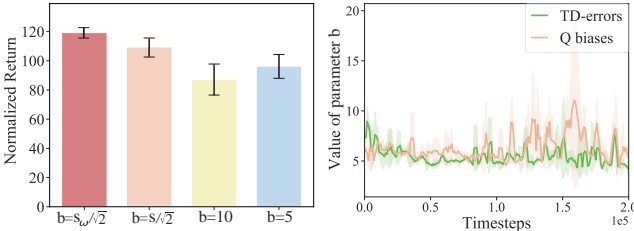

Figure 17: (a) Normalized returns with different values of parameter $b$. (b) Curves of the estimated value of $b$ during online fine-tuning using TD-errors and Q biases respectively. These experiments are conducted in the walker2d-medium task.

### C.4.5 ABLATION FOR ASYMMETRIC DISTRIBUTION

**Reparameterized Monte Carlo estimation of the KL divergence between asymmetric Laplace distributions.** We model the Q-value bias noise using an asymmetric Laplace distribution with location parameter $\mu \in \mathbb{R}$ and asymmetric scale parameters $b_1 > 0$ and $b_2 > 0$. Its probability density function is given by

$$f(x \mid \mu, b_1, b_2) = \begin{cases} \dfrac{1}{b_1 + b_2} \exp\left(\dfrac{x - \mu}{b_2}\right), & x < \mu, \\ \dfrac{1}{b_1 + b_2} \exp\left(-\dfrac{x - \mu}{b_1}\right), & x \geq \mu. \end{cases} \tag{40}$$

The corresponding cumulative distribution function (CDF) is

$$F(x \mid \mu, b_1, b_2) = \begin{cases} \dfrac{b_2}{b_1 + b_2} \exp\left(\dfrac{x - \mu}{b_2}\right), & x < \mu, \\ 1 - \dfrac{b_1}{b_1 + b_2} \exp\left(-\dfrac{x - \mu}{b_1}\right), & x \geq \mu, \end{cases} \tag{41}$$

which admits a closed-form inverse. For $u \in (0, 1)$, the inverse CDF $F^{-1}$ is

$$
x = F^{-1}(u \mid \mu, b_1, b_2) =
\begin{cases}
\mu + b_2 \log\left( u \, \dfrac{b_1 + b_2}{b_2} \right), & u < \dfrac{b_2}{b_1 + b_2}, \\
\mu - b_1 \log\left( (1 - u) \, \dfrac{b_1 + b_2}{b_1} \right), & u \geq \dfrac{b_2}{b_1 + b_2}.
\end{cases}
\tag{42}
$$

Let $q(x) = \mathrm{AL}(\mu_q, b_1, b_2)$ and $p(x) = \mathrm{AL}(\mu_p, b_1, b_2)$ denote two asymmetric Laplace distributions. To obtain a gradient estimator that is compatible with backpropagation, we apply the reparameterization trick. Instead of sampling $x \sim q$ directly, we first draw $u \sim \mathrm{Uniform}(0, 1)$ and then map $u$ to $x$ via the differentiable transformation equation 42:

$$
u \sim \mathcal{U}(0, 1), \qquad x = T(u; \mu_q, b_1, b_2) = F^{-1}(u \mid \mu_q, b_1, b_2).
\tag{43}
$$

Under this reparameterization, the KL divergence can be written as an expectation with respect to the fixed base distribution $\mathcal{U}(0, 1)$:

$$
\mathrm{KL}(q \,\|\, p) = \mathbb{E}_{u \sim \mathcal{U}(0,1)} \big[ \log q\big(T(u; \mu_q, b_1, b_2)\big) - \log p\big(T(u; \mu_q, b_1, b_2)\big) \big].
\tag{44}
$$

In practice, we approximate equation 44 by Monte Carlo sampling. Given $S$ i.i.d. samples $u^{(s)} \sim \mathcal{U}(0, 1)$, we compute

$$
\widehat{\mathrm{KL}}(q \,\|\, p) = \frac{1}{S} \sum_{s=1}^{S} \left[ \log q\big(x^{(s)}\big) - \log p\big(x^{(s)}\big) \right], \qquad x^{(s)} = T\big(u^{(s)}; \mu_q, b_1, b_2\big).
\tag{45}
$$

Since the transformation $T(\cdot; \mu_q, b_1, b_2)$ is differentiable with respect to $\mu_q$ (and, if required, with respect to $b_1$ and $b_2$), the estimator $\widehat{\mathrm{KL}}(q \,\|\, p)$ defines a valid pathwise gradient estimator. This enables end-to-end optimization of the parameters of $q$ (e.g., Q-value predictions) using standard gradient-based methods, while retaining a flexible asymmetric Laplace likelihood model.

**We conduct experiments to compare asymmetric and symmetric Laplace modeling for the Q-bias.** Consistent with the training procedure described in the main text, we assume that $\varepsilon_\theta$ and $\varepsilon_{\hat\theta}$ follow an asymmetric Laplace distribution with shared scale parameters $b_1$ and $b_2$. We estimate $b_1$ and $b_2$ for each training batch via maximum likelihood estimation, and compute the Kullback–Leibler divergence $D_{\mathrm{KL}}$ using the Monte Carlo and reparameterization techniques introduced above. All other components of the model and hyperparameters are kept identical to those settings in LAROO .

We report experimental results on several MuJoCo environments. The results in Table 14 indicate that, due to the additional computational complexity and approximation errors, the asymmetric Laplace model yields worse performance compared to LAROO.

Table 14: Comparison with asymmetric Laplace modeling and symmetric Laplace modeling.

| Task | Type | Laplace modeling (LAROO) | asymmetric Laplace modeling |
|---|---|---|---|
| Walker2d | medium | **120.4 ± 2.9** | 36.2 ± 13.3 |
| | medium-expert | **126.7 ± 1.0** | 39.1 ± 14.0 |
| halfcheetah | medium | **92.5 ± 1.2** | 83.3 ± 4.8 |
| | medium-expert | **99.5 ± 1.8** | 92.9 ± 1.6 |
| hopper | medium | **106.7 ± 2.6** | 83.3 ± 4.8 |
| | medium-expert | **112.3 ± 0.7** | 92.9 ± 1.6 |

## C.5 THE USE OF LARGE LANGUAGE MODELS

In our study, we use the large language models only for polishing sentences and checking the grammatical errors.

# D  THE Q BIAS IN MORE RL DOMAIN

## D.1  THE Q BIAS OF SPARSE REWARD DOMAINS IN O2O RL

In this subsection, we supplement more distribution of Q bias to test whether the heavy-tailed Q bias exists across O2O RL settings. We test several existing O2O RL methods in sparse reward environments (Antmaze) and semi-sparse reward environments (Adroit). The experiment settings keep the same as those in Appendix C.

We present the Q bias distributions of Antmaze tasks in Figure 18 and these of Adroit tasks in Figure 19. We observe that the Q bias is significantly large in Adroit tasks, because the Q-value network struggles to learn from sparse reward and accurately estimate each state-action pair (Chen et al., 2022).

**The experimental results validate that the heavy-tailed issue of Q bias generally exists across the O2O RL domains**. We have discussed the reasons that may contribute to the heavy-tailed bias in O2O RL settings in Appendix A.3.

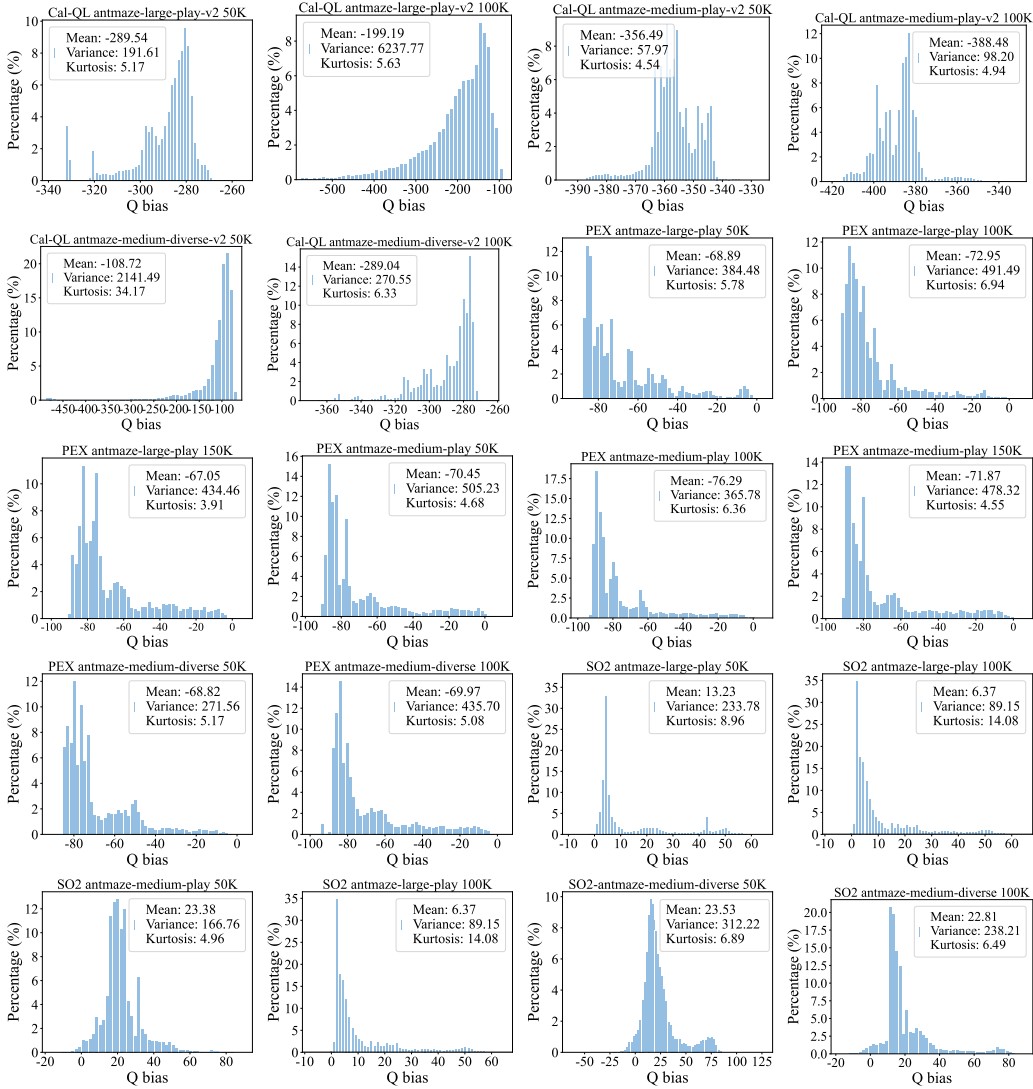

Figure 18: The heavy-tailed distribution of Q bias in sparse reward domain. We test three O2O RL methods (SO2 (Zhang et al., 2024), Cal-QL (Nakamoto et al., 2023), PEX (Zhang et al., 2023)) in Antmaze tasks (large-play, medium-play and medium-diverse).

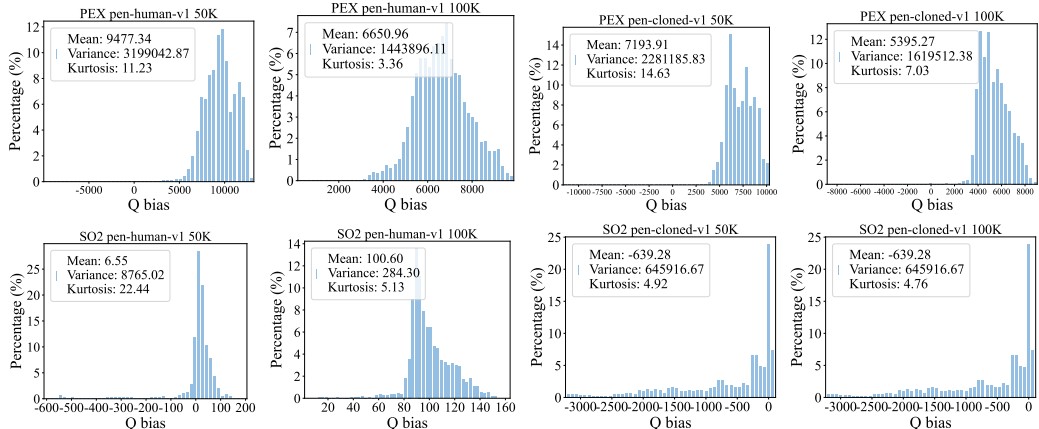

Figure 19: The heavy-tailed distribution of Q bias in semi-sparse reward domain. We test SO2 (Zhang et al., 2024) and PEX (Zhang et al., 2023)) in Adroit tasks (pen-human, pen-cloned).

## D.2 THE Q BIAS IN ONLINE RL

In this subsection, we test the distribution of Q-bias in the O2O RL setting. We conduct these tests to gain a deeper understanding of the Q bias distribution in O2O RL. Our research remains primarily focused on O2O settings.

We adopt two effective, widely used online algorithms SAC (Haarnoja et al., 2018) and TD3 (Fujimoto et al., 2018), across several MuJoCo tasks. We choose four fixed checkpoints: 100K, 300K, 600K and 1M (million) steps to better understand the evolving distribution of Q bias during training. The resulting distributions are shown in Figure 20.

In early training stages (100K), due to the poor estimation ability of Q-value network, Q bias exhibits complex and irregular distributional characteristics, including multi-peak and dispersion, making it difficult to fit with a specific distribution. During training (300K, 600K), we also observed the occurrence of heavy-tailed Q-bias, mainly due to a systematic overestimation problem. In later stages (1M), the Q-bias distribution gradually became concentrated and Gaussian-like, which is reasonable according to the convergence theorem of Q-functions.

In summary, in pure online RL, training data are collected on-policy and evolve more smoothly with the policy, so such sharp non-uniform shifts are less severe, and heavy-tailed Q-bias is not consistently sustained.

## D.3 THE Q BIAS IN OFFLINE RL

In this subsection, we examine the distribution of Q-bias in the offline RL setting. We adopt the effective offline algorithms—TD3BC (Fujimoto & Gu, 2021), CQL (Kumar et al., 2020) and IQL (Kostrikov et al., 2022)—across several MuJoCo tasks. We also choose four fixed checkpoints: 100K, 300K, 600K and 1M (million) steps to better observe the Q bias in offline RL. To obtain ground-truth Q-values for state–action pairs in the offline datasets, we initialize each pair as the starting point, perform a full-episode rollout, and estimate its true Q-value via Monte Carlo evaluation. The corresponding Q-bias distributions are presented in Figure 21.

Experimental results demonstrate that Q bias also show heavy-tailed behavior in offline RL, exhibiting negative skewness with notably high biases in the left tail (i.e., large underestimation). This phenomenon is reasonable because these offline methods usually employ a conservative Q-learning approach.

To assess whether LAROO remains effective in offline settings, we integrated the Laplace noise modeling into several offline RL algorithms and evaluated its impact on performance. The experimental results in Table 15 show only modest improvements. This is primarily because offline methods rely on strictly conservative Q-estimation strategies, which make it difficult for LAROO to correct extremely low Q bias.

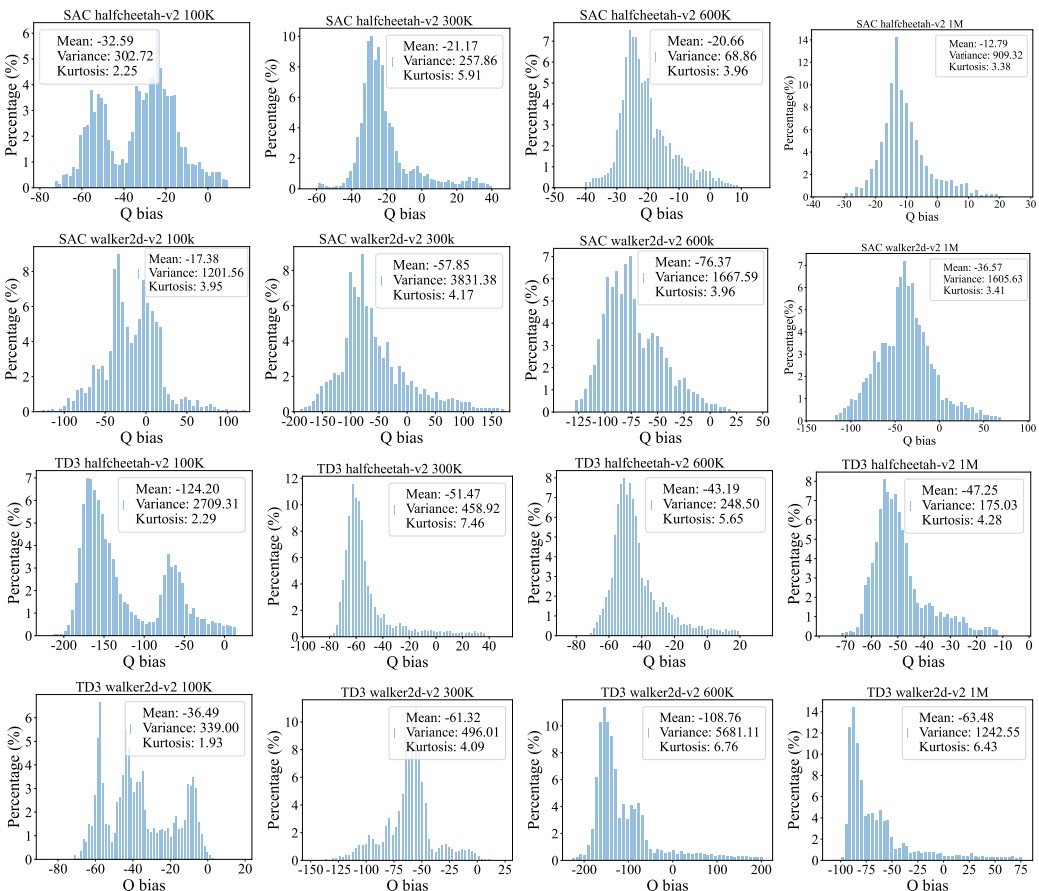

Figure 20: The distribution of Q bias during online training in pure online RL settings. We test SAC (Haarnoja et al., 2018) and TD3 (Fujimoto et al., 2018)) in MuJoCo tasks.

Table 15: Experimental results of integrating LAROO with several offline methods. Each result is averaged over three random seeds.

| Task | Type | TD3BC | TD3BC+LAROO | CQL | CQL+LAROO | IQL | IQL+LAROO |
|------|------|-------|-------------|-----|-----------|-----|-----------|
| Walker2d | medium | $82.7 \pm 4.5$ | $87.1 \pm 6.2$ | $80.7 \pm 3.3$ | $82.4 \pm 3.1$ | $80.9 \pm 3.2$ | $80.6 \pm 4.2$ |
| | medium-replay | $85.6 \pm 5.2$ | $89.5 \pm 3.2$ | $76.1 \pm 13.2$ | $88.2 \pm 4.9$ | $82.1 \pm 3.0$ | $79.8 \pm 2.0$ |
| | medium-expert | $110.3 \pm 1.0$ | $111.5 \pm 1.7$ | $109.5 \pm 0.4$ | $115.2 \pm 2.3$ | $111.7 \pm 0.9$ | $113.5 \pm 1.1$ |
| Halfcheetah | medium | $48.1 \pm 0.2$ | $55.1 \pm 0.8$ | $47.0 \pm 0.2$ | $49.4 \pm 3.1$ | $48.3 \pm 6.8$ | $51.0 \pm 4.2$ |
| | medium-replay | $44.8 \pm 0.6$ | $50.8 \pm 3.2$ | $47.1 \pm 1.7$ | $51.2 \pm 0.9$ | $44.5 \pm 0.8$ | $46.2 \pm 2.0$ |
| | medium-expert | $91.3 \pm 6.1$ | $95.9 \pm 3.7$ | $95.6 \pm 0.4$ | $99.1 \pm 2.3$ | $94.7 \pm 0.5$ | $96.3 \pm 1.1$ |

Table 16: Normalized returns of integrating LAROO with online RL methods at one million training step. Each result is averaged over three random seeds.

| Task | TD3 | TD3 + LAROO | SAC | SAC + LAROO |
|------|-----|-------------|-----|-------------|
| Walker2d-v2 | $89.3 \pm 1.3$ | $94.8 \pm 1.7$ | $114.5 \pm 2.2$ | $115.2 \pm 2.3$ |
| Halfcheetah-v2 | $54.3 \pm 1.7$ | $59.3 \pm 0.9$ | $70.1 \pm 4.2$ | $70.3 \pm 3.3$ |
| Hopper-v2 | $93.0 \pm 5.3$ | $99.5 \pm 1.4$ | $105.5 \pm 7.1$ | $110.2 \pm 3.5$ |

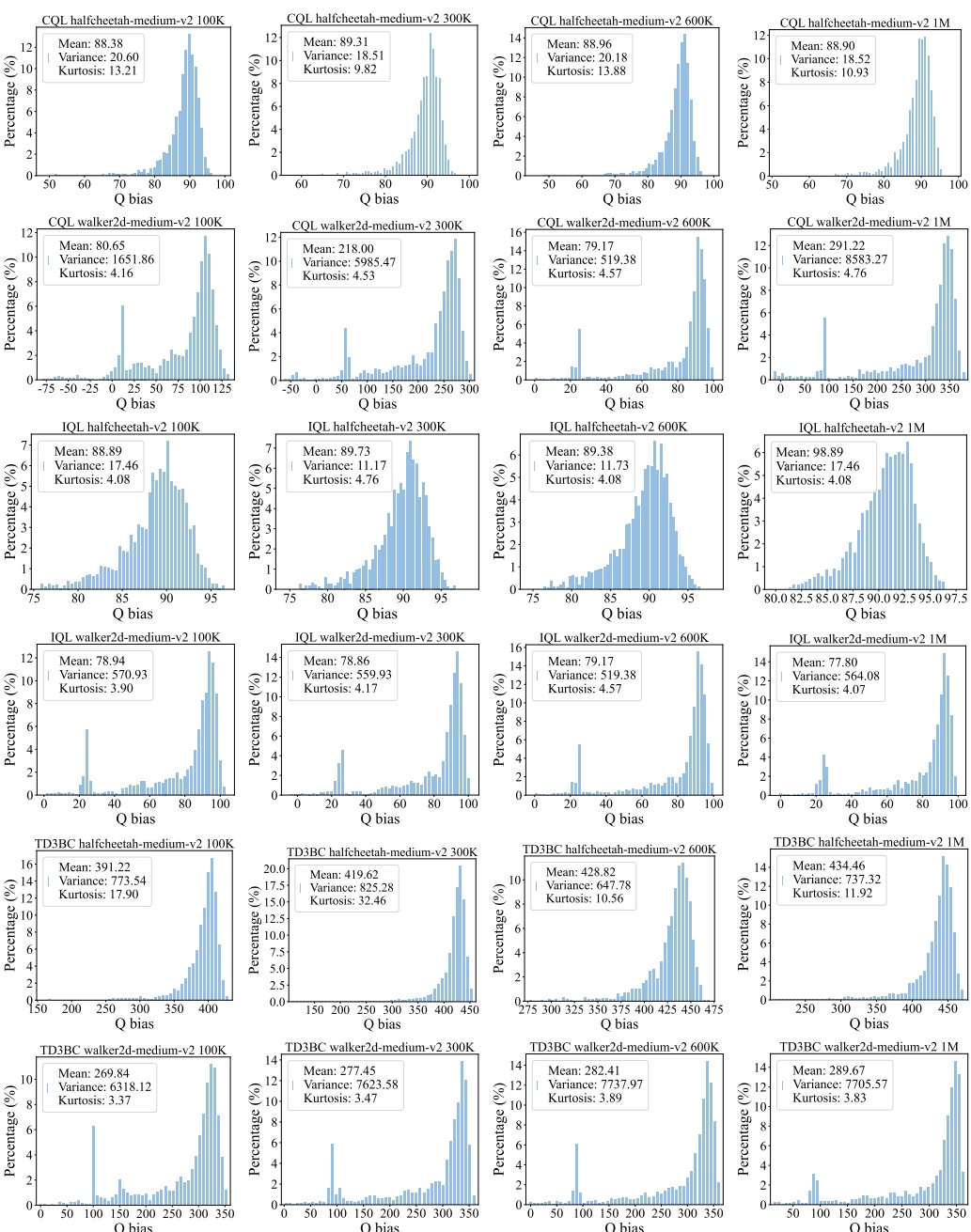

Figure 21: The distribution of Q bias during training in pure offline RL settings. We test CQL (Kumar et al., 2020), TD3BC (Fujimoto & Gu, 2021) and IQL (Kostrikov et al., 2022)) in MuJoCo tasks.

