# OpenReview forum: "Tackling Heavy-Tailed Q-Value Bias in Offline-to-Online Reinforcement Learning with Laplace-Robust Modeling"
_ICLR.cc/2026/Conference — ICLR 2026 Poster_

### Official Review · Reviewer_qzBg · 2025-10-23

**Soundness:** 3
**Presentation:** 3
**Contribution:** 3
**Rating:** 6
**Confidence:** 3

**Summary:**

This paper is the first to empirically reveal the phenomenon that Q-value estimation bias in the online fine-tuning process of O2O pervasively follows a heavy-tailed distribution.To address this, the authors propose the LAROO (Laplace-based robust offline-to-online RL) approach, designed to mitigate the heavy-tailedness of Q-value estimation bias, thereby improving training stability and performance. The approach models the Q-value estimation bias using an adaptive Laplace-distributed noise, based on which a robust value loss function is constructed to reduce the variance of the Q-value bias during the learning process. Concurrently, it incorporates ensemble Q-models to shift the mean of the bias towards zero, ultimately achieving more robust and stable value estimation.

**Strengths:**

- First revealed and empirically validated the pervasive heavy-tailed Q-bias phenomenon.
- Through rigorous theoretical derivation, the LAROO method constructs a robust loss function based on the Laplace distribution, which is insensitive to outliers and reduces the variance of Q-bias.
- Extensive experimental results demonstrate superior performance over published O2O methods across multiple environments.

**Weaknesses:**

1.  The paper lacks comprehensive ablation studies validating the specific contribution of the noise model to training stability.
2.  The contribution of the non-novel ensemble model towards bias correction and final performance improvement is difficult to disentangle.

**Questions:**

1.  As noted in Weakness 1, training stability is the primary effect of the noise model, but no training curve plots or related metrics are provided in the ablation experiments.
2.  The non-novel ensemble method significantly reduces the bias mean, and comparing Figures 13(e) and 13(f)  suggests it also helps reduce variance and kurtosis. As noted in Weakness 2, this raises the question whether the ensemble model might be more critical to performance improvement than the noise model.Could the authors provide ablation results across more environments for further analysis?
3.  What is the experimental setup for LAROO w/o ensemble in Table 12, and why do the results differ from those in Table 10, where N = 1 and UTD = 1?

---

> ### Author Response · Authors · 2025-11-22
>
> We appreciate the reviewer's insightful and constructive comments and suggestions. We respond to each comment as follows and accordingly revise our manuscript, with all updates highlighted in *blue* for your convenience. We sincerely hope that our responses could properly address your concerns. If so, we would deeply appreciate it if you could raise your score. If not, please let us know your further concerns, and we will continue actively responding to the comments and enhancing our submission.
>
>
>
> **Q1**. The paper lacks comprehensive ablation studies validating the specific contribution of the noise model to training stability. As noted in Weakness 1, training stability is the primary effect of the noise model, but no training curve plots or related metrics are provided in the ablation experiments.
>
> **A1**. Thanks for your valuable and insightful comments.
>
> - We supplement the training curves and compute the related metrics of training stability (i.e., variance and NCD of results), comparing the stability of LAROO without and with noise model. The NCD values are reported below, where lower values indicate greater training stability, and the training curves are provided in Figure 16. More detailed results can be found in Appendix C.4.3.
> - The experimental results show that removing the noise model substantially reduces training stability with larger NCD values. The training curves exhibit stronger oscillations, and the variance of the final performance increases. This effect is particularly obvious in the `medium-expert `task, where the distribution shift between offline and online data is larger than that in other tasks.
>
> | NCD values             | LAROO w/o ensemble | LAROO w/o noise model | LAROO |
> | ---------------------- | ------------------ | --------------------- | ----- |
> | Walker2d-medium        | 0.023              | **0.024**             | 0.020 |
> | Walker2d-medium-replay | **0.015**          | 0.014                 | 0.013 |
> | Walker2d-medium-expert | 0.034              | **0.038**             | 0.031 |
> | Hopper-medium          | 0.017              | **0.018**             | 0.016 |
> | Hopper-medium-replay   | 0.014              | **0.017**             | 0.014 |
> | Hopper-medium-expert   | 0.034              | **0.036**             | 0.029 |

---

> > ### Author Response · Authors · 2025-11-22
> >
> > **Q2**. The contribution of the non-novel ensemble model towards bias correction and final performance improvement is difficult to disentangle. In detail, the non-novel ensemble method significantly reduces the bias mean, and comparing Figures 13(e) and 13(f) suggests it also helps reduce variance and kurtosis. As noted in Weakness 2, this raises the question of whether the ensemble model might be more critical to performance improvement than the noise model. Could the authors provide ablation results across more environments for further analysis?
> >
> > **A2**. Thanks for your kind and insightful comments.
> >
> > - We conduct additional ablation comperiments to compare the contribution of ensemble-model with noise model. Specifically, we evaluate two settings: one without Laplace noise modeling and one without the ensemble model. We assess both training stability and final performance by plotting training curves and computing the corresponding stability metrics. Please refer to Appendix C.4.3 for detailed ablation studies.
> >
> > - The supplementary results show that the noise model plays a more critical role than the ensemble model in stability and performance.
> >
> >   - For bias correction, as shown in Figure 14 (a) and (d), the results show that the ensemble model struggles to handle the issue of heavy-tailed Q bias. Meanwhile, the noise model can reduce variance and kurtosis more than ensemble models.
> >
> >   - As shown in Figure 16 and Table 12,  `LAROO w/o noise model` performs lower than ` LAROO w/o ensemble model` in 8 out of 9 MuJoCo tasks, which indicates that the noise model is more important for improving final performance. The lowest value is highlighted in bold in Table 12 below.
> >
> >   - The original Figure 13 has been updated to Figure 14.
> >
> >   - | Returns (Table 12)        | LAROO w/o ensemble | LAROO w/o noise model | LAROO           |
> >     | :------------------------ | ------------------ | --------------------- | --------------- |
> >     | Walker2d-medium           | 106.2 $\pm$ 2.9    | **104.7 $\pm$ 1.7**   | 120.4 $\pm$ 1.9 |
> >     | Walker2d-medium-replay    | 105.0 $\pm$ 3.8    | **88.3 $\pm$ 10.1**   | 124.8 $\pm$ 2.0 |
> >     | Walker2d-medium-expert    | 111.8 $\pm$ 6.3    | **105.4 $\pm$ 3.7**   | 126.7 $\pm$ 1.0 |
> >     | Halfcheetah-medium        | 89.5 $\pm$ 2.1     | **87.7 $\pm$ 1.2**    | 92.5 $\pm$ 1.2  |
> >     | Halfcheetah-medium-replay | **81.9 $\pm$ 2.6** | 83.9 $\pm$ 2.8        | 84.7 $\pm$ 1.9  |
> >     | Halfcheetah-medium-expert | 92.4 $\pm$ 3.1     | **90.3 $\pm$ 4.1**    | 99.5 $\pm$ 1.8  |
> >     | Hopper-medium             | 98.3 $\pm$ 6.3     | **93.6 $\pm$ 7.4**    | 106.7 $\pm$ 2.6 |
> >     | Hopper-medium-replay      | 105.6 $\pm$ 1.2    | **90.9 $\pm$ 1.3**    | 109.5 $\pm$ 1.8 |
> >     | Hopper-medium-expert      | 108.3 $\pm$ 3.2    | **99.9 $\pm$ 17.6**   | 112.3 $\pm$ 0.7 |
> >
> >
> > **Q3**. What is the experimental setup for LAROO w/o ensemble in Table 12, and why do the results differ from those in Table 10, where N = 1 and UTD = 1?
> >
> > **A3**. We appreciate your careful and kind comments. The setups differ as follows:
> >   - In LAROO w/o ensemble, we remove the ensemble model, retain the noise model and set the hyperparameters listed in Table 3, including an update-to-data ratio (UTD) of five. A high UTD improves training efficiency and is widely used by O2O RL baselines (e.g., ENOTO, BOORL), so we follow the same setting.
> >   - Meanwhile, the strong baseline Cal-QL does not use high UTD. For a fair comparison with Cal-QL, Table 10 reports results under N=1 and UTD=1, where we only employ the Laplace noise modeling without ensemble or high UTD.  Even under this setting, LAROO outperforms Cal-QL in 10 out of 13 tasks, indicating that the performance gains of LAROO are not driven solely by ensembles or a high UTD.
> >   - Consequently, because Table 12 and Table 10 use different UTD ratios, their results are naturally different.

---

> > > ### Comment · Reviewer_qzBg · 2025-11-26
> > >
> > > I would like to thank the authors for their detailed response.
> > > The provided ablation studies regarding training stability (specifically the NCD metrics and training curves) have effectively addressed my concerns regarding the specific contribution of the noise model. Therefore, I will maintain my score.

---

> > > ### Author Response · Authors · 2025-12-02
> > >
> > > To better quantify the contribution of the Laplace noise model to training stability, we validate that it reduces variance by **76.1%** in Table 12  and mitigates performance drop (i.e., NCD) by **18.9%** across six `Hopper` and `Walker2d` tasks.

---

### Official Review · Reviewer_SoSF · 2025-10-28

**Soundness:** 3
**Presentation:** 3
**Contribution:** 2
**Rating:** 4
**Confidence:** 4

**Summary:**

This paper reveals that in offline-to-online RL, the Q-value biases follow a heavy-tailed distribution. To address this, this paper proposes LAROO, which introduces Laplace -based loss function to replace L2 loss for Q update, and uses ensemble models for Q target estimation to reduce the estimation bias. Theoretical analysis is provided to show that LAROO exhibits a smaller estimation bias than typical L2-based Q updates. Experimental results demonstrate that LAROO outperforms previous baselines on D4RL datasets.

**Strengths:**

**Clarity.** Generally, this paper is well-written and easy to follow. The motivation and observations are clear.

**Novelty.** I think the key finding that the Q bias follows a heavy-tailed distribution is interesting and the method that levearges Laplace-based noise model is reasonable. Also, the authors provide clear theoretical analysis on why the proposed method could reduce the Q bias.

**Significance.** The experimental results show that LAROO outperforms the previous baselines.

**Weaknesses:**

There are some points that need further clarifications.
- The motivation that minimizing the KL divergence between $p(\\mathcal{Q}|\\mathcal{T}Q _ \\theta)$ and $q(\\mathcal{Q}|Q _ \\theta)$ in Line 225 is not clear. It seems just for the derivation of $D _ b(x)$. I wonder why minimizing such KL divergence could deal with the heavy-tailed Q bias issue and why it is reasonable. Could the authors give more clarifications on it?
- In Line 277, the authors use TD-error as a surrogate for Laplace-based Q bias. That is to say, the TD-error is also assumed to follow a prior Laplace distribution. Then according to MLE, we could directly use L1 loss for Q update, then what is the advantage of using Equation (6)? Since Equation (6) is derived by minimizing the KL divergence, this is also related to the previous issue.
- The heavy-tailed Q bias issue is not first observed in offline RL. Robust-IQL[1] also observes the heavy-tailed Q bias issue and addresses it with Huber loss, which is mroe easy to implement. This work uses Laplace distribution instead, could you demonstrate whether Huber loss is not used for your work?
- LAROO seems not designed specifically for the offline-to-online setting, since Laplace-based Q update and ensemble models could also be applied to pure online setting. I wonder if the heavy-tailed Q bias issue also exists in pure online settings when using standard online RL algorithms like TD3 or SAC. If it does, what is the specific advantage of LAROO that makes it suitable for offline-to-online RL? If it does not, why does this issue manifest in the offline-to-online setting but not in pure online RL?
-  (minor) In line 288, 'They' -> 'It'.

[1] Towards Robust Offline RL Under Diverse Data Corruption. ICLR 2024

**Questions:**

Please refer to the weaknesses to address the concerns. I will check the authors' responses to decide whether to revise the rating.

---

> ### Author Response · Authors · 2025-11-22
>
> We appreciate the reviewer's insightful and constructive comments and suggestions. We respond to each comment as follows and accordingly revise our manuscript, with all updates highlighted in *blue* for your convenience. We sincerely hope that our responses could properly address your concerns. If so, we would deeply appreciate it if you could raise your score. If not, please let us know your further concerns, and we will continue actively responding to the comments and enhancing our submission.
>
> **Q1**. The motivation that minimizing the KL divergence between $p(\mathcal{Q}(s,a)\mid Q_\theta(s,a))$ and $p(\mathcal{Q}(s,a)\mid \mathcal{T}Q_\theta(s,a))$ in Line 225 is not clear. It seems just for the derivation Q-value loss function. I wonder why minimizing such KL divergence could deal with the heavy-tailed Q bias issue and why it is reasonable. Could the authors give more clarifications on it?
>
> 1. What  is $p(\mathcal{Q}(s,a)\mid Q\_\theta(s,a))$?
> 2. What is $p(\mathcal{Q}(s,a)\mid \mathcal{T}Q\_\theta(s,a))$？
> 3. why $\min D_{KL} (p(\mathcal{Q}(s,a)\mid Q_\theta(s,a))||p(\mathcal{Q}(s,a)\mid \mathcal{T}Q_\theta(s,a)))$？
>
> **A1**. Thanks for your valuable and highly insightful comments. We will clarify our motivation with Bayesian principles for easy understanding, and the meaning of (1)-(3).
>
> - **a.** the  $p(\mathcal{Q}(s,a)\mid \mathcal{T}Q_\theta(s,a))$ is the posterior given the bellman target.
>
>   **b.** the $p(\mathcal{Q}(s,a)\mid Q_\theta(s,a))$ is the posterior given the estimated Q-values.
>
>     **c.** Minimizing the KL divergence encourages the posterior  $p(\mathcal{Q}(s,a)\mid Q_\theta(s,a))$ to approximate the posterior  $p(\mathcal{Q}(s,a)\mid \mathcal{T}Q_\theta(s,a))$. Since the posterior  $p(\mathcal{Q}(s,a)\mid \mathcal{T}Q_\theta(s,a))$ incorporates true environmental return information (i.e., future rewards), it serves as a closer approximation to the Bayes-optimal posterior (i.e., $ p(\mathcal{Q}\mid \text{all environmental evidence} )$ ) , compared with the posterior $p(\mathcal{Q}(s,a)\mid Q_\theta(s,a))$.
>
> - Relationship between Bayesian posterior and likelihood. When using non-informative priors, the posterior distribution in Bayesian inference is numerically equivalent to the likelihood function in frequentist statistics [1]. In ours, the prior $p(\mathcal{Q}(s,a))$ can be non-informative because the true Q-value is always unknown. Therefore,  the Bellman-induced posterior  $p(\mathcal{Q}(s,a)\mid \mathcal{T}Q_\theta(s,a))$ is the same as the concept of "the likelihood of true Q-value" in our manuscript (Line 225).
>
>
>
> **Q2**. Following **Q1**,  why minimizing such KL divergence could deal with the heavy-tailed Q bias issue?
>
> **A2**. Thanks for your insightful questions.  Building on **A1**, minimizing this KL divergence is effective, because the Laplace likelihood induces a robust learning process for Q-network under heavy-tailed noise:
>
> - Specifically, $D\_{\text{KL}}(p\|q) = \mathbb{E}\_{p}[\log p - \log q]$, minimizing this KL divergence is equivalent to minimizing the expected negative log-likelihood $\mathbb{E}\_{p}[-\log q(\mathcal{Q}(s,a)\mid Q\_\theta(s,a))]$ under the posterior $p$. When using a Laplace likelihood $q(\mathcal{Q}(s,a)\mid Q\_\theta(s,a))$, the resulting negative log-likelihood is proportional to $|\mathcal{Q}(s,a)- Q_\theta(s,a)|$ , which grows linearly rather than quadratically with the Q bias, it produces a robust traininng process with two key effects:
>   - **Outliers are down-weighted**: extremely large Q biases no longer dominate the KL objective, so they cannot cause unstable or aggressive updates of Q-network.
>   - **Optimization is driven by the central mass of Q bias**: the optimization of KL divergence focuses on fitting the bulk of Q-values where most samples lie, rather that being governed by rare, extreme Q biases in the tail.
> - Consequently, by choosing a Laplace likelihood inside the KL divergence, the training process naturally penalizes heavy-tailed noise in Q-value estimates. This is reflected in the final loss function, which down-weights large Q-value losses and yields bounded gradients.

---

> ### Author Response · Authors · 2025-11-22
>
> **Q3**. (Following Q1, Q2). In Line 277, the authors use TD-error as a surrogate for Laplace-based Q bias. That is to say, the TD-error is also assumed to follow a prior Laplace distribution. Then according to MLE, we could directly use L1 loss for Q update, then what is the advantage of using Equation (6)? Since Equation (6) is derived by minimizing the KL divergence, this is also related to the previous issue.
>
> **A3**. Thanks for your valuable and insightful commments.  We clarify (1) the role of TD-error as a surrogate, and (2) the advantage of using Equation (6) compared with L1 loss.
>
> - (1) On the surrogate use of TD-error. We apologize for the earlier confusion caused. On Line 277, we use the TD-error as a surrogate only to approximate the variance of Q-bias, so that we can adaptively update the Laplace noise modeling. We do not introduce additional assumptions about TD-errors.
>
>   - This surrogate is reasonable for variance approximation, because:
>
>   - $$\mathcal{T}Q_{\hat \theta}(s,a) - Q_{\theta}(s,a) =  \big(\mathcal{Q}(s,a)- \varepsilon_{\hat \theta}\big) - \big(\mathcal{Q}(s,a)- \varepsilon_{\theta}\big) = \varepsilon_{\theta} - \varepsilon_{\hat \theta}$$
>
>     Given the independence of $\varepsilon_{\theta}$ and $\varepsilon_{\hat \theta}$, the variance of TD-errors twice the variance of Q bias.
>
> - (2) Because of Response (1), Eq. (6) is different from the MLE with L1 loss. Moreover, we provide a brief analysis of the advantage of Eq. (6): it better preserves the distributional information of Q bias than the L1 loss.
>
>   - In Eq.(4), the likelihood  $p(\mathcal{Q}(s,a)\mid \mathcal{T}Q_{\hat \theta}(s,a))$ explicitly captures the distributional information of heavy-tailed bias in bellman target with Laplace modelings for Q updates.
>
>   - Instead, if we ignore the information of estimation bias on the bellman target, i.e., the  $p(\mathcal{Q}(s,a)\mid \mathcal{T}Q_{\hat \theta}(s,a))$ is a Dirac delta function centered on $\mathcal{T}Q_{\hat \theta}(s,a)$ , the KL divergence reduces to the negative log likelihood  $-q(\mathcal{T}Q_{\hat \theta}(s,a)|Q_\theta(s,a))$, which is exactly equivalent to applying the L1 loss under a Laplace assumption. The proof is as follows:
>
>     - $$ D_{KL}(p(x)\|q(x)) =H(p(x),q(x))-H(p(x)) =-\int_{-\infty}^{\infty}p(x)\log q(x)dx+\int_{-\infty}^{\infty}p(x)\log p(x)dx $$
>
>       Where the entropy $H(p(x))$  is constant. In our case, if the  $p(\mathcal{Q}(s,a)\mid \mathcal{T}Q_{\hat \theta}(s,a))$ is a Dirac delta function centered on $\mathcal{T}Q_{\hat \theta}(s,a)$, the KL divergence becomes:
>
>       $$-\int_{-\infty}^{\infty}p(\mathcal{Q}(s,a)\mid \mathcal{T}Q_{\hat \theta}(s,a))\log q(\mathcal{Q}(s,a)|Q_\theta(s,a)) d\mathcal{Q}(s,a)= -\log q(\mathcal{T}Q_{\hat \theta}(s,a)|Q_\theta(s,a))$$
>
> - In summary, under heavy-tailed Q bias, it is a natural and reasonable choice to model the Q-bias for both estimated Q-values and bellman targets, and derive the Equation (6) for updates. In contrast, directly using L1 loss for updates may lose the distributional information of bias in Bellman targets.
>
> - We compare the LAROO with L1 loss function. The experimental results show L1 loss function is also better than the MSE loss in O2O setting, but its performance is lower than LAROO. Please refer to Appendix C.4.1 for details.
>
> | Normalized Returns        | L1              | Huber loss          | LAROO               |
> | ------------------------- | --------------- | ------------------- | ------------------- |
> | Walker2d-medium           | 85.2 $\pm$ 11.6 | 110.7 $\pm$ 5.7     | **120.4 $\pm$ 2.9** |
> | Walker2d-medium-replay    | 118.1 $\pm$ 0.7 | 110.4 $\pm$ 0.9     | **124.8 $\pm$ 2.0** |
> | Walker2d-medium-expert    | 120.5 $\pm$ 0.3 | 110.4 $\pm$ 10.5    | **126.7 $\pm$ 1.0** |
> | Halfcheetah-medium        | 90.2 $\pm$ 1.2  | 91.5 $\pm$ 4.2      | **92.5 $\pm$ 1.2**  |
> | Halfcheetah-medium-replay | 78.1 $\pm$ 2.9  | 81.5 $\pm$ 1.7      | **84.7 $\pm$ 1.9**  |
> | Halfcheetah-medium-expert | 92.5 $\pm$ 3.2  | **100.7 $\pm$ 5.3** | 99.5 $\pm$ 1.8      |
> | Hopper-medium             | 89.3 $\pm$ 16.9 | 90.4 $\pm$ 15.2     | **106.3 $\pm$ 2.6** |
> | Hopper-medium-replay      | 100.2 $\pm$ 2.6 | 102.2 $\pm$ 13.0    | **109.5 $\pm$ 1.8** |
> | Hopper-medium-expert      | 101.2 $\pm$ 5.2 | 98.4 $\pm$ 13.7     | **112.3 $\pm$ 0.7** |

---

> > ### Author Response · Authors · 2025-11-22
> >
> > **Q4**. The heavy-tailed Q bias issue is not first observed in offline RL. Robust-IQL [2] also observes the heavy-tailed Q bias issue and addresses it with Huber loss, which is easier to implement. This work uses Laplace distribution instead, could you demonstrate whether Huber loss is not used for your work?
> >
> > **A4**. Thanks for your kind and thoughtful questions. We do not adopt the Huber loss in our method, mainly because LAROO differs from Huber-based robust learning both in the underlying motivation and in the loss formulation.
> >
> >   - The key differences between LAROO and Huber loss are mainly two aspects:
> >     - In underlying motivation: following **A2**, **A3**, directly using L1 or Huber loss corresponds to applying a fixed robust regression objective to TD errors, which does not explicitly model the heavy-tailed Q bias in bellman target and may discard distributional information of Q bias. In contrast, LAROO better preserves the Q-bias information with a parameterized Laplace distribution.
> >     - In loss formulation: LAROO adaptively updates the Laplace noise parameters with MBBE. When heavy-tailed Q biases become more frequent, the learned Laplace scale parameter increases accordingly, and the gradient bound of Eq.(6) adaptively tightens, as discussed in Line 254-258.
> >       By contrast, Robust-IQL employs a Huber loss with non-adaptive gradient bounds and exhibits limited ability to adjust to changing degrees of heavy-tailedness.
> >   - In our ablation experiments, we compared the performance of Huber loss and LAROO. The experiments show LAROO performs better than the Huber loss. You can refer to Section 4.1 in Appendix for details.
> >
> >
> >
> >
> > **Q5**. LAROO seems not designed specifically for the offline-to-online setting, since Laplace-based Q update and ensemble models could also be applied to pure online setting. I wonder if the heavy-tailed Q bias issue also exists in pure online settings when using standard online RL algorithms like TD3 or SAC.
> >
> > **A5**. Thanks for your insightful comments. We examine the Q bias of SAC and TD3 in three MuJoCo tasks (`Halfcheetah`, `Walker2d` and `Hopper`). We set four checkpoint steps (100K, 300K, 600K and 1M) to fully observe the evolving distribution of Q bias during training. Please refer to Appendix D.2 for details.
> >
> >   - We summarize significant distributional patterns of Q bias at different training stages in online RL:
> >     - Early stage: due to the limited accuracy of the Q-value network at the beginning, Q bias exhibits complex and irregular distributional characteristics, such as multi-peak and high dispersion, making it difficult to fit with the Laplace distribution.
> >     - Middle stage: as training progresses, we observe the emergence of heavy-tailed Q bias mainly due to systematic overestimation, and the .
> >     - Final stage: the Q bias gradually becomes concentrated and Gaussian-like, which aligns with the expected convergence of Q-value estimates.

---

> > > ### Author Response · Authors · 2025-11-22
> > >
> > > **Q6**. Following Q5,
> > >
> > >  1. If it does, what is the specific advantage of LAROO that makes it suitable for offline-to-online RL?
> > >
> > >  2. If it does not, why does this issue manifest in the offline-to-online setting but not in pure online RL?
> > >
> > > **A6**.  Thanks for your insightful commments.  We address the two sub-questions as follows.
> > >
> > >   - (For 1.) LAROO is well suited to offline-to-online (O2O) RL, because Q-bias exhibits a stably heavy-tailed structure under large distribution shift between online and offline data, whereas this property of Q bias does not hold in online RL.
> > >     - In offline-to-online RL, the substantial offline–online shift causes the pretrained Q-network to exhibit persistently heavy-tailed estimation bias throughout the short fine-tuning horizon (typically 100K–200K steps). This relative stationarity of the Q-bias distribution makes Laplace-based noise modeling particularly well suited for capturing the heavy-tailed characteristics and correcting extreme biases.
> > >     - However, in online RL, the Q-bias evolves substantially across training stages and the heavy-tailed property is not consistently sustained. Especially in the early stage, it is highly irregular (often multimodal and dispersed), making it difficult to model accurately with a Laplace (or any single parametric) distribution.
> > >   - (For 2.) We explore why Q-bias exhibits heavy tails in offline-to-online (O2O) RL through experiments and attribute it to two factors:
> > >     -  (i) the Q-function with Max operator amplifies overestimation by selecting a large bellman target, and (ii) a non-uniform distribution shift between online and offline data. That is, pretrained Q-value networks have large Q bias in online samples that lie far from the offline data support, which manifest as the Q bias distribution's tails. In contrast, online samples closer to the offline distribution result in minor biases and constitute the central mass of Q bias distribution.
> > >     -  We validate the unique reason (ii) in offline-to-online settings through experiments. We compute the shifted distance of online data to offline distribution with two metrics, and validate a positive correlation between Q bias and shifted distance with Spearman's rank correlation coefficient. Please refer to Figure 7 and Appendix A.3 for details in the updated manuscript.
> > >     -  The reason (i) also contributes to heavy-tailed Q bias in pure online RL, as shown in Appendix D.2. Meanwhile, in pure online RL, training data are collected on-policy and evolve more smoothly with the policy, so such sharp non-uniform shifts (ii) are less severe, and heavy-tailed Q-bias is not consistently sustained.
> > >
> > > **Q7**. (minor) In line 288, 'They' -> 'It'.
> > >
> > > **A7**.  Thanks for your careful and kind comments, and we correct it in our manuscript.
> > >
> > >
> > >
> > > [1] Gelman et al. Bayesian data analysis. Chapman and Hall/CRC, 1995.
> > > [2] Rui Yang et al. Towards Robust Offline Reinforcement Learning under Diverse Data Corruption. ICLR 2024.

---

> > > ### Comment · Reviewer_SoSF · 2025-11-26
> > >
> > > Thank the authors for detailed response. Could I understand this way: L1 loss assumes the bellman error follows a Laplace distribution, while your method assume the Q value estimation error follows a Laplace distribution while the bellman error does not?

---

> > > > ### Author Response · Authors · 2025-11-26
> > > >
> > > > We sincerely appreciate the reviewer's insightful and constructive comments on Q-bias, which is a central focus of our study. We sincerely hope that our responses could properly address your concerns. If so, we would deeply appreciate it if you could raise your score. If not, please let us know your further concerns, and we will continue actively responding to the comments and enhancing our submission.
> > > >
> > > > **Q8** Could I understand this way: L1 loss assumes the bellman error follows a Laplace distribution, while your method assume the Q value estimation error follows a Laplace distribution while the bellman error does not?
> > > >
> > > > **A8** Thank you for your comments. We agree with your understanding. We summarize our assumption and its motivation below.
> > > > - In Assumptions 4.1 and 4.2, we assume the Q-value estimation bias follows the Laplace distribution. Importantly, the TD-error (Bellman error) is not Laplace-distributed under these Assumptions. More explicitly,
> > > > $$\text{TD-error}=\mathcal{T}Q_{\hat \theta}(s,a) - Q_{\theta}(s,a) = \big(\mathcal{Q}(s,a)- \varepsilon_{\hat \theta}\big) - \big(\mathcal{Q}(s,a)- \varepsilon_{\theta}\big) = \varepsilon_{\theta} - \varepsilon_{\hat \theta}$$
> > > > When the bias $\varepsilon_\{\theta}$ and $\varepsilon_\{\hat \theta}$ follow the Laplace distribution, the TD-error corresponds to the difference $\varepsilon_\{\theta} - \varepsilon_\{\hat \theta}$, which is not Laplace-distributed and can exhibit a more complex form than the Laplace.
> > > > - Given the heavy-tailed nature of Q bias, it is more natural and theoretically consistent to model the latent Q-bias itself with Laplace distribution for Q-value update, rather than the TD-error. We also explain the reasons and advantages of Eq. (6) over a direct L1 objective in **A3** and **A4**.

---

> > > > > ### Comment · Reviewer_SoSF · 2025-11-26
> > > > >
> > > > > Thank the authors for responding. The motivation of LAROO is now more clear. I encourage the authors to incorporate the above discussions into the revision, which would enhance the clarity. Correspondingly, I will raise my score.

---

> > > > > > ### Author Response · Authors · 2025-11-27
> > > > > > **Thanks for your support**
> > > > > >
> > > > > > Dear Reviewer SoSF,
> > > > > >
> > > > > > Thanks for your kind support and the further improvements you have suggested for our manuscript. We sincerely appreciate your insightful feedback, which has significantly helped us in ensuring a more comprehensive evaluation of our approach.
> > > > > >
> > > > > > Thank you again for your valuable comments and guidance.
> > > > > >
> > > > > > Best,
> > > > > >
> > > > > > Authors

---

### Official Review · Reviewer_U4Kt · 2025-10-31

**Soundness:** 3
**Presentation:** 2
**Contribution:** 3
**Rating:** 4
**Confidence:** 3

**Summary:**

This paper identifies and analyzes a previously unreported phenomenon in offline-to-online reinforcement learning (O2O RL): the Q-value estimation bias (Q-bias) tends to follow a heavy-tailed distribution during online fine-tuning. Such heavy-tailed behavior of Q-bias may introduce instability and impede effective performance improvement in fine-tuning. To address this, they propose Laplace-based Robust Offline-to-Online RL (LAROO), which models the Q-bias using a Laplace-distributed noise and introduces a robust policy evaluation loss derived from KL divergence minimization between Laplace distributions. LAROO further integrates an ensemble-based Q-value re-centering mechanism that shifts the Q-bias mean toward zero. Theoretical analysis shows that the proposed loss yields smaller single-step estimation bias than the widely-used L2 loss. Empirical results show improved stability and sample efficiency.

**Strengths:**

S1. (Empirical observation of heavy-tailed Q-bias)
In the MuJoCo domain, this work provides the empirical observation that Q-value biases in O2O RL exhibit heavy-tailed and positively skewed distributions during online fine-tuning (Figures 1, 4, 5). This is a significant observation beyond the mean and variance analysis. Moreover, the experiments (Figure 7) demonstrate that large-magnitude positive Q-biases may correlate with performance degradation.

S2. (Laplace-based policy evaluation)
The introduction of Laplace-distributed noise for modeling heavy-tailed bias is well-motivated. The derived loss function $D_b(x)$ effectively suppresses the influence of extreme outlier errors, serving as an empirically effective alternative to the L2 loss (standard Bellman loss).

S3. (Theoretical ground)
The theoretical analysis demonstrates that the proposed loss function reduces single-step estimation bias compared to the L2 loss, which plausibly contributes to the observed stability and consistent performance improvements.

S4. (Plug-in compatibility)
The proposed method can be used as a plug-in component for existing O2O methods. The experimental results show that replacing the Bellman loss in baselines with the proposed loss consistently improves their performance.

**Weaknesses:**

W1. (Questionable symmetry assumption of the Laplace model)
Although empirical results (Figures 1, 4, 5) show that Q-bias distributions are typically heavy-tailed and right-skewed (i.e., asymmetric positive bias), LAROO assumes a symmetric Laplace distribution. This modeling choice simplifies the formulation but may fail to accurately capture the empirical asymmetry. Indeed, Figure 8 indicates that LAROO reduces the positive tail but may over-correct by introducing spurious negative bias.

W2. (Limited justification for ensemble-based correction)
The proposed method integrates a random subset selection from ensemble Q-functions to re-center Q-bias, but it is somewhat heuristic. This paper lacks an intuitive explanation or theoretical reasoning as to why this mechanism effectively re-centers the bias.

W3. (Q-bias analysis across limited domains)
The analysis of Q-bias and its heavy-tailed behavior is restricted to dense-reward environments such as MuJoCo. It remains unclear whether similar heavy-tailed characteristics would emerge in sparse-reward domains (e.g., AntMaze) or semi-sparse domains (e.g., Adroit, OGBench[1]).

W4. (Theoretical and empirical disconnect)
The theoretical analysis establishes that the proposed loss reduces single-step estimation bias, but the connection to heavy-tailed variance reduction (the core empirical claim) remains indirect. It would strengthen the contribution to include a theoretical link between the Laplace modeling and variance-bounded Q estimation under heavy-tailed noise.


[1] Park, Seohong, et al. "Ogbench: Benchmarking offline goal-conditioned rl." arXiv preprint arXiv:2410.20092 (2024).

**Questions:**

Could you provide further clarification on the weaknesses? In particular, W1 and W3 will have the most significant impact on the overall rating.

---

> ### Author Response · Authors · 2025-11-22
>
> We appreciate the reviewer's insightful and constructive comments and suggestions. We respond to each comment as follows and accordingly revise our manuscript, with all updates highlighted in *blue* for your convenience. We sincerely hope that our responses could properly address your concerns. If so, we would deeply appreciate it if you could raise your score. If not, please let us know your further concerns, and we will continue actively responding to the comments and enhancing our submission.
>
>
>
> **Q1**. (Symmetry assumption of the Laplace model) Although empirical results (Figures 1, 4, 5) show that Q-bias distributions are typically heavy-tailed and right-skewed (i.e., asymmetric positive bias), LAROO assumes a symmetric Laplace distribution. This modeling choice simplifies the formulation but may fail to accurately capture the empirical asymmetry.
>
> **A1**. We appreciate the insightful and valuable questions regarding the symmetry assumption.
>
> - Asymmetric modeling provides enhanced flexibility but leads to significantly complex Q-value optimization.  In our methods, the Q-value update relies on a tractable KL divergence defined in Eq. (4). However,  the KL divergence between two asymmetric Laplace distributions with different parameters has no concise closed-form expression and requires Monte Carlo approximations or other numerical procedures, which are computationally expensive and introduce estimation errors [1] [2].
> - We empirically compare the symmetric Laplace model with asymmetric modeling for Q bias. Under  asymmetric Laplace modeling, we employ a reparameterized Monte Carlo estimation to compute the KL divergence [2]. As shown below, the results demonstrate that the complex approximation leads to unstable optimization and noticeably worse performance in asymmetric modeling. You can refer to Appendix C.4.5 for more details.
> - Therefore, the symmetric Laplace model provides a more efficient and practical choice, with only minor loss in fidelity to the empirical Q bias distribution. While asymmetric Laplace modeling is more flexible, its complex and unstable computation outweighs the potential gains in our settings.
>
> |                           | Symmetric Laplace modeling (LAROO) | Asymmetric Laplace modeling |
> | ------------------------- | ---------------------------------- | --------------------------- |
> | Walker2d-medium           | **120.4 $\pm$ 2.9**                | 39.8 $\pm$ 13.4             |
> | Walker2d-medium-expert    | **126.7 $\pm$ 1.0**                | 29.4 $\pm$ 3.9              |
> | Halfcheetah-medium        | **92.5 $\pm$ 1.2**                 | 59.7 $\pm$ 9.5              |
> | Halfcheetah-medium-expert | **99.5 $\pm$ 1.8**                 | 58.8 $\pm$ 7.8              |
> | Hopper-medium             | **106.7 $\pm$ 2.6**                | 63.0 $\pm$ 22.1             |
> | Hopper-medium-expert      | **112.3 $\pm$ 0.7**                | 24.7 $\pm$ 2.2              |
>
>
>
> **Q2**.  (Following Q1) Indeed, Figure 8 indicates that LAROO reduces the positive tail but may over-correct by introducing spurious negative bias.
>
> **A2**. We sincerely appreciate the reviewer's insightful observation regarding the negative bias in Figure 8. We explain this phenomenon mainly from two parts:
>
> - **Where the negative bias comes from:** The negative bias primarily arises from the ensemble component rather than the symmetric Laplace modeling. Because the ensemble model adopts a conservative estimation strategy (i.e., taking the minimum value across a subset of ensembles), it may introduce significantly negative Q-bias. This observation is supported by the experimental results in Figure 14, where the Q bias `without the ensemble model` (i.e., only use the KL loss based on symmetric Laplace modeling) exhibits minor negative distribution regions.
>
> - **How LAROO mitigates this negative bias:** The symmetric Laplace noise model can capture the heavy-tailed characteristics arising from both large positive or negative biases, and reduces the variance of Q-value estimation in practice, thereby decreasing the occurrence of large negative bias. Meanwhile, in the following Answer **A5**, we provide a theoretical analysis that LAROO achieves lower Q-value variance than the standard L2 loss. Consequently, the Laplace modeling helps mitigate the negative bias introduced by the ensemble component.
>
>
>
> [1] Exact Expressions for Kullback–Leibler Divergence for Univariate Distributions.
>
> [2] Automatic differentiable Monte Carlo: Theory and application.

---

> ### Author Response · Authors · 2025-11-22
>
> **Q3**. (Limited justification for ensemble-based correction) The proposed method integrates a random subset selection from ensemble Q-functions to re-center Q-bias, but it is somewhat heuristic. This paper lacks an intuitive explanation or theoretical reasoning as to why this mechanism effectively re-centers the bias.
>
> **A3**. Thank you for the valuable comment. We have strengthened the intuitive explanation for the ensemble-based Q-bias correction in the revised manuscript.
>
> - In Q-learning, Q-functions tend to exhibit overestimation mainly because the max-operator in the Bellman backup amplifies positive random errors [3]. It causes the Q bias to have a positive mean far above zero.
>
> - Intuition for subset selection: When we take a random subset of the ensemble Q-functions, the Q bias re-center to zero because: (1) different Q-functions are approximately independent, the positive errors and negative errors are independently sampled across different heads. (2) the probability of selecting the most overoptimistic head is reduced. Thus, the resulting Q-bias distribution naturally shifts toward zero, i.e., re-center the bias.
>
> - Theoretical support. This ensemble strategy follows prior work [3], which has provided theoretical support for bias reduction through subset selection in ensembles. In particular,  Theorem 1 in [3] shows:
>
>   - $\mathcal{E}[Z_{1,N}] \geq 0$ for all $N \geq 0$
>
>   - $\mathcal{E}[Z_{M+1,N}] \leq \mathcal{E}[Z_{M,N}] $ for any $M < N$.
>
>   - Here, the $\mathcal{E}[Z_{M,N}]$ denotes the expected post-update bias when selecting a subset of size $M$ from an ensemble of size $N$.  Statement (1) indicates that the Q-bias is inherently positive (overestimation), while statement (2) shows that increasing the subset size monotonically reduces the positive bias, providing a formal justification for our re-centering mechanism.
>
>
>
> [3] Randomized Ensembled Double Q-Learning: Learning Fast Without a Model. ICLR 2021.
>
>
>
> **Q4**. (Q-bias analysis across limited domains) The analysis of Q-bias and its heavy-tailed behavior is restricted to dense-reward environments such as MuJoCo. It remains unclear whether similar heavy-tailed characteristics would emerge in sparse-reward domains (e.g., AntMaze) or semi-sparse domains (e.g., Adroit, OGBench).
>
> **A4**. Thank you for your valuable comments.
>
> - We conduct experiments in sparse-reward domains `AntMaze` and in semi-sparse domains `Adroit`, and present the distribution of Q bias in Figure 17, 18  in our revision. The experimental results validate that the heavy-tailed issue of Q bias generally exists across the sparse-reward domains. This further enhances the general applicability of our research in O2O RL settings.  Please refer to Appendix D.1 for details.
>   - Specifically, in AntMaze tasks, we evaluate the distribution of Q-bias across three tasks (`large-play`, `medium-play`, `medium-diverse`) at training steps (50K, 100K) under three O2O methods (Cal-QL, PEX, SO2).
>   - In Adroit tasks, we further examine the Q bias distribution on the `pen-cloned` and `pen-human` tasks at 50K and 100K steps.
> - Additionally, we explore why Q-bias exhibits heavy tails in offline-to-online (O2O) RL through experiments and attribute it to two factors: (i) the Q-function with Max operator amplifies overestimation by selecting a large bellman target, and (ii) a non-uniform distribution shift between online and offline data. Concretely, pretrained Q-value networks often have large Q bias in online samples that lie far from the offline data support, which manifest as the Q bias distribution's tails. In contrast, online samples closer to the offline distribution result in minor biases and constitute the central mass of Q bias distribution.
>   - We conduct experiments to validate the factor (ii). We compute the shifted distance of online data to offline distribution with two metrics, and validate a positive correlation between Q bias and shifted distance with Spearman's rank correlation coefficient. Please refer to Figure 7 and Appendix A.3 for details in the updated manuscript.

---

> ### Author Response · Authors · 2025-11-22
>
> **Q5**. (Theoretical and empirical disconnect) The connection to heavy-tailed variance reduction (the core empirical claim) remains indirect. It would strengthen the contribution to include a theoretical link between the Laplace modeling and variance-bounded Q estimation under heavy-tailed noise.
>
> **A5**. Thank you for your valuable and insightful comments.
>
> - We provide the theoretical analysis of LAROO in variance reduction and give the Theorem B.4 as follows:
>
>   - With Assumption 4.1 and 4.2, when $b > 1$, the variance of post-update Q-value $Q_{\theta_{new}}$ with $D_b(x)$ function is smaller than that with $l_2$ loss function.
>
> - The proof is structured as follows:  we first provide the necessary Lemma B.3, and then present the Theorem B.4 in variance reduction. You can refer to Appendix B.5 for more details.
>
>   Lemma B.3: Assume three random variables $X$, $Y$ and $Z$. $Y$ is independent from $X$ and $Z$. $X$ and $Z$ are not independent, i.e., $\text{Cov}(X,Z) \neq 0$. Then, the variance of $X+YZ$ is:
>
> $$
> \begin{aligned}
>       \text{Var}(X+YZ)= & \text{Var}(X)+\text{Var}(Y)\text{Var}(Z)+\text{Var}(Y)(E[Z])^2 + \\\\
>       & \text{Var}(Z)(\mathbb{E}[Y])^2 +2\mathbb{E}[Y]\text{Cov}(X,Z)
>   \end{aligned}
> $$
>
> Based on the Lemma, we introduce the proof of Theorem B.4:
>
> Proof. We first show the variance of Q-value estimation with $l_2$ loss function. Following Equation 25,
>
> $Q_{\theta_{new}}(s,a)\approx Q_\theta(s,a)+\beta \big(y-Q_\theta(s,a) \big)\|\nabla_\theta Q_\theta(s,a)\|_2^2$
>
> Where $y$ is the expected Bellman target. With Assumption 4.1, we can have
>
> $y-Q_\theta(s,a) = \big(\mathcal{Q}(s,a)- \varepsilon_{\hat \theta}\big) - \big(\mathcal{Q}(s,a)- \varepsilon_{\theta}\big) = \varepsilon_{\theta} - \varepsilon_{\hat \theta}$
>
> Under Assumption 4.1 on the independent approximation noise, $\varepsilon_{\theta}$ and $\varepsilon_{\hat \theta}$ are independent of each other, and both are independent of $s$, $a$ and $\theta$. Therefore, the variable $\varepsilon_{\theta}-\varepsilon_{\hat \theta}$ is independent from $Q_{\theta}(s,a)$ and $\|\nabla_\theta Q_\theta(s,a)\|_2^2$.
>
> Let $X = Q_{\theta}(s,a)$, $Y = \varepsilon\_{\theta} - \varepsilon\_{\hat{\theta}}$, and $Z = \|\nabla_{\theta} Q_{\theta}(s,a)\|_2^2$.
>
> The noise $\varepsilon_{\theta}$ and $\varepsilon_{\hat \theta}$ have the same mean.  $\mathbb{E}(Y)=\mathbb{E}(\varepsilon_{\theta} - \varepsilon_{\hat \theta})=0$. For simplicity, we denote the mean and variance of $\|\nabla_\theta Q_\theta(s,a)\|_2^2$ as $\mathbb{E}[\nabla]$ and $\text{Var}(\nabla)$, respectively.
>
> According to Lemma B.3 above, the variance of post-update Q-values with $l_2$ function $\text{Var}(Q_{\theta_{news}}(s,a))$ is as follows, denoted as $\text{Var}_{l_2}(Q)$:
>
> $$
> \text{Var}_{l_2}(Q) = \text{Var}(Q\_{\theta}(s,a)) + \beta^2\text{Var}(\varepsilon\_{\theta} - \varepsilon\_{\hat \theta}) \text{Var}(\nabla) + \beta^2 \text{Var}(\varepsilon\_{\theta} - \varepsilon\_{\hat \theta}) (\mathbb{E}[\nabla])^2
> $$
>
> Then, we demonstrate the variance of Q-value estimation with $D_b(x)$ loss. We denote it as $\text{Var}\_{D_b}(Q)$.  Similarly, let $X=Q_{\theta}(s,a)$, $Y=D^{\prime}\_b(\varepsilon\_{\theta} - \varepsilon\_{\hat \theta})$, $Z = \|\nabla\_{\theta} Q\_{\theta}(s,a)\|\_2^2$. According to the $Q_{\theta_{new}}(s,a)$ derived with $D_b(x)$, we can get $\text{Var}_{D_b}(Q)$ as:
>
> $$
> \begin{aligned}
> & Q\_{\theta\_{new}}(s,a)  \approx Q\_\theta(s,a)+\beta D\_b^{\prime}\big(y-Q\_\theta(s,a)\big)\|\nabla\_\theta Q\_\theta(s,a)\|\_2^2 \\\\
> & \text{Var}\_{D_b}(Q) =  \text{Var}(Q_{\theta}(s,a)) + \beta^2 \text{Var}\big(  D_b^{\prime}(\varepsilon_{\theta} - \varepsilon_{\hat{\theta}})\big) \text{Var}(\nabla) + \beta^2 \text{Var}\big(  D_b^{\prime}(\varepsilon_{\theta} - \varepsilon_{\hat{\theta}})\big) (\mathbb{E}[\nabla])^2
> \end{aligned}
> $$
> Next, with Assumption 4.2 of Laplace-based noise, the variance of $\varepsilon\_{\theta}$, and $\varepsilon\_{\hat \theta}$ can be approximated with the scale parameter of Laplace distribution, i.e., $\text{Var}(\varepsilon\_{\theta}) = \text{Var}(\varepsilon\_{\theta}) = 2b^2$. We have $\text{Var}(\varepsilon\_{\theta} - \varepsilon\_{\hat \theta}) = 4b^2$.
>
> On the other hand, the $ |D_b^{\prime}(x)| < 1/b$, the $D_b^{\prime}(\varepsilon_{\theta} - \varepsilon_{\hat{\theta}}) \in (-1/b,1/b)$, so the $\text{Var}\big( D\_b^{\prime}(\varepsilon\_{\theta} - \varepsilon\_{\hat{\theta}})\big) < 1/b^2$ according to Popoviciu's inequality.
>
> Finally, we can compare $\text{Var}\_{D\_b}(Q)$ with  $\text{Var}\_{l\_2}(Q)$:
>
> $$
> \begin{aligned}
>     \text{Var}\_{D\_b}(Q) - \text{Var}\_{l\_2}(Q)
>     &= \beta^2\Big(\text{Var}\big(  D\_b^{\prime}(\varepsilon\_{\theta} - \varepsilon\_{\hat{\theta}})\big) - \text{Var}(\varepsilon\_{\theta} - \varepsilon\_{\hat \theta}) \Big) \big(\text{Var}(\nabla) + (\mathbb{E}[\nabla])^2\big) \\\\
>     & < (1/b^2 - 4b^2) \big(\text{Var}(\nabla) + (\mathbb{E}[\nabla])^2\big) \\\\
>     & < 0
> \end{aligned}
> $$
>
> Proof is done.

---

> > ### Author Response · Authors · 2025-11-27
> > **Eagerly await your valuable feedback.**
> >
> > Dear Reviewer U4Kt,
> >
> >   We deeply appreciate your valuable feedback and the time you've taken to review our work, especially during this busy period.
> >
> >   We are reaching out to kindly inquire about the current status of your review regarding our submission. We sincerely hope that our responses have adequately addressed your concerns. Furthermore, we are eager to address any additional queries you might have, which will enable us to enhance our work further.
> >
> >   Once again, thank you for your guidance and support.
> >
> >   Best, Authors

---

### Official Review · Reviewer_2zgx · 2025-11-01

**Soundness:** 3
**Presentation:** 3
**Contribution:** 3
**Rating:** 6
**Confidence:** 3

**Summary:**

This paper identifies that the Q-value estimation error during offline-to-online (O2O) RL exhibits a heavy-tailed distribution, and proposes to address this using Laplace-distributed noise modeling. Concretely, a new loss function is derived that is less sensitive to outliers. The proposed method achieves state-of-the-art performance on O2O RL benchmarks.

**Strengths:**

- The investigation of the Q-estimation error distribution is thorough and convincing.
- The derivation of the Laplace-based robust loss function (D_b) is concise yet effectively mitigates the impact of outlier errors.
- The experiment results show that the proposed method consistently outperforms the considered baselines.

**Weaknesses:**

- It would be helpful for the authors to examine whether the heavy-tailed error distribution only arises in the O2O RL setting. Similar distributions may also appear in other RL paradigms, such as purely online or offline RL. If that is the case, applying the proposed Laplace-based modeling to those settings could further validate its generality and effectiveness in handling heavy-tailed errors.

**Questions:**

- Could the authors also evaluate the proposed method on the D4RL Kitchen benchmark, which, similar to Adroit, is a challenging environment in D4RL?

---

> ### Author Response · Authors · 2025-11-22
>
> We appreciate the reviewer's insightful and constructive comments and suggestions. We respond to each comment as follows and accordingly revise our manuscript, with all updates highlighted in *blue* for your convenience. We sincerely hope that our responses could properly address your concerns. If so, we would deeply appreciate it if you could raise your score. If not, please let us know your further concerns, and we will continue actively responding to the comments and enhancing our submission.
>
> **Q1**. It would be helpful for the authors to examine whether the heavy-tailed error distribution only arises in the O2O RL setting. Similar distributions may also appear in other RL paradigms, such as purely online or offline RL. If that is the case, applying the proposed Laplace-based modeling to those settings could further validate its generality and effectiveness in handling heavy-tailed errors.
>
> **A1**. Thanks for your constructive and valuable suggestions. We examine the distribution of Q bias in purely online and offline RL, and present these distributions in the revised manuscript. We set four checkpoint steps (100K, 300K, 600K and 1M) to fully observe the evolving distribution of Q bias during training. You can refer to Appendix D2, D3 for more details.
>
> - In online RL, we examine the Q bias of SAC and TD3 in three MuJoCo tasks (`Halfcheetah`, `Walker2d` and `Hopper`) and find significant distributional patterns during different training stages.
>
>   - Early stage: due to the limited accuracy of the Q-value network at the beginning, Q bias exhibits complex and irregular distributional characteristics, such as multi-peak and high dispersion, making it difficult to fit with the Laplace distribution.
>   - Middle stage: as training progresses, we observe the emergence of heavy-tailed Q bias mainly due to systematic overestimation.
>   - Final stage: the Q bias gradually becomes concentrated and Gaussian-like, which aligns with the expected convergence of Q-value estimates.
>
> - In offline RL, we examine the Q bias of TD3BC, CQL and IQL in MuJoCo tasks at different training steps.
>
>   - Specifically, for each state-action pair in offline dataset, we treat it as the starting point, roll out a full episode from that pair and get its true Q-value using Monte Carlo returns. We then compare this true Q-value with the network’s estimate to compute the Q-bias.
>   - The results demonstrate that Q bias often shows a heavy-tailed behavior in offline RL, exhibiting negative skewness with notable biases in the left tail (i.e., large underestimation). This pattern largely contributes to conservative Q-value estimation in offline RL, which introduces explicit conservative penalties that push Q-values downward.
>
> - We further evaluate whether LAROO remains effective in offline/online RL settings. The experimental results show only modest improvements in final normalized returns. In offline RL, the strictly conservative nature of baseline algorithms makes it difficult for LAROO to correct extremely low Q-bias. In online RL, the irregular and highly complex nature of Q-bias, especially in early training, makes it challenging to fit well with the Laplace or other distribution. In summary, offline/online RL does not induce a distribution shift that contributes to heavy-tailed Q bias throughout the training process, limiting LAROO's advantage in these settings.
>
>
>
>   - | Task\method               | TD3BC           | TD3BC + LAROO   | CQL             | CQL + LAROO     | IQL             | IQL + LAROO     |
>     | ------------------------- | --------------- | --------------- | --------------- | --------------- | --------------- | --------------- |
>     | Walker2d-medium           | 82.7 $\pm$ 4.5  | 87.1 $\pm$ 6.2  | 80.7 $\pm$ 3.3  | 82.4 $\pm$ 3.1  | 80.9 $\pm$ 3.2  | 80.6 $\pm$ 4.2  |
>     | Walker2d-medium-replay    | 85.6 $\pm$  5.2 | 89.5 $\pm$ 3.2  | 76.1 $\pm$ 13.2 | 88.2 $\pm$ 4.9  | 82.1 $\pm$ 3.0  | 79.8 $\pm$ 2.0  |
>     | Walker2d-medium-expert    | 110.3 $\pm$ 1.0 | 111.5 $\pm$ 1.7 | 109.5 $\pm$ 0.4 | 115.2 $\pm$ 2.3 | 111.7 $\pm$ 0.9 | 113.5 $\pm$ 1.1 |
>     | Halfcheetah-medium        | 48.1 $\pm$ 0.2  | 55.1 $\pm$ 0.8  | 47.0 $\pm$ 0.2  | 49.4 $\pm$ 3.1  | 48.3 $\pm$ 6.8  | 51.0 $\pm$ 4.2  |
>     | Halfcheetah-medium-replay | 44.8 $\pm$ 0.6  | 50.8 $\pm$ 3.2  | 47.1 $\pm$ 1.7  | 51.2 $\pm$ 0.9  | 44.5 $\pm$ 0.8  | 46.2 $\pm$ 2.0  |
>     | Halfcheetah-medium-expert | 91.3 $\pm$ 6.1  | 95.9 $\pm$ 3.7  | 95.6 $\pm$ 0.4  | 99.1 $\pm$ 2.3  | 94.7 $\pm$ 0.5  | 96.3 $\pm$ 1.1  |
>
>   - | Task\method    | TD3            | TD3 + LAROO    | SAC             | SAC + LAROO     |
>     | -------------- | -------------- | -------------- | --------------- | --------------- |
>     | Walker2d-v2    | 89.3 $\pm$ 1.3 | 94.8 $\pm$ 1.7 | 114.5 $\pm$ 2.2 | 115.2 $\pm$ 2.3 |
>     | Halfcheetah-v2 | 54.3 $\pm$ 1.7 | 59.3 $\pm$ 0.9 | 70.1 $\pm$ 4.2  | 70.3 $\pm$ 3.3  |
>     | Hopper-v2      | 93.0 $\pm$ 5.3 | 99.5 $\pm$ 1.4 | 105.5 $\pm$ 7.1 | 110.2 $\pm$ 3.5 |

---

> ### Author Response · Authors · 2025-11-22
>
> **A1.2** Following the answer above,
>   - We further investigate two factors contributing to the heavy-tailed Q bias in O2O RL scenarios: (i) the Q-function with Max operator amplifies overestimation by selecting a large bellman target, and (ii) a non-uniform distribution shift between online and offline data. That is, pretrained Q-value networks often have large Q bias in online samples that lie far from the offline data support, which manifest as the Q bias distribution's tails. In contrast, online samples closer to the offline distribution result in minor biases and constitute the central mass of Q bias distribution. We conduct experiments to validate the reason (ii). You can refer to Appendix A.3 for more details.
>     - We validate the reason (ii) in offline-to-online settings through experiments. We compute the shifted distance of online data to offline distribution with two metrics, and validate a positive correlation between Q bias and shifted distance with Spearman's rank correlation coefficient. Please refer to Figure 7 and Appendix A.3 for details in the updated manuscript.
>
> **Q2**. Could the authors also evaluate the proposed method on the D4RL Kitchen benchmark, which, similar to Adroit, is a challenging environment in D4RL?
>
> **A2**. Thanks for your valuable suggestions. We conduct additional experiments in three Kitchen-v0 tasks (`Kitchen-complete`, `Kitchen-partial` and `Kitchen-mixed`). Each experimental result is averaged over three random seeds. The results are shown as follows. LAROO outperforms the second-best baseline SO2 by **76.9%**, which further confirms the superiority of LAROO in training stability and performance improvement in O2O RL.
>
> | Task                                   | PEX             | SO2             | Cal-QL          | ENOTO               | LAROO               |
> | -------------------------------------- | --------------- | --------------- | --------------- | ------------------- | ------------------- |
> | Kitchen-complete-v0                    | 47.5 $\to$ 40.0 | 37.5 $\to$ 39.2 | 42.5 $\to$ 42.8 | 43.6 $\to$ **43.7** | 37.5 $\to$ **44.0** |
> | Kitchen-partial-v0                     | 35.0 $\to$ 30.4 | 60.0 $\to$ 63.8 | 32.5 $\to$ 36.1 | 39.5 $\to$ 46.5     | 60.0 $\to$ **65.0** |
> | Kitchen-mixed-v0                       | 43.5 $\to$ 47.5 | 61.3 $\to$ 66.2 | 40.0 $\to$ 45.5 | 50.8 $\to$ 53.4     | 61.3 $\to$ **68.2** |
> | $\delta_{\mathrm{sum}}(0.1\mathrm{M})$ | -8.1            | 10.4            | 9.4             | 9.7                 | **18.4**            |

---

> ### Author Response · Authors · 2025-11-27
> **Eagerly await your valuable feedback.**
>
> Dear Reviewer 2zgx,
>
> We deeply appreciate your valuable feedback and the time you've taken to review our work, especially during this busy period.
>
> We are reaching out to kindly inquire about the current status of your review regarding our submission. We sincerely hope that our responses have adequately addressed your concerns. Furthermore, we are eager to address any additional queries you might have, which will enable us to enhance our work further.
>
> Once again, thank you for your guidance and support.
>
> Best, Authors

---

### Author Response · Authors · 2025-12-02
**Summary for Area Chairs**

Dear ACs and PCs,

Thank you for your time and effort in handling our submission, "Tackling Heavy-Tailed Q-Value Bias in Offline-to-Online Reinforcement Learning with Laplace-Robust Modeling" (ID: 24168).

We deeply regret the recent information leak within the community. To provide a clear and concise overview of the author-reviewer discussion, we have summarized the key points as follows.

**Strengths of Our Approach**.

- **Significant observation of heavy-tailed Q bias**. This work reveals that the Q-value estimation bias often follows a heavy-tailed distribution in offline-to-online (O2O) RL for the first time. Reviewers have praised this observation, stating it "thorough and convincing" (`Reviewer 2zgx`), "significant and previously unreported" (`Reviewer U4Kt`) and "novel" (`Reviewer SoSF`).
- **Well-motivated and Effective Method**. Our paper introduces LAROO, which employs a parametric Laplace-based noise model to capture the heavy-tailed bias and reduces the impact of extreme biases for training stability and performance. This method is "well-motivated" (`Reviewer U4Kt`) and "concise yet effectively" (`Reviewer 2zgx`).
- **Theoretical Foundation**.  LAROO provides rigorous theoretical analysis and clear proofs in reducing estimation bias, further supporting the observed improvements (`Reviewers U4Kt, SoSF and qzBg`).
- **Comprehensive Experiments and High Performance**. Extensive experimental results demonstrate superior performance over existing O2O methods across multiple environments (`Reviewers 2zgx, SoSF and qzBg`).
- **Plug-in compatibility**. LAROO "can be used as a plug-in component for existing O2O methods" to further enhance their performance (`Reviewer U4Kt`).
- **Well-Written paper**. This paper was found to be "well-written" with clear motivations and "easy to follow" (`Reviewer SoSF`).

**Major Concerns and Our Responses**.

1. **Reviewer 2zgx**  requested more empirical experiments: (1) evaluating LAROO on the D4RL Kitchen benchmark, and (2) testing Q bias in offline and online RL to assess whether LAROO can be extended to more RL settings.

   - For (1), we conduct experiments on three Kitchen-v0 tasks, where LAROO outperforms the second-best baseline by **76.9%**.

     For (2), we run offline and online experiments on four MuJoCo tasks to evaluate Q bias of five methods, showing that unlike O2O RL, offline/online RL does not induce a distribution shift that causes heavy-tailed Q bias, limiting LAROO's advantage in these settings.

2. **Reviewer SoSF** raised concerns about unclear motivations in Eq. (4), and misunderstood assumptions about TD-errors.

   - We clarify the motivations and assumptions with rigorous analysis, highlighting LAROO's advantage in preserving the heavy-tailed information of bias during Q-value updates.

   - The Reviewer was satisfied with our response and **raised the score from 4 to 6, resulting in an averaging rate of 5.5**.  We kindly believe that **this mild improvement is trustworthy**, as it occurred following a comprehensive discussion before November 26th, **prior to the large leak event**.

3. **Reviewer qzBg** expressed concerns about (1) validating the contribution of the Laplace noise model to training stability, and (2) comparing the contribution of ensemble-model with noise model.

   - For (1),  we show that the Laplace noise model reduces variance by **76.1%** and mitigates performance drop (NCD metric [1]) by **18.9%**  in six tasks, significantly improving stability.

     For (2),  `LAROO without noise model` performs worse than `LAROO without ensemble model` in 8 out of 9 MuJoCo tasks, indicating that the noise model is more critical.

   - Reviewer qzBg acknowledged our rebuttal with **"addressed my concerns"** and maintained the score.

4. **Reviewer U4Kt** raised concerns about (1) disregarding the asymmetric property of Q bias, (2) validating heavy-tailed Q bias in sparse-reward domains, and (3) the need for analysis in estimation variance.

   - For (1), we validate that explicitly modeling the asymmetric property of Q bias results in worse performance than LAROO, due to added computational complexity and instability.

     For (2), we present that Q bias exhibits heavy-tailedness across five sparse-reward tasks.

     For (3), we provide a theoretical analysis of **LAROO in reducing estimation variance**.

   - We believe that our responses **have adequately addressed Reviewer U4Kt's concerns**.  Since the **reviewer was absent** during discussions, we kindly remind the AC to place less emphasis on his rating in the final decision.

   [1] Feng et al. Suf: Stabilized unconstrained fine-tuning for offline-to-online reinforcement learning. AAAI 2024.

In response to reviewers' feedback, we have addressed all raised points and made significant enhancements in our manuscript.

Thank you once again for your dedicated efforts, especially during this challenging time. We look forward to your final decision.

Best,

Authors

---

### Meta-Review · Area_Chair_Xzxb · 2026-01-06

**Summary:**

This paper makes a clear empirical observation in offline-to-online RL: Q-value estimation bias during fine-tuning often exhibits a heavy-tailed distribution, and proposes LAROO, which replaces the standard L2 Bellman loss with a Laplace-derived robust loss and combines it with an ensemble-based re-centering mechanism. The approach is conceptually simple (a loss swap + a bias-correction trick), and the reported results show consistent gains on standard O2O benchmarks, with additional evidence added during rebuttal to broaden and strengthen the empirical story.

**Reviewer Concerns:**

The main initial weaknesses—generality beyond O2O, missing Kitchen evaluation, symmetry vs. skewness of Q-bias, disentangling ensemble vs. noise-model contribution, and clarifying why this is not just L1/Huber—were addressed with targeted additions: the authors measured Q-bias distributions in pure online and offline RL (showing O2O has a more persistent heavy-tailed regime), added Kitchen results with strong gains, evaluated sparse/semi-sparse domains (AntMaze/Adroit) to confirm heavy tails, provided ablations isolating “w/o noise” vs “w/o ensemble” including stability metrics/curves, and compared against L1 and Huber to justify the Laplace-KL formulation. The symmetry concern is handled pragmatically: asymmetric Laplace KL is expensive/unstable in their setup, and empirical tests suggest it underperforms, though this should be clearly communicated as a trade-off rather than an ideal modeling choice.

**Reviewer Scores:**

The reviewer set includes one marginal accept and several borderline scores that specifically requested extra evaluations and ablations; after rebuttal, at least one reviewer explicitly raised their score, and another maintained a positive score after the stability ablations addressed their key concern. Given the breadth of added experiments (Kitchen + sparse domains + ablations + L1/Huber comparisons) and the fact that concerns were concrete and directly answered, I would expect the overall trajectory to land on the accept side, with remaining skepticism largely about modeling ideality (symmetry) rather than effectiveness.

---

### Decision · Program_Chairs · 2026-01-26

Accept (Poster)